# Zebra: In-Context and Generative Pretraining for Solving Parametric PDEs

## Abstract

Solving time-dependent parametric partial differential equations (PDEs) is challenging, as models must adapt to variations in parameters such as coefficients, forcing terms, and boundary conditions. Data-driven neural solvers either train on data sampled from the PDE parameters distribution in the hope that the model generalizes to new instances or rely on gradient-based adaptation and meta-learning to implicitly encode the dynamics from observations. This often comes with increased inference complexity. Inspired by the in-context learning capabilities of large language models (LLMs), we introduce `Zebra`, a novel generative autoregressive transformer designed to solve parametric PDEs without requiring gradient adaptation at inference. By leveraging in-context information during both pre-training and inference, `Zebra` dynamically adapts to new tasks by conditioning on input sequences that incorporate context trajectories or preceding states. This approach enables `Zebra` to flexibly handle arbitrarily sized context inputs and supports uncertainty quantification through the sampling of multiple solution trajectories. We evaluate `Zebra` across a variety of challenging PDE scenarios, demonstrating its adaptability, robustness, and superior performance compared to existing approaches.

## 1 Introduction

Training partial differential equation (PDE) solvers is a challenging task due to the variety of behaviors that can arise in physical phenomena, and neural solvers have limited generalization capability(Chen et al., 2018; Raissi et al., 2019; Li et al., 2021). We tackle the parametric PDE problem (Cohen & Devore, 2015), where a model is trained on trajectories defined by varying PDE parameters with the goal of generalizing across a wide range of parameters. The parameters may include initial and boundary conditions, physical coefficients, and forcing terms. We focus on pure data-driven approaches that do not leverage any prior knowledge on the underlying equations.

A natural approach to this problem is to sample from the parameter distribution, i.e., to train using different PDE instances or parameter values, along with multiple trajectories for each PDE instance. This requires a training set representative of the distribution of the underlying dynamical system, which is difficult to meet in practice given the complexity of physical phenomena. Other approaches explicitly condition on specific PDE parameters, (Brandstetter et al., 2022b; Takamoto et al., 2023) relying on the availability of such prior knowledge. This requires a physical model of the observed system, making the incorporation of PDE parameters into neural solvers challenging beyond basic PDE coefficients. An alternative approach involves online adaptation to new PDE instances by leveraging observations from novel *environments*. Here we consider that an environment is charaterized by a set of parameters. This adaptation is often implemented through meta-learning, where the model is trained on a variety of simulations corresponding to different environments—i.e., varying PDE parameter values—so that it can quickly adapt to new, unseen PDE simulation instances using a few trajectory examples(Kirchmeyer et al., 2022; Yin et al., 2022). This method offers a high flexibility but requires gradient updates for adaptation, adding computational overhead. Another common setting involves leveraging historical data to condition the neural network, allowing it to generalize to new PDE instances without retraining (Li et al., 2021; McCabe et al., 2023). Again the generalization ability is limited to dynamics close to the ones used for training. Exploring another direction and motivated by the successes encountered in natural language processing and vision, some authors have begun investigating the development of foundation models for spatio-temporal dynamic physical processes (Subramanian et al., 2023; Herde et al., 2024; McCabe et al., 2023).

This approach involves training a large model on a variety of physics-based numerical simulations with the expectation that it will generalize to new situations or equations. While they consider multiple physics we focus on solving parametric PDEs, i.e. multiple variations of the same physical phenomenon.

We explore here a new direction inspired by the successes of in-context learning (ICL) and its ability to generalize to downstream tasks without retraining (Brown et al., 2020; Touvron et al., 2023). We propose a framework, denoted `Zebra`, relying on *in-context pretraining* (ICP), for solving parametric PDEs and learning to condition neural solvers to adapt fast to new situations or said otherwise for solving for new parameter values. As for ICL in language the model is trained to generate appropriate responses given context examples and a query. The context examples could be trajectories from the same dynamics starting from different initial conditions, or simply a brief history of past system states for the target trajectory. The query will consist for example of an initial state condition, that will serve as inference starting point for the forecast. This approach offers key advantages compared to existing methods. It can leverage contexts of different types and sizes, it requires only a few context examples to adapt to new dynamics and can handle as well 0-shot learning. It allows us to cover a large variety of situations.

On the technical side, `Zebra` introduces a novel generative autoregressive solver for parametric PDEs. It employs an encode-generate-decode framework: first, a vector-quantized variational autoencoder (VQ-VAE) (Oord et al., 2017) is learnt to compress physical states into discrete tokens and to decode it back to the original physical space. Next, a generative autoregressive transformer is pre-trained using a next token objective. To leverage the in-context properties of the model, `Zebra` is directly pretrained on arbitrary-sized contexts such as extra trajectories or historical states of the target dynamics. At inference, `Zebra` can handle varying context sizes for conditioning and support uncertainty quantification, enabling generalization to unseen PDE parameters without gradient updates.

Our main contributions include:

- We introduce a generative autoregressive transformer for modeling physical dynamics. It operates on compact discretized representations of physical state observations. This discretization is performed through a VQ-VAE. The encoder tokenizes observations into sequences of tokens, while the decoder reconstructs the original states. This framework represents the first successful application of generative modeling using quantized representations of physical systems.

- To harness the in-context learning strengths of autoregressive transformers, we develop a new pretraining strategy that conditions the model on historical states or example trajectories with similar dynamics, allowing it to handle arbitrary-sized context token inputs.

- We evaluate `Zebra` on a range of parametric PDEs on two distinct settings. In the first, the model infers dynamics from a context trajectory that shares similar behavior with the target but differs in initial conditions, representing a one-shot setting. `Zebra`'s performance is benchmarked against domain-adaptation baselines specifically trained for such tasks. In the second scenario, only a limited number of historical frames of the target trajectory are available, requiring the model to deduce the underlying dynamics solely from these inputs. `Zebra` consistently demonstrates competitive performance across both evaluation contexts.

## 2 PROBLEM SETTING

### 2.1 SOLVING PARAMETRIC PDE

We aim to solve parametric time-dependent PDEs beyond the typical variation in initial conditions. Our goal is to train models capable of generalizing across a wide range of PDE parameters. To this end, we consider time-dependent PDEs with different initial conditions, and with additional degrees of freedom, namely: (1) coefficient parameters — such as fluid viscosity or advection speed — denoted by vector $\mu$ ; (2) boundary conditions $\mathcal{B}$, e.g. Neumann or Dirichlet; (3) forcing terms $\delta$, including damping parameter or sinusoidal forcing with different frequencies. To simplify notation we denote $\xi := \{\mu, \mathcal{B}, \delta\}$ and we define $\mathcal{F}_\xi$ as the set of PDE solutions corresponding to the

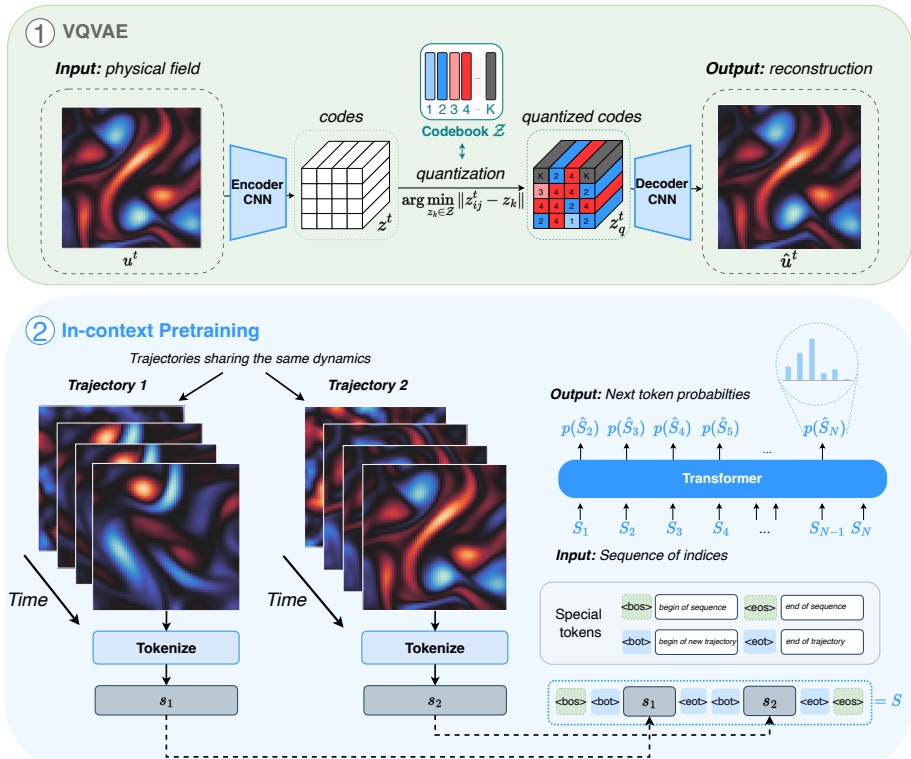

Figure 1: `Zebra` Framework for solving parametric PDEs. 1) A finite vocabulary of physical phenomena is learned by training a VQ-VAE on spatial representations. 2) During the pretraining, multiple trajectories sharing the same dynamics are tokenized and concatenated into a common sequence $S$. A transformer is used to predict the next tokens in these sequences, conditioned on the context. This enables the model to perform both zero-shot and few-shot generation, without gradient-based updates.

PDE parameters $\mu$, boundary conditions $\mathcal{B}$ and forcing term $\delta$, and refer to $\mathcal{F}_\xi$ as a PDE partition. Formally, a solution $\boldsymbol{u}(x, t)$ within $\mathcal{F}_\xi$ satisfies:

$$\frac{\partial \boldsymbol{u}}{\partial t} = F\left(\delta, \mu, t, x, \boldsymbol{u}, \frac{\partial \boldsymbol{u}}{\partial x}, \frac{\partial^2 \boldsymbol{u}}{\partial x^2}, \dots\right), \quad \forall x \in \Omega, \forall t \in (0, T] \tag{1}$$

$$\mathcal{B}(\boldsymbol{u})(x, t) = 0, \quad \forall x \in \partial\Omega, \forall t \in (0, T] \tag{2}$$

$$\boldsymbol{u}(0, x) = \boldsymbol{u}^0, \quad \forall x \in \Omega \tag{3}$$

where $F$ is a function of the solution $\boldsymbol{u}$ and its spatial derivatives on the domain $\Omega$, and also includes the forcing term $\delta$ ; $\mathcal{B}$ is the boundary condition constraint (e.g., spatial periodicity, Dirichlet, or Neumann) that must be satisfied at the boundary of the domain $\partial\Omega$; and $\boldsymbol{u}^0$ is the initial condition sampled with a probability measure $\boldsymbol{u}^0 \sim \boldsymbol{p}^0(.)$.

## 2.2 GENERALIZATION FOR PARAMETRIC PDE

Solving time-dependent parametric PDEs requires developing neural solvers capable of generalizing to a whole distribution of PDE parameters. In practice, changes in the PDE parameters often lead to distribution shifts in the trajectories which makes the problem challenging. Different directions are currently being explored briefly reviewed below. We focus on pure data-driven approaches that do not make use of any prior knowledge on the equations. We make the assumption that the models are learned from numerical simulations so that it is possible to generate from multiple parameters. This emulates real situations where for example, a physical phenomenon is observed in different contexts.

**0-shot learning with temporal conditioning**  A first direction consists in adapting the classical ERM framework to parametric PDE solving by sampling multiple instances of a PDE, in the hope that this will generalize to unseen conditions in a 0-shot setting. It is usually assumed that for both learning and inference, a sequence of past states is provided as initial input to the model, leveraging its potential to infer the dynamics characteristics in order to forecast future values. The neural solver $\mathcal{G}_\theta$ is then conditioned by a sequence of past states for a trajectory $\boldsymbol{u}^{t-m\Delta t:t} := (\boldsymbol{u}^{t-m\Delta t}, \dots, \boldsymbol{u}^t)$ where $m \geq 1$. Depending on the architecture, this can be implemented by stacking the information in the channel dimension (Li et al., 2021), or by creating an additional temporal dimension as done in video prediction contexts (McCabe et al., 2023; Ho et al., 2022). This approach makes an implicit i.i.d. assumption on the training - test distributions which is often not met with dynamical phenomena. It offers a limited flexibility in cases where only limited historical context is accessible.

**Few-shot learning by fine tuning**  Another category of methods leverages fine tuning. As for the 0-shot setting above, a model is pretrained on a distribution of the PDE parameters. At inference, for a new environment, fine tuning is performed on a sample of the environment trajectories. This approach often relies on large fine tuning samples and involves updating all or a subset of parameters (Subramanian et al., 2023; Herde et al., 2024).

**Adaptive conditioning**  A more flexible approach relies on adaptation at inference time through meta-learning. It posits that a set of environments $e$ are available from which trajectories are sampled, each environment $e$ being defined by specific PDE parameter values (Zintgraf et al., 2019a; Kirchmeyer et al., 2022). The model is trained from a sampling from the environments distribution to adapt fast to a new environment. The usual formulation is to learn shared and specific environment parameters $\mathcal{G}_{\theta+\Delta\theta_\xi}$, where $\theta$ and $\Delta\theta_\xi$ are respectively the shared and specific parameters. At inference, for a new environment, only a small number of parameters $\theta_\xi$ is adapted from a small sample of observations.

Table 1: **Key distinctions with Baselines**. `Zebra` is the only method that supports both adaptive conditioning, temporal conditioning, and does not require gradient computations at inference.

| Method | Adaptive conditioning | Temporal conditioning | In-context |
|---|:---:|:---:|:---:|
| CAPE (Takamoto et al. (2023)) | ✗ | ✗ | ✗ |
| *Vanilla* CODA (Kirchmeyer et al. (2022)) | ✓ | ✗ | ✗ |
| MPP (McCabe et al. (2023)) | ✗ | ✓ | ✗ |
| `Zebra` | ✓ | ✓ | ✓ |

## 3 ZEBRA FRAMEWORK

We introduce `Zebra`, a novel framework designed to solve parametric PDEs through in-context learning and flexible conditioning. `Zebra` utilizes an autoregressive transformer to model partial differential equations (PDEs) within a compact, discrete latent space. A spatial CNN encoder is employed to map physical spatial observations into these latent representations, while a CNN decoder accurately reconstructs them. As illustrated in Figure 1, our pretraining pipeline consists of two key stages: 1) Learning a finite vocabulary of physical phenomena, and 2) Training the transformer using an in-context pretraining strategy, enabling the model to effectively condition on contextual information. At inference, `Zebra` allows both adaptive and temporal conditioning through in-context learning (Table 1).

### 3.1 LEARNING A FINITE VOCABULARY OF PHYSICAL PHENOMENA

In order to leverage the auto-regressive transformer architecture and adopt a next-token generative pretraining, we need to convert physical observations into discrete representations. To keep the modeling with the transformer computationnaly tractable, we do not quantize the observations directly but rather quantize compressed latent representations by employing a VQVAE (Oord et al., 2017).

Our encoder spatially compresses the input function $\boldsymbol{u}^t$ by reducing its spatial resolution $H \times W$ to a lower resolution $h \times w$ while increasing the channel dimension to $d$. This is achieved through a

convolutional model $\mathcal{E}_w$, which maps the input to a continuous latent variable $\mathbf{z}^t = \mathcal{E}_w(\boldsymbol{u}^t)$, where $\mathbf{z}^t \in \mathbb{R}^{h \times w \times d}$. The latent variables are then quantized to discrete codes $\mathbf{z}_q^t$ using a codebook $\mathcal{Z}$ of size $K = |\mathcal{Z}|$ and through the quantization step $q$. For each spatial code $\mathbf{z}_{[ij]}^t$, the nearest codebook entry $z_k$ is selected:

$$\mathbf{z}_{q,[ij]}^t = q(\mathbf{z}_{[ij]}^t) := \arg \min_{z_k \in \mathcal{Z}} \|\mathbf{z}_{[ij]}^t - z_k\|.$$

The decoder $\mathcal{D}_\psi$ reconstructs the signal $\hat{\boldsymbol{u}}^t$ from the quantized latent codes $\hat{\mathbf{z}}_q^t$. Both models are jointly trained to minimize the reconstruction error between the function $\boldsymbol{u}^t$ and its reconstruction $\hat{\boldsymbol{u}}^t = \mathcal{D}_\psi \circ q \circ \mathcal{E}_w(\boldsymbol{u}^t)$. The codebook $\mathcal{Z}$ is updated using an exponential moving average (EMA) strategy, which stabilizes training and ensures high codebook occupancy.

The training objective is:

$$\mathcal{L}_{\text{VQ}} = \frac{\|\boldsymbol{u}^t - \hat{\boldsymbol{u}}^t\|_2}{\|\boldsymbol{u}^t\|_2} + \alpha \|\text{sg}[\mathbf{z}_q^t] - \mathcal{E}_w(\boldsymbol{u}^t)\|_2^2,$$

where the first term is the Relative L2 loss commonly used in PDE modeling, and the second term is the commitment loss, ensuring encoder outputs are close to the codebook entries. The parameter $\alpha$, set to 0.25, balances the two components. Here, sg denotes the stop-gradient operation that detaches a tensor from the computational graph. We provide additional details on the architecture in Appendix C.

Once this training step is done, we can tokenize a trajectory $\boldsymbol{u}^{t:t+m\Delta t}$ by applying our encoder in parallel on each timestamp to obtain discrete codes $\mathbf{z}_q^{t:t+m\Delta t}$ and retrieve the corresponding index entries $s^{t:t+m\Delta t}$ from the codebook $\mathcal{Z}$. Similarly, we detokenize discrete indices with the decoder.

## 3.2 IN-CONTEXT MODELING

We design sequences that enable Zebra to perform in-context learning on trajectories that share underlying dynamics. To incorporate varying amounts of contextual information, we draw a number $n$ between 1 and $n_{\max}$, then sample $n$ trajectories sharing the same dynamics, each with $m$ snapshots starting from time $t$, denoted as $(\boldsymbol{u}_1^{t:t+m\Delta t}, \ldots, \boldsymbol{u}_n^{t:t+m\Delta t})$. These trajectories are tokenized into index representations $(s_1^{t:t+m\Delta t}, \ldots, s_n^{t:t+m\Delta t})$, which are flattened into sequences $s_1, \ldots, s_n$, maintaining the temporal order from left to right. In practice, we fix $n_{\max} = 6$ and $m = 9$.

Since our model operates on tokens from a codebook, we found it advantageous to introduce *special tokens* to structure the sequences. The tokens <bot> (beginning of trajectory) and <eot> (end of trajectory) clearly define the boundaries of each trajectory within the sequence. Furthermore, as we sample sequences with varying context sizes, we maximize the utilization of the transformer's context window by stacking sequences that could also represent different dynamics. To signal that these sequences should not influence each other, we use the special tokens <bos> (beginning of sequence) and <eos> (end of sequence). The final sequence design is:

$$S = \text{<bot>}[s_1]\text{<eot>}\text{<bot>}[s_2]\text{<eot>}\ldots\text{<bot>}[s_n]\text{<eot>}$$

And our pretraining dataset is structured as follows:

$$\text{<bos>}[S_1]\text{<eos>}\text{<bos>}[S_2]\text{<eos>}\ldots\text{<bos>}[S_l]\text{<eos>}$$

## 3.3 NEXT-TOKEN PRETRAINING

The transformer is trained using self-supervised learning on a next-token prediction task with teacher forcing (Radford et al., 2018). Given a sequence $S$ of discrete tokens of length $N$, the model is optimized to minimize the negative log-likelihood (cross-entropy loss):

$$\mathcal{L}_{\text{Transformer}} = -\mathbb{E}_S \sum_{i=1}^{N} \log p(S_{[i]}|S_{[i'<i]}),$$

where the model learns to predict each token $S_{[i]}$ conditioned on all previous tokens $S_{[i'<i]}$. Due to the transformer's causal structure, earlier tokens in the sequence are not influenced by later ones, while later tokens benefit from more context, allowing for more accurate predictions. This structure

naturally supports both generation in a zero-shot and few-shot setting within a unified framework. Our transformer implementation is based on the Llama architecture (Touvron et al. (2023)). Additional details can be found in Appendix C. Up to our knowledge, this is the first adaptation of generative auto-regressive transformers to the modeling of physical dynamics.

### 3.4 INFERENCE: FLEXIBLE CONDITIONING

In this section, we outline the inference pipeline for Zebra across various scenarios. For simplicity, we assume that all observations have already been tokenized and omit the detokenization process. Let $s_*$ represent the target token sequence for which we aim to predict the following timestamps.

- **Temporal conditioning** with $\ell$ frames: The prompt is structured as $S = $ `<bos><bot>`$[s_*^{0:\ell\Delta t}]$, and the transformer generates the subsequent tokens based on this input.

- **Adaptive conditioning** with $n$ examples and an initial condition: The prompt is structured as $S = $ `<bos><bot>`$[s_1^{0:m\Delta t}]$`<eot>`$\ldots$`<bot>`$[s_n^{0:m\Delta t}]$`<eot><bot>`$[s_*^0]$, allowing the model to adapt based on the provided examples and initial condition.

- **Adaptive conditioning** with $n$ examples and $\ell$ frames: This setup combines context from multiple trajectories with the initial timestamps, structured as $S = $ `<bos><bot>`$[s_1^{0:m\Delta t}]$`<eot>`$\ldots$`<bot>`$[s_n^{0:m\Delta t}]$`<eot><bot>`$[s_*^{0:\ell\Delta t}]$.

At inference, we adjust the **temperature parameter** $\tau$ to calibrate the level of diversity of the next-token distributions. The temperature $\tau$ scales the logits $y_i$ before the softmax function :

$$p(S_{[i]} = k | S_{[i' < i]}) = \text{softmax}\left(\frac{y_k}{\tau}\right) = \frac{\exp\left(\frac{y_k}{\tau}\right)}{\sum_j \exp\left(\frac{y_j}{\tau}\right)}$$

When $\tau > 1$, the distribution becomes more uniform, encouraging exploration, whereas $\tau < 1$ sharpens the distribution, favoring more deterministic predictions.

## 4 EXPERIMENTS

In this section, we experimentally validate that our framework enables various types of conditioning during inference. As the first model capable of performing in-context learning with an autoregressive transformer for PDEs, Zebra can tackle a wide range of tasks that existing frameworks are unable to address without gradient-based adaptation or finetuning. We conduct *pretraining* as outlined in Section 3 for each dataset described in Section 4.1 and evaluate Zebra on distinct tasks *without additional finetuning*. We begin by assessing Zebra's performance in the challenging one-shot setting, focusing on adaptation methods as the main baselines (Section 4.2). Next, we compare its performance in the more traditional temporal conditioning tasks in Section 4.3. We then explore its generalization in the out-of-distribution regime in Section 4.2. Lastly, we examine the uncertainty quantification enabled by Zebra's generative nature and analyze the model's generated trajectories in Appendix D.1 and Appendix D.2, respectively.

### Table 2: **Dataset Summary**

| Dataset Name | Number of env. | Trajectories per env. | Main parameters |
|---|---|---|---|
| *Advection* | 1200 | 10 | Advection speed |
| *Heat* | 1200 | 10 | Diffusion and forcing |
| *Burgers* | 1200 | 10 | Diffusion and forcing |
| *Wave boundary* | 4 | 3000 | Boundary conditions |
| *Combined equation* | 1200 | 10 | $\alpha, \beta, \gamma$ |
| *Wave 2D* | 1200 | 10 | Wave celerity and damping |
| *Vorticity 2D* | 1200 | 10 | Diffusion |

Table 3: **One-shot adaptation**. Conditioning from a similar trajectory. Test results in relative L2 on the trajectory. '–' indicates inference has diverged.

| | Advection | Heat | Burgers | Wave b | Combined | Wave 2D | Vorticity 2D |
|---|---|---|---|---|---|---|---|
| CAPE | 0.00941 | 0.223 | 0.213 | 0.978 | **0.00857** | – | – |
| CODA | **0.00687** | 0.546 | 0.767 | 1.020 | 0.0120 | 0.777 | 0.678 |
| [CLS] ViT | 0.140 | **0.136** | 0.116 | 0.971 | 0.0446 | 0.271 | 0.972 |
| MPP-in-context | 0.0902 | 0.472 | 0.582 | 0.472 | 0.0885 | 0.390 | 0.173 |
| Zebra | 0.00794 | 0.154 | **0.115** | **0.245** | 0.00965 | **0.207** | **0.119** |

## 4.1 DATASETS DETAILS

As in Kirchmeyer et al. (2022), we generate data in batches where each batch of trajectories shares the same PDE parameters. For each batch or environment, the resulting trajectories sharing the same dynamics have different initial conditions. We consider different whole factors of variations across multiple datasets and drastically increase the different number of environments compared to previous studies (Yin et al. (2022), Kirchmeyer et al. (2022)). We conduct experiments across seven datasets: five in 1D—*Advection*, *Heat*, *Burgers*, *Wave-b*, *Combined*—and two in 2D—*Wave 2D*, *Vorticity*. These datasets were selected to encompass different physical phenomena and test generalization under changes to various PDE terms, as described below.

**Varying PDE coefficients** The changing factor is the set of coefficients $\mu$ in Equation 1. For the *Burgers*, *Heat*, and *Vorticity 2D* equations, the viscosity coefficient $\nu$ varies across environments. For *Advection*, the advection speed $\beta$ changes. In *Wave-c* and *Wave-2D*, the wave's celerity $c$ is unique to each environment, and the damping coefficient $k$ varies across environments in *Wave-2D*. In the *Combined* equation, three coefficients $(\alpha, \beta, \gamma)$ vary, each influencing different derivative terms respectively: $-\frac{\partial u^2}{\partial x}, +\frac{\partial^2 u}{\partial x^2}, -\frac{\partial^3 u}{\partial x^3}$ on the right-hand side of Equation 1.

**Varying boundary conditions** In this case, the varying parameter is the boundary condition $\mathcal{B}$ from Equation 2. For *Wave-b*, we explore two types of boundary conditions—Dirichlet and Neumann—applied independently to each boundary, resulting in four distinct environments.

**Varying forcing term** The varying parameter is the forcing term $\delta$ in Equation 1. In *Burgers* and *Heat*, the forcing terms vary by the amplitude, frequency, and shift coefficients of $\delta(t, x) = \sum_{j=1}^{5} A_j \sin\left(\omega_j t + 2\pi \frac{l_j x}{L} + \phi_j\right)$.

A detailed description of the datasets is provided in Appendix B, and a summary of the number of environments used during training, the number of trajectories sharing the same dynamics, and the varying PDE parameters across environments is presented in Table 2. For testing, we evaluate all methods on trajectories with new initial conditions on unseen environments. Specifically, we used 120 new environments for the 2D datasets and 12 for the 1D datasets, with each environment containing 10 trajectories.

## 4.2 CONTEXT ADAPTATION FROM SIMILAR TRAJECTORIES

**Setting** We evaluate Zebra's ability to perform in-context learning by *leveraging example trajectories that follow the same underlying dynamics as the target*. Formally, in the $n$-shot adaptation setting, we assume access to a set of $n$ context trajectories $\{\boldsymbol{u}_1^{0:m\Delta t}, \dots, \boldsymbol{u}_n^{0:m\Delta t}\}$ at inference time, all of which belong to the same dynamical system $\mathcal{F}_\xi$. The goal of the adaptation task is to accurately predict a future trajectory $\boldsymbol{u}_*^{\Delta t:m\Delta t}$ from a new initial condition $\boldsymbol{u}_*^0$, knowing that the target dynamics is shared with the provided context example trajectories. In this comparison, Zebra is the only model that performs in-context learning from these example trajectories.

**Sampling** For Zebra, we employ a random sampling procedure for generating the next tokens for all datasets, setting a low temperature ($\tau = 0.1$) to prioritize accuracy over diversity. Predictions are generated using a single sample under this configuration.

**Baselines** We evaluate Zebra against CODA (Kirchmeyer et al., 2022) and CAPE (Takamoto et al., 2023). CODA is a meta-learning framework designed for learning parametric PDEs. It leverages common knowledge from multiple environments where trajectories from a same environment $e$ share the same PDE parameter values. CODA training performs adaptation in the parameter space by learning shared parameters across all environments and a context vector $c^e$ specific to each environment. At the inference stage, CODA adapts to a new environment in a one-shot manner by only tuning $c^e$ with several gradient steps. CAPE is not designed to perform adaptation via extra-trajectories, but instead needs the correct parameter values as input to condition a neural solver. We adapt it to our setting, by learning a context $c^e$ instead of using the real parameter values. During adaptation, we only tune this context $c^e$ via gradient updates. Additionally, we introduce a baseline based on a vision transformer (Peebles & Xie, 2023), integrating a [CLS] token that serves as a learned parameter for each environment. This token lets the model handle different dynamics, and during inference, we adapt the [CLS] vector via gradient updates, following the same approach used in CODA and CAPE. We refer to this baseline as [CLS] ViT. As an additional baseline, we include MPP-in-context, which has been adapted from McCabe et al. (2023) by stacking similar trajectories in the temporal dimension (as we do) to enable in-context conditioning and one-shot adaptation.

**Metrics** We evaluate the performance using the Relative $L^2$ norm between the predicted rollout trajectory $\hat{u}_*^{\text{trajectory}}$ and the ground truth $\boldsymbol{u}_*^{\text{trajectory}}$: $L_{\text{test}}^2 = \frac{1}{N_{\text{test}}} \sum_{j \in \text{test}} \frac{||\hat{u}_j^{\text{trajectory}} - u_j^{\text{trajectory}}||_2}{||u_j^{\text{trajectory}}||_2}$.

**Results** As evidenced in Table 3, Zebra demonstrates strong overall performance in the one-shot adaptation setting, often surpassing baseline methods that have been trained specifically for this task. In more challenging datasets, such as *Burgers*, *Wave-b*, and the 2D cases, Zebra consistently achieves lower relative L2 errors, highlighting its capacity to model complex dynamics effectively. Notably, Zebra excels in 2D environments, outperforming both CODA and [CLS] ViT and avoiding the divergence issues encountered by CAPE. While Zebra performs comparably to CODA on simpler datasets like *Advection* and *Combined*, its overall stability and versatility across a range of scenarios, particularly in 2D settings, highlight its competitiveness. Although there is room for improvement in specific cases, such as the *Heat* dataset, Zebra stands out as a reliable and scalable solution for in-context adaptation for parametric PDEs, offering a robust alternative to existing gradient-based methods. We further analyze the influence of the number $n$ of context examples on the rollout performance with Zebra, as illustrated in Figure 2. While there is a general decreasing trend—indicating that more context examples tend to reduce rollout loss—there is still noticeable variance in the results. This suggests that the relationship between the number of context examples and performance is not perfectly linear. We hypothesize that this analysis would benefit from being conducted with more than a single generated trajectory to ensure more robust estimations.

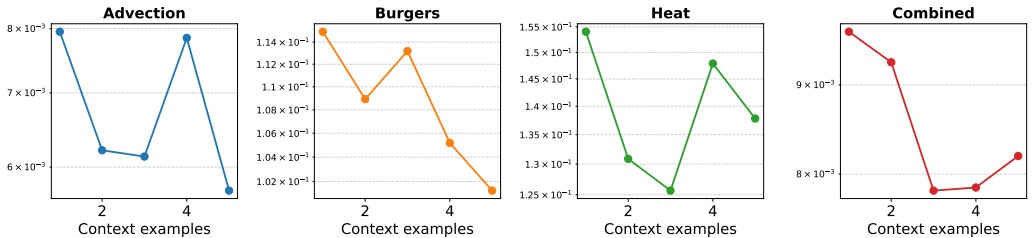

Figure 2: **Influence of the number of examples**. Zebra's rollout loss for a different number of trajectory examples. The x-axis is the # of context examples and the y-axis is the Relative $L^2$.

### 4.3 TEMPORAL CONDITIONING

**Setting** We then evaluate the temporal conditioning capabilities of Zebra, i.e. its generalization capabilities when *conditioned by the initial states of the target trajectories*. Formally, given a new trajectory $\boldsymbol{u}_*$, we suppose that a set of $\ell$ states $\boldsymbol{u}_*^{0:(\ell-1)\Delta t}$ is already available from the trajectory and we wish to predict the states at the following timestamps $\boldsymbol{u}_*^{\ell\Delta t:m\Delta t}$.

**Baselines**   We test `Zebra` against CODA (Kirchmeyer et al., 2022), CAPE (Takamoto et al., 2023) and MPP (McCabe et al., 2023). CODA is designed for adapting neural networks to a new environment given an extra-trajectory sampled from this environment. In this setup, we do not have access to extra-trajectories but to the first timestamps of the target trajectory. We thus modify CODA for this setting; the model is adapted with the $\ell$ first states by learning only $c^e$, and then starts from $\boldsymbol{u}^{(\ell-1)\Delta t}$ to predict the rest of the trajectory. We adapt CAPE to that setting too, as done with CODA. MPP is a vision transformer conditioned by a sequence of frames, using temporal and spatial self-attention blocks to capture spatio-temporal dependencies. MPP does not require additional adaptation at inference and can be used in zero-shot on new trajectories. It is pretrained with a fixed number of input frames in the vanilla version. We also include MPP-in-context, a variant that was pretrained with context trajectories, as explained in Section 4.2. For `Zebra`, we employ the sampling procedure described in Section 4.2.

Table 4: **Zero-shot prediction from 2 frames**. Conditioning from a trajectory history with 2 frames as input. Test results in relative L2 on the trajectory. '–' indicates inference has diverged.

|  | *Advection* | *Heat* | *Burgers* | *Wave b* | *Combined* | *Wave 2D* | *Vorticity 2D* |
|---|---|---|---|---|---|---|---|
| CAPE | 0.00682 | 0.234 | 0.225 | 1.10 | 0.0125 | – | – |
| CODA | **0.00560** | 0.378 | 0.472 | 0.994 | 0.0197 | 0.974 | 0.623 |
| MPP[2] | 0.0075 | **0.0814** | **0.100** | 1.0393 | 0.0250 | 0.285 | 0.101 |
| MPP[3] | 0.919 | 1.0393 | 0.581 | **0.900** | 0.201 | 0.596 | 0.219 |
| MPP-in-context | 0.197 | 0.204 | 0.176 | 1.13 | 0.0985 | 0.363 | 0.1393 |
| `Zebra` | 0.00631 | 0.227 | 0.221 | 0.992 | **0.0084** | **0.201** | **0.0874** |

**Results**   Table 4 highlights `Zebra`'s strong zero-shot prediction performance using only 2 frames ($\ell = 2$) as context, outperforming competing methods across a wide range of PDEs. Notably, `Zebra` excels on both 1D and 2D datasets, delivering consistent and robust results even in complex dynamics like *Wave 2D* and *Vorticity 2D*. CAPE and CODA, while competitive in some datasets, either diverge or struggle with accuracy in more challenging scenarios, particularly in 2D problems.

MPP trained with two frames (MPP[2]) is overall a very strong baseline in this setting; it performs best on *Heat* and *Burgers* and obtains good results in the 2D cases. However, if we take a model that has been pretrained specifically on three frames (MPP[3]), and test it under this setting, the performance degrades drastically. In contrast, Zebra exhibits a high flexibility. It can be used with any number of frames, as long it does not exceed the maximum sequence size seen during training. Furthermore, keeping this setting with two initial frames as inputs, we expose in Figure 3 the gains we could expect on the rollout loss if we had access in addition to the input frames to an example trajectory as described in Section 4.2.

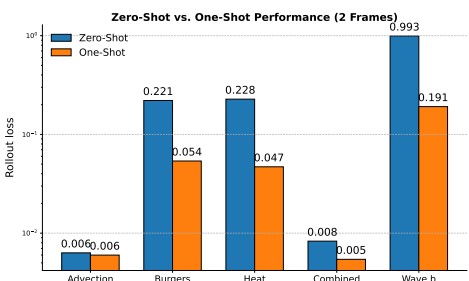

Figure 3: **Zero-shot vs one-shot performance** of `Zebra` with 2 frames.

We can observe that Zebra consistently improves its accuracy when prompted with an additional example. Most notably, `Zebra`'s behavior goes from a random prediction on *Wave b* in zero-shot to more confident predictions thanks to the additional example.

## 4.4   OUT-OF-DISTRIBUTION GENERALIZATION

**Datasets**   We focus on the following distribution shifts from these datasets (i) *Heat*: we vary the forcing coefficients from Appendix B.3 and sample $A_j \in [-1.0, 1.0]$, $\omega_j \in [-0.8, -0.8]$; (ii) *Vorticity 2D*: We sample the viscosity within the range $[5 \times 10^{-4}, 10^{-3}]$ for numerical comparisons. In Figure 4 we also evaluate a shift to a more turbulent regime by sampling the viscosity in $[10^{-5}, 10^{-4}]$ ; (iii) *Wave 2D*: we sample the wave celerity $c$ in $[500, 550]$, and the damping term $k$ in $[50, 60]$. **Setting**   We evaluate the different models on the one-shot and zero-shot settings for trajectories with *out-of-distribution* parameters. Note that this setting is particularly challenging.

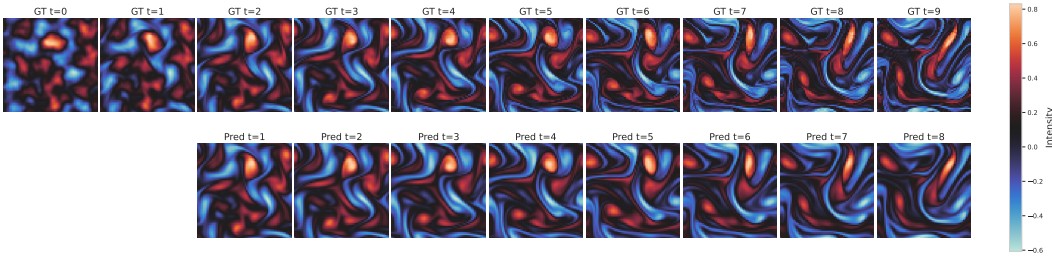

Figure 4: **Zero-shot** prediction on *Vorticity* in the turbulent OoD regime $\nu \in [1e-5, 1e-4]$

**Results** We report all metrics in Table 5 for both zero-shot and one-shot experiments. Overall, all methods are impacted by the shift in distribution, with performance consistently degrading across all tasks. Zebra is best on 5 experiments out of 6. CODA and CAPE perform the worst in these scenarios. This is expected for the 2D datasets, as they already struggled to generalize within the training distribution. On the *Heat* dataset, errors for CAPE and CODA double in the one-shot setting, whereas Zebra maintains similar accuracy, highlighting its robustness to distribution shifts. In the one-shot setting, MPP-in-context outperforms CAPE and CODA baselines. MPP[2] performs well on the zero-shot setting with two context frames, but cannot solve initial value problems when only one initial condition is provided as for classical solvers. Overall, out-of-distribution generalization appears as a complex task for strong distributional shifts. Comparing Zebra to CAPE and CODA, adaptation through in-context learning appears as a better alternative than gradient-based adaptation.

Table 5: **Out-of-distribution results**. Test results in relative L2 on the trajectory. '–' indicates inference has diverged. For each dataset, the left column shows results for One-shot adaptation , while the right column shows results for Zero-shot prediction.

|  | Heat | | Wave 2D | | Vorticity 2D | |
| --- | --- | --- | --- | --- | --- | --- |
|  | One-shot | Zero-shot | One-shot | Zero-shot | One-shot | Zero-shot |
| CAPE | 0.47 | 0.33 | – | – | – | – |
| CODA | 1.03 | 0.66 | 1.51 | 1.32 | 1.71 | 1.59 |
| MPP[2] | – | **0.19** | – | 0.70 | – | 0.22 |
| MPP-in-context | 0.52 | 0.32 | **0.68** | 0.66 | 0.30 | 0.28 |
| Zebra | **0.15** | 0.34 | **0.68** | **0.55** | 0.24 | **0.21** |

## 5 LIMITATIONS

The quality of the generated trajectories is limited by the decoder's ability to reconstruct details from the quantized latent space. While the reconstructions are excellent for many applications, we believe there is room for improvement. Future work could explore scaling the codebook size, as suggested by Yu et al. (2023a) and Mentzer et al. (2023), to enhance the model's reconstruction capabilities. Additionally, investigating approaches that avoid vector quantization (Li et al., 2024) could offer even further improvements, provided that in-context learning capabilities are preserved. Lastly, our encoder and decoder are built using convolutional blocks, which restricts their use to regular domains. More flexible architectures, such as those proposed by Serrano et al. (2024), could help extend the model to more complex and irregularly sampled systems.

## 6 CONCLUSION

This study introduces Zebra, a novel generative model that adapts language model pretraining techniques for solving parametric PDEs. We propose a pretraining strategy that enables Zebra to develop in-context learning capabilities. Our experiments demonstrate that the pretrained model performs competitively against specialized baselines across various scenarios. Additionally, as a generative model, Zebra facilitates uncertainty quantification and can generate new trajectories, providing valuable flexibility in applications.

## 7 REPRODUCIBILITY STATEMENT

We describe the pretraining strategy in Section 3, and provide details on the architecture and its hyperparameters in Appendix C. The datasets used are described in Appendix B. We plan to release the code, the weights of the models, and the datasets used in this study upon acceptance.

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

# A    RELATED WORK

## A.1    LEARNING PARAMETRIC PDEs

**The classical ML paradigm**    The classical ML paradigm for solving parametric PDEs consists in sampling from the PDE parameter distribution trajectories to generalize to new PDE parameter values. It is the classical ERM approach. The natural way for generalizing to new PDE parameters is to explicitly embed them in the neural network (Brandstetter et al., 2022b). Takamoto et al. (2023) proposed a channel-attention mechanism to guide neural solvers with the physical coefficients given as input; it requires complete knowledge of the physical system and are not designed for other PDE parameter values, e.g., boundary conditions. It is commonly assumed that prior knowledge are not available, but instead rely on past states of trajectories for inferring the dynamics. Neural solvers and operators learn parametric PDEs by stacking the past states as channel information as done in Li et al. (2021), or by creating additional temporal dimension as done in video prediction contexts (Ho et al., 2022; McCabe et al., 2023). Their performance drops when shifts occur in the data distribution, which is often met with parametric PDEs, as small changes in the PDE parameters can lead to various dynamics. To better generalize to new PDE parameter values, Subramanian et al. (2023) instead leverages fine-tuning from pretrained models to generalize to new PDE parameters. It however often necessitates a relatively large number of fine tuning samples to effectively adapt to new PDE parameter values, by updating all or a subset of parameters (Herde et al., 2024; Hao et al., 2024).

**Adaptive conditioning**    To better adapt to new PDE parameters values at inference, several works have explored learning on multiple environments. During training, a limited number of environments are available, each corresponding to a specific PDE instance. Yin et al. (2022) introduced LEADS, a multi-task framework for learning parametric PDEs, where a shared model from all environments and a model specific to each environment are learned jointly. At inference, for a new PDE instance, the shared model remain frozen and only a model specific to that environment is learned. Kirchmeyer et al. (2022) proposed to perform adaptive conditioning in the parameter space; the framework adapts the weights of a model to each environment via a hyper-network conditioned by a context vector $c^e$ specific to each environment. At inference, the model adapts to a new environment by only tuning $c^e$. Park et al. (2023) bridged the gap from the classical gradient-based meta-learning approaches by addressing the limitations of second-order optimization of MAML and its variants (Finn et al., 2017; Zintgraf et al., 2019b). Other works have also extended these frameworks to quantify uncertainty of the predictions : Jiaqi et al. (2024) proposed a conditional neural process to capture uncertainty in the context of multiple environments with sparse trajectories, while Nzoyem et al. (2024) leveraged information from multiple environments to enable more robust predictions and uncertainty quantification.

## A.2    GENERATIVE MODELS

**Auto-regressive Transformers for Images and Videos**    Recent works have explored combining language modeling techniques with image and video generation, typically using a VQ-VAE (Oord et al., 2017) paired with a causal transformer (Esser et al., 2021) or a bidirectional transformer (Chang et al., 2022). VQGAN (Esser et al., 2021) has become the leading framework by incorporating perceptual and adversarial losses to improve the visual realism of decoder outputs from quantized latent representations. However, while these methods succeed in generating visually plausible images, they introduce a bias—driven by perceptual and adversarial losses—that leads the network to prioritize perceptual similarity and realism, often causing reconstructions to deviate from the true input. In contrast, `Zebra` focuses on maximizing reconstruction accuracy, and did not observe benefits from using adversarial or perceptual losses during training.

In video generation, models like Magvit (Yu et al., 2023a) and Magvit2 (Yu et al., 2023b) adopt similar strategies, using 3D CNN encoders to compress sequences of video frames into spatiotemporal latent representations by exploiting the structural similarities between successive frames in a video. However, such temporal compression is unsuitable for modeling partial differential equations (PDEs), where temporal dynamics can vary significantly between frames depending on the temporal resolution. With `Zebra`, we spatially compress observations using the encoder and learn the temporal dynamics with an auto-regressive transformer, avoiding temporal compression.

## B  DATASET DETAILS

### B.1  ADVECTION

We consider a 1D advection equation with advection speed parameter $\beta$:

$$\partial_t u + \beta \partial_x u = 0$$

For each environment, we sample $\beta$ with a uniform distribution in $[0, 4]$. We sample 1200 parameters, and 10 trajectories per parameter, constituting a training set of 12000 trajectories. At test time, we draw 12 new parameters and evaluate the performance on 10 trajectories each.

We fix the size of the domain $L = 128$ and draw initial conditions as described in Equation (5) in appendix B.5 and generate solutions with the method of lines and the pseudo-spectral solver described in Brandstetter et al. (2022b). We take 140 snapshots along a 100s long simulations, which we downsample to 14 timestamps for training. We used a spatial resolution of 256.

### B.2  BURGERS

We consider the Burgers equation as a special case of the combined equation described in Appendix B.5 and initially in Brandstetter et al. (2022b), with fixed $\gamma = 0$ and $\alpha = 0.5$. However, in this setting, we include a forcing term $\delta(t, x) = \sum_{j=1}^{J} A_j \sin(\omega_j t + 2\pi \ell_j x / L + \phi_j)$ that can vary across different environments. We fix $J = 5$, $L = 16$. We draw initial conditions as described in Equation (5).

For each environment, we sample $\beta$ with a log-uniform distribution in $[1e - 3, 5]$, and sample the forcing term coefficients uniformly: $A_j \in [-0.5, 0.5]$, $\omega_j \in [-0.4, -0.4]$, $\ell_j \in \{1, 2, 3\}$, $\phi_j \in [0, 2\pi]$. We create a dataset of 1200 environments with 10 trajectories for training, and 12 environments with 10 trajectories for testing.

We use the solver from Brandstetter et al. (2022b), and take 250 snapshots along the 4s of the generations. We employ a spatial resolution of 256 and downsample the temporal resolution to 25 frames.

### B.3  HEAT

We consider the heat equation as a special case of the combined equation described in Appendix B.5 and initially in Brandstetter et al. (2022b), with fixed $\gamma = 0$ and $\alpha = 0$. However, in this setting, we include a forcing term $\delta(t, x) = \sum_{j=1}^{J} A_j \sin(\omega_j t + 2\pi \ell_j x / L + \phi_j)$ that can vary across different environments. We fix $J = 5$, $L = 16$. We draw initial conditions as described in Equation (5).

For each environment, we sample $\beta$ with a log-uniform distribution in $[1e - 3, 5]$, and sample the forcing term coefficients uniformly: $A_j \in [-0.5, 0.5]$, $\omega_j \in [-0.4, -0.4]$, $\ell_j \in \{1, 2, 3\}$, $\phi_j \in [0, 2\pi]$. We create a dataset of 1200 environments with 10 trajectories for training, and 12 environments with 10 trajectories for testing.

We use the solver from Brandstetter et al. (2022b), and take 250 snapshots along the 4s of the generations. We employ a spatial resolution of 256 and downsample the temporal resolution to 25 frames.

### B.4  WAVE BOUNDARY

We consider a 1D wave equation as in Brandstetter et al. (2022b).

$$\partial_{tt} u - c^2 \partial_{xx} u = 0, \quad x \in [-8, 8]$$

where $c$ is the wave velocity ($c = 2$ in our experiments). We consider Dirichlet $\mathcal{B}[u] = u = 0$ and Neumann $\mathcal{B}[u] = \partial_x u = 0$ boundary conditions.

We consider 4 different environments as each boundary can either respect Neumann or Dirichlet conditions, and sample 3000 trajectories for each environment. This results in 12000 trajectories for

training. For the test set, we sample 30 new trajectories from these 4 environments resulting in 120 test trajectories.

The initial condition is a Gaussian pulse with a peak at a random location. Numerical ground truth is generated with the solver proposed in Brandstetter et al. (2022b). We obtain ground truth trajectories with resolution $(n_x, n_t) = (256, 250)$, and downsample the temporal resolution to obtain trajectories of shape $(256, 60)$.

### B.5 COMBINED EQUATION

We used the setting introduced in Brandstetter et al. (2022b), but with the exception that we do not include a forcing term. The combined equation is thus described by the following PDE:

$$[\partial_t u + \partial_x (\alpha u^2 - \beta \partial_x u + \gamma \partial_{xx} u)](t, x) = \delta(t, x), \tag{4}$$

$$\delta(t, x) = 0, \quad u_0(x) = \sum_{j=1}^{J} A_j \sin(2\pi \ell_j x / L + \phi_j). \tag{5}$$

For training, we sampled 1200 triplets of parameters uniformly within the ranges $\alpha \in [0, 1]$, $\beta \in [0, 0.4]$, and $\gamma \in [0, 1]$. For each parameter instance, we sample 10 trajectories, resulting in 12000 trajectories for training and 120 trajectories for testing. We used the solver proposed in Brandstetter et al. (2022a) to generate the solutions. The trajectories were generated with a spatial resolution of 256 for 10 seconds, along which 140 snapshots are taken. We downsample the temporal resolution to obtain trajectories with shape $(256, 14)$.

### B.6 VORTICITY

We propose a 2D turbulence equation. We focus on analyzing the dynamics of the vorticity variable. The vorticity, denoted by $\omega$, is a vector field that characterizes the local rotation of fluid elements, defined as $\omega = \nabla \times \mathbf{u}$. The vorticity equation is expressed as:

$$\frac{\partial \omega}{\partial t} + (\mathbf{u} \cdot \nabla)\omega - \nu \nabla^2 \omega = 0 \tag{6}$$

Here, $\mathbf{u}$ represents the fluid velocity field, $\nu$ is the kinematic viscosity with $\nu = 1/Re$. For the vorticity equation, the parametric problem consists in learning dynamical systems with changes in the viscosity term.

For training, we sampled 1200 PDE parameter values in the range $\nu = [1e - 3, 1e - 2]$. For test, we evaluate our model on 120 new parameters not seen during training in the same paramter range. For each parameter instance, we sample 10 trajectory, resulting in 12000 trajectories for training and 1200 for test.

**Data generation** For the data generation, we use a 5 point stencil for the classical central difference scheme of the Laplacian operator. For the Jacobian, we use a second order accurate scheme proposed by Arakawa that preserves the energy, enstrophy and skew symmetry (Arakawa, 1966). Finally for solving the Poisson equation, we use a Fast Fourier Transform based solver. We discretize a periodic domain into $512 \times 512$ points for the DNS and uses a RK4 solver with $\Delta t = 1e - 3$ on a temporal horizon $[0, 2]$. We then perform a temporal and spatial down-sample operation, thus obtaining trajectories composed of 10 states on a $64 \times 64$ grid.

We consider the following initial conditions:

$$E(k) = \frac{4}{3}\sqrt{\pi}\left(\frac{k}{k_0}\right)^4 \frac{1}{k_0} \exp\left(-\left(\frac{k}{k_0}\right)^2\right) \tag{7}$$

Vorticity is linked to energy by the following equation :

$$\omega(k) = \sqrt{\frac{E(k)}{\pi k}} \tag{8}$$

### B.7 WAVE 2D

We propose a 2D damped wave equation, defined by

$$\frac{\partial^2 \omega}{\partial t^2} - c^2 \Delta \omega + k \frac{\partial \omega}{\partial t} = 0 \tag{9}$$

where $c$ is the wave speed and $k$ is the damping coefficient. We are only interested in learning $\omega$. To tackle the parametric problem, we sample 1200 parameters in the range $c = [0, 50]$ and $k = [100, 500]$. For validation, we evaluate our model on 120 new parameters not seen during training in the same paramter range. For each parameter instance, we sample 10 trajectory, resulting in 12000 trajectories for training and 1200 for validation.

**Data generation** For the data generation, we consider a compact spatial domain $\Omega$ represented as a $64 \times 64$ grid and discretize the Laplacian operator similarly. $\Delta$ is implemented using a $5 \times 5$ discrete Laplace operator in simulation. For boundary conditions, null neumann boundary conditions are imposed. We set $\Delta t = 6.25e - 6$ and generate trajectories on the temporal horizon $[0, 5e - 3]$. The simulation was integrated using a fourth order runge-kutta schema from an initial condition corresponding to a sum of gaussians:

$$\omega_0(x, y) = C \sum_{i=1}^{p} \exp\left(-\frac{(x - x_i)^2 + (y - y_i)^2}{2\sigma_i^2}\right) \tag{10}$$

where we choose $p = 5$ gaussians with $\sigma_i \sim \mathcal{U}_{[0.025, 0.1]}$, $x_i \sim \mathcal{U}_{[0,1]}$, $y_i \sim \mathcal{U}_{[0.,1]}$. We fixed $C$ to 1 here. Thus, all initial conditions correspond to a sum of gaussians of varying amplitudes.

## C ARCHITECTURE DETAILS

### C.1 BASELINE IMPLEMENTATIONS

For all baselines, we followed the recommendations given by the authors. We report here the architectures used for each baseline:

- CODA: For CODA, we implemented a U-Net Ronneberger et al. (2015) and a FNO (Li et al., 2020) as the neural network decoder. For all the different experiments, we reported in the results the best score among the two backbones used. We trained the different models in the same manner as Zebra, i.e. via teacher forcing (Radford et al., 2018). The model is adapted to each environment using a context vector specific to each environment. For the size of the context vector, we followed the authors recommendation and chose a context size equals to the number of degrees of freedom used to define each environment for each dataset. At inference, we adapt to a new environment using 250 gradient steps.

- CAPE: For CAPE (Takamoto et al., 2023), we adapted the method to an adaptation setting. Instead of giving true physical coefficients as input, we learn to auto-decode a context vector $c^e$ as in CODA, which implicitly embeds the specific characteristics of each environment. During inference, we only adapt $c^e$ with 250 gradient steps. For the architectures, we use UNET and FNO as the backbones, and reported the best results among the two architectures for all settings.

- [CLS] ViT: For the ViT, we use a simple vision transformer architecture Dosovitskiy et al. (2021), but adapt it to a meta-learning setting where the CLS token encodes the specific variations of each environment. At inference, the CLS token is adapted to a new environment with 100 gradient steps.

- MPP: For MPP, we used the same model as the one used in the paper (McCabe et al., 2023). As MPP was initially designed for 2D data, we also implemented a 1D version of MPP, to evaluate it both on our 1D and 2D datasets. At inference, MPP can be directly evaluated on new trajectories.

### C.2 ZEBRA ADDITIONAL GENERATION DETAILS

We provide illustrations of our inference pipeline in Figure 5 and in Figure 6 both in the case of adaptive conditioning and temporal conditioning. We also include a schematic view of the different

generation possibilities with Zebra in Figure 7, using the sequence design adopted during pretraining

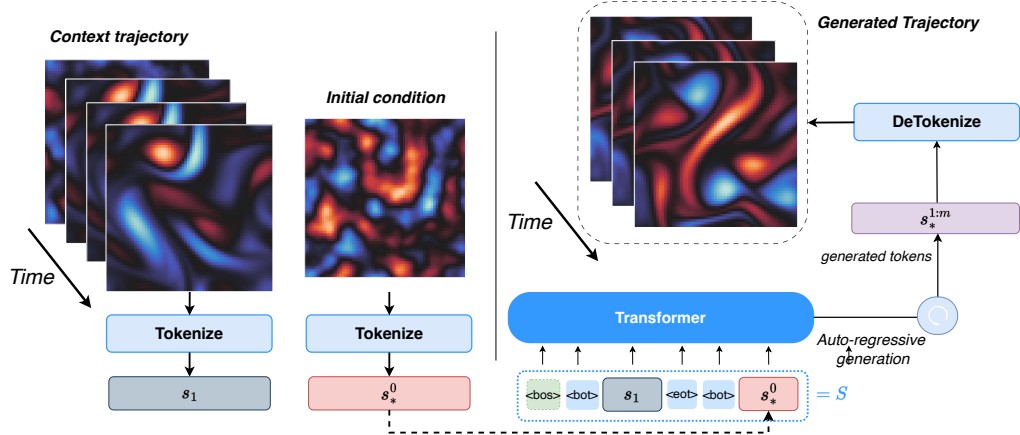

Figure 5: Zebra's inference pipeline from context trajectory. The context trajectory and initial conditions are tokenized into index sequences that are concatenated according to the sequence design adopted during pretraining. The transformer then generates the next tokens to complete the sequence. We detokenize these indices to get back to the physical space.

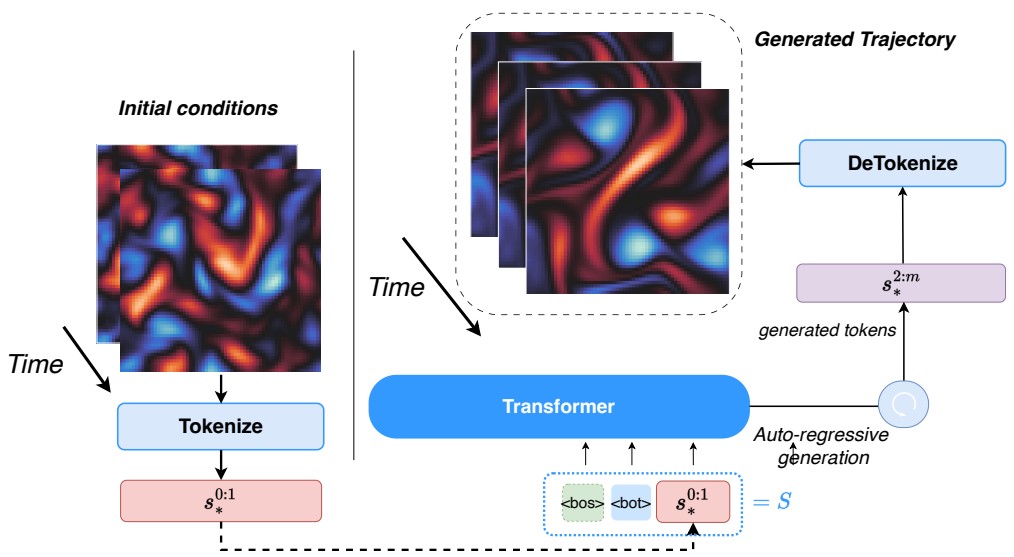

Figure 6: Zebra's inference pipeline from observations of several initial frames. The initial timestamps are tokenized into index sequences that are concatenated according to the sequence design adopted during pretraining. The transformer then generates the next tokens to complete the sequence. We detokenize these indices to get back to the physical space.

**a) Conditional generation from similar trajectories**

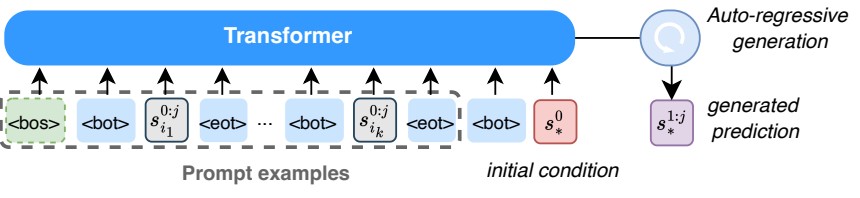

**b) Conditional generation from past trajectory**

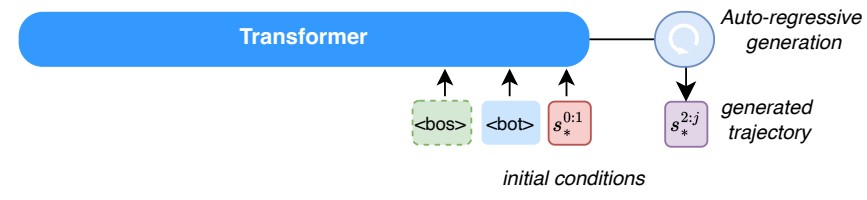

**c) Unconditional generation**

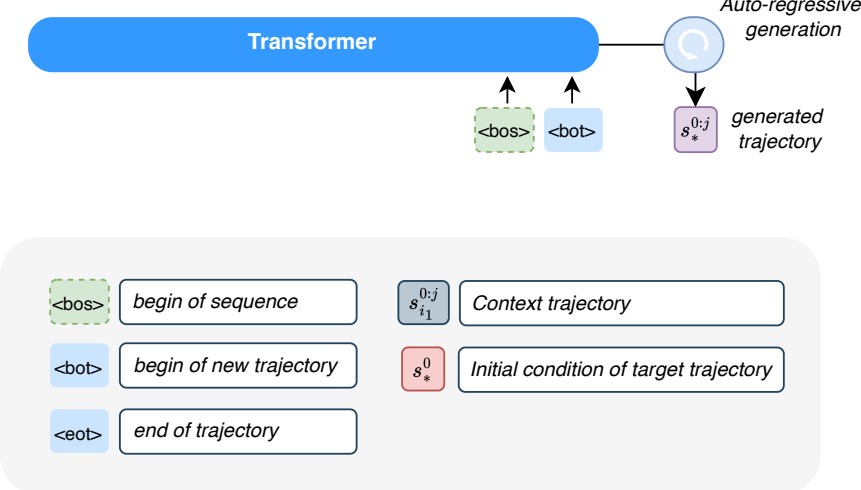

Figure 7: Generation possibilities with Zebra.

## C.3 AUTO-REGRESSIVE TRANSFORMER

Zebra's transformer is based on Llama's architecture, which we describe informally in Figure 8. We use the implementation provided by HuggingFace (Wolf, 2019) and the hyperparameters from Table 6 in our experiments. For training the transformer, we used a single NVIDIA TITAN RTX for the 1D experiments and used a single A100 for training the model on the 2D datasets. Training the transformer on 2D datasets took 20h on a single A100 and it took 15h on a single RTX for the 1d dataset.

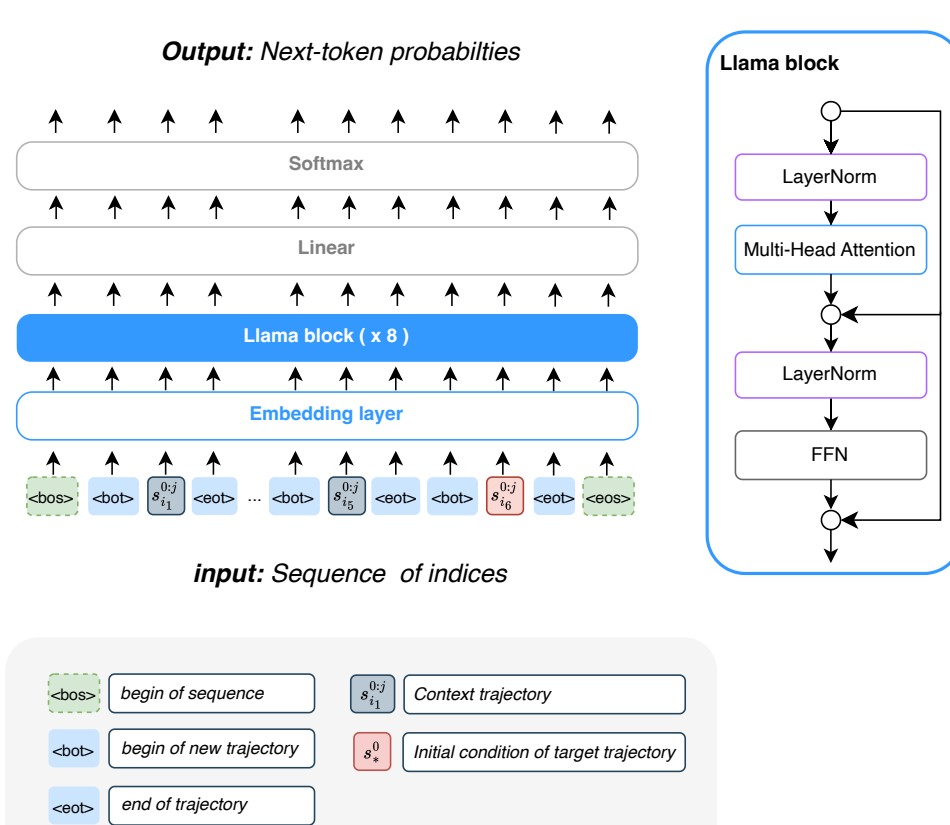

Figure 8: Zebra's transformer architecture is based on Llama (Touvron et al., 2023).

Table 6: Hyperparameters for Zebra's Transformer

| Hyperparameters | Advection | Heat | Burgers | Wave b | Combined | Vorticity 2D | Wave 2D |
|---|---|---|---|---|---|---|---|
| max_context_size | 2048 | 2048 | 2048 | 2048 | 2048 | 8192 | 8192 |
| batch_size | 4 | 4 | 4 | 4 | 4 | 2 | 2 |
| num_gradient_accumulations | 1 | 1 | 1 | 1 | 1 | 4 | 4 |
| hidden_size | 256 | 256 | 256 | 256 | 256 | 384 | 512 |
| mlp_ratio | 4.0 | 4.0 | 4.0 | 4.0 | 4.0 | 4.0 | 4.0 |
| depth | 8 | 8 | 8 | 8 | 8 | 8 | 8 |
| num_heads | 8 | 8 | 8 | 8 | 8 | 8 | 8 |
| vocabulary_size | 264 | 264 | 264 | 264 | 264 | 2056 | 2056 |
| start learning_rate | 1e-4 | 1e-4 | 1e-4 | 1e-4 | 1e-4 | 1e-4 | 1e-4 |
| weight_decay | 1e-4 | 1e-4 | 1e-4 | 1e-4 | 1e-4 | 1e-4 | 1e-4 |
| scheduler | Cosine | Cosine | Cosine | Cosine | Cosine | Cosine | Cosine |
| num_epochs | 100 | 100 | 100 | 100 | 100 | 30 | 30 |

## C.4 VQVAE

We provide a schematic view of the VQVAE framework in Figure 9 and detail the architectures used for the encoder and decoder on the 1D and 2D datasets respectively in Figure 10 and Figure 11. As detailed, we use residual blocks to process latent representations, and downsampling and up-sampling block for decreasing / increasing the spatial resolutions. We provide the full details of the hyperparameters used during the experiments in Table 7. For training the VQVAE, we used a single NVIDIA TITAN RTX for the 1D experiments and used a single V100 for training the model on the

2D datasets. Training the encoder-decoder on 2D datasets took 20h on a single V100 and it took 4h on a single RTX for 1D dataset.

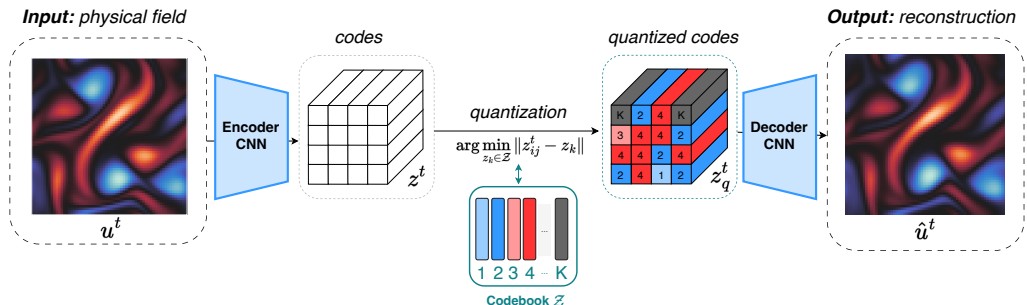

Figure 9: `Zebra`'s VQVAE is used to obtain compressed and discretized latent representation. By retrieving the codebok index for each discrete representation, we can obtain discrete tokens encoding physical observations that can be mapped back to the physical space with high fidelity.

Table 7: Hyperparameters for Zebra's VQVAE

| Hyperparameters | Advection | Heat | Burgers | Wave b | Combined | Vorticity 2D | Wave 2D |
|---|---|---|---|---|---|---|---|
| start_hidden_size | 64 | 64 | 64 | 64 | 64 | 128 | 128 |
| max_hidden_size | 256 | 256 | 256 | 256 | 256 | 1024 | 1024 |
| num_down_blocks | 4 | 4 | 4 | 4 | 4 | 2 | 3 |
| codebook_size | 256 | 256 | 256 | 256 | 256 | 2048 | 2048 |
| code_dim | 64 | 64 | 64 | 64 | 64 | 16 | 16 |
| num_codebooks | 2 | 2 | 2 | 2 | 2 | 1 | 2 |
| shared_codebook | True | True | True | True | True | True | True |
| tokens_per_frame | 32 | 32 | 32 | 32 | 32 | 256 | 128 |
| start learning_rate | 3e-4 | 3e-4 | 3e-4 | 3e-4 | 3e-4 | 3e-4 | 3e-4 |
| weight_decay | 1e-4 | 1e-4 | 1e-4 | 1e-4 | 1e-4 | 1e-4 | 1e-4 |
| scheduler | Cosine | Cosine | Cosine | Cosine | Cosine | Cosine | Cosine |
| num_epochs | 1000 | 1000 | 1000 | 1000 | 1000 | 300 | 300 |

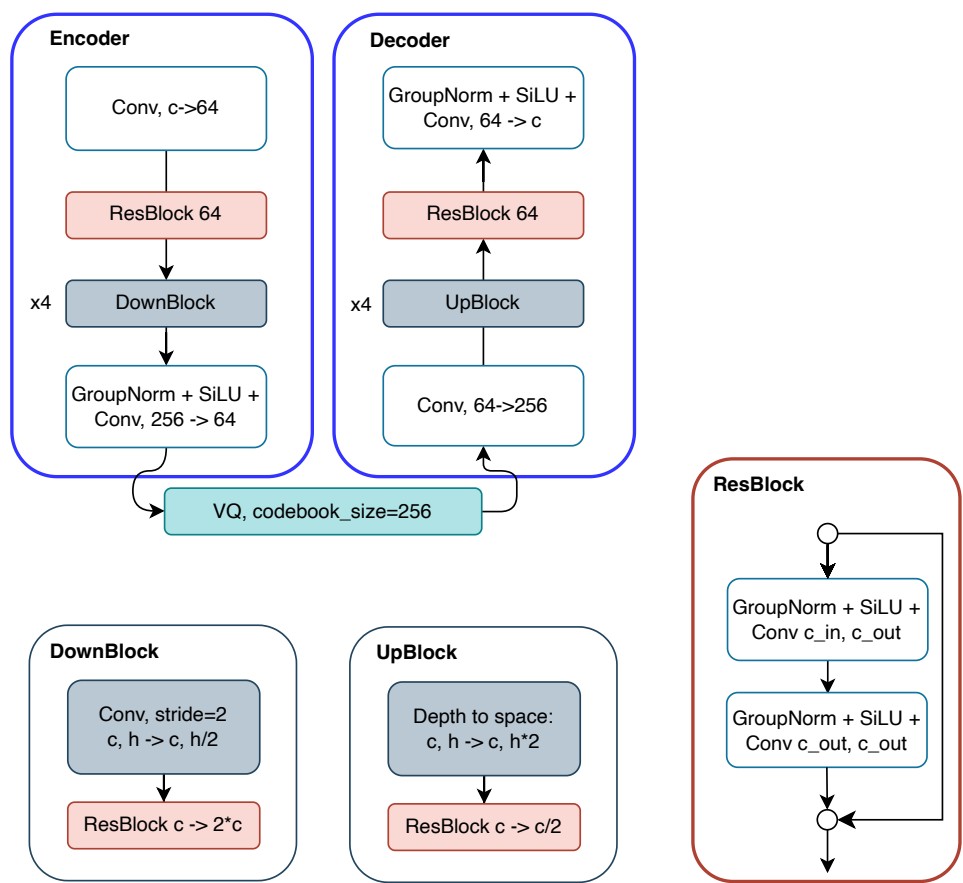

Figure 10: Architecture of Zebra's VQVAE for 1D datasets. Each convolution acts only on the spatial dimension and uses a kernel of size 3. The Residual Blocks are used to process information and increase or decrease the channel dimensions, while the Up and Down blocks respectively up-sample and down-sample the resolution of the inputs. In 1D, we used a spatial compression factor of 16 on all datasets. Every downsampling results in a doubling of the number of channels, and likewise, every upsampling is followed by a reduction of the number of channels by 2. We choose a maximum number of channels of 256.

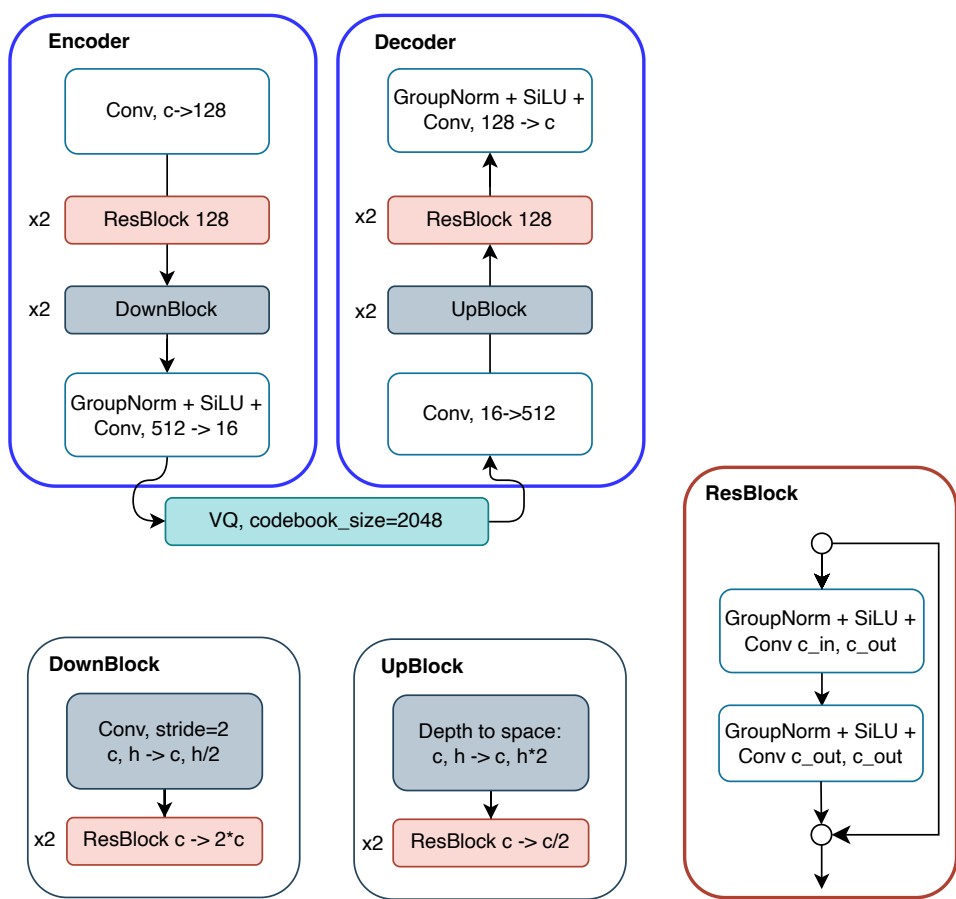

Figure 11: Architecture of Zebra's VQVAE for 2D datasets. Each convolution acts only on the spatial dimensions and uses a kernel of size 3. The Residual Blocks are used to process information and increase or decrease the channel dimensions, while the Up and Down blocks respectively up-sample and down-sample the resolution of the inputs. In 2D, we used a spatial compression factor of 4 for *Vorticity*, and 8 for *Wave2D*. Every downsampling results in a doubling of the number of channels, and likewise, every upsampling is followed by a reduction of the number of channels by 2. We choose a maximum number of channels of 1024.

# D  ADDITIONAL QUANTITATIVE RESULTS

## D.1  UNCERTAINTY QUANTIFICATION

Since `Zebra` is a generative model, it allows us to sample multiple plausible trajectories for the same conditioning input, enabling the computation of key statistics across different generations. By calculating the pointwise mean and standard deviation, we can effectively visualize the model's uncertainty in its predictions. In Figure 12, the red curve represents the ground truth, the blue curve is the predicted mean and the blue shading indicates the empirical confidence interval ($3 \times$ standard deviation).

Motivated by this observation, we investigate how varying the model's temperature parameter $\tau$ affects its predictions; specifically in the one-shot adaptation setting described in Section 4.2. By adjusting $\tau$, we aim to assess its impact on both the accuracy and variability of the predictions. We employ three metrics to evaluate the model's uncertainty:

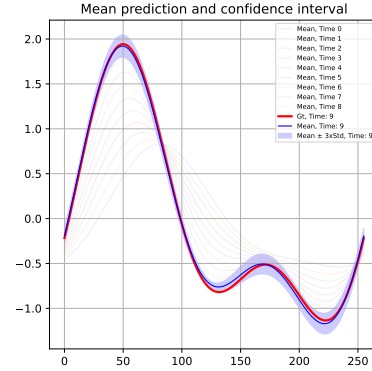

Figure 12: **Uncertainty quantification** with `Zebra` in a one-shot setting on *Heat*.

1. **Relative $L^2$ loss**: This assesses the accuracy of the generated trajectories by measuring the bias of the predictions relative to the ground truth.

2. **Relative standard deviation**: We estimate the variability of the predictions using the formula: Relative Std $= \frac{||\hat{\sigma}_*||_2}{||\hat{m}_*||_2}$ where $\hat{m}_*$ and $\hat{\sigma}_*$ represent the empirical mean and standard deviation of the predictions, computed pointwise across 10 generations.

3. **Confidence level**: We create pointwise empirical confidence intervals $CI(x) = [\hat{m}_*(x) - 3\hat{\sigma}_*(x), \hat{m}_*(x) + 3\hat{\sigma}_*(x)]$ and compute the confidence level as: Confidence level $= \frac{1}{n_x} \sum_x \mathbf{1}_{u_*(x) \in CI(x)}$. This score indicates how often the ground truth falls within the empirical confidence interval generated from sampling multiple trajectories.

When modeling uncertainty, the model achieves a tradeoff between the quality of the mean prediction approximation and the guarantee for this prediction to be in the corresponding confidence interval. Figure 13 illustrates the trade-off between mean prediction accuracy and uncertainty calibration. At lower temperatures, we achieve the most accurate predictions, but with lower variance, i.e. with no guarantee that the target value is within the confidence interval around the predicted mean. Across most datasets, the confidence level then remains low (less than 80% for $\tau < 0.25$), indicating that the true solutions are not reliably captured within the empirical confidence intervals. Conversely, increasing the temperature results in less accurate mean predictions and higher relative standard deviations, but the confidence intervals become more reliable, with levels exceeding 95% for $\tau > 0.5$. Therefore, the temperature can be calibrated depending on whether the focus is on accurate point estimates or reliable uncertainty bounds.

To complement our main analysis, we examine how the model's uncertainty evolves as additional information is provided as input. Specifically, we compare Zebra's average error and relative uncertainty when conditioned on one example trajectory, with one or two frames as initial conditions. Table 8 reports the relative L2 loss and relative standard deviation for both scenarios. The results clearly show that including the first two frames as initial conditions reduces both the error and the relative standard deviation consistently. This indicates that, while some of the uncertainty remains aleatoric, the epistemic uncertainty is reduced as more input information becomes available.

## D.2  ANALYSIS OF THE GENERATION

**Setting**   In this section, we aim to evaluate whether our pretrained model can generate new samples given the observation of a trajectory in a new environment. The key difference with previous settings is that we do not condition the transformer with tokens derived from a real initial condition. We expect the model to generate trajectories, including the initial conditions, that altogether follow the

Table 8: **Uncertainty quantification in the one-shot setting**. Conditioning from a trajectory example and 1 frame or 2 frames as initial conditions. Metrics include relative $L^2$ loss (average accuracy) and relative standard deviation (average spread around the average prediction). The temperature is fixed at 0.1.

|  |  | Advection | Heat | Burgers | Wave b | Combined |
|---|---|---|---|---|---|---|
| Rel. $L^2$ | 1 frame | 0.006 | 0.156 | 0.115 | 0.154 | 0.008 |
| Rel. $L^2$ | 2 frames | 0.004 | 0.047 | 0.052 | 0.075 | 0.005 |
| Rel. Std. | 1 frame | 0.003 | 0.062 | 0.048 | 0.074 | 0.005 |
| Rel. Std. | 2 frames | 0.002 | 0.019 | 0.018 | 0.040 | 0.003 |

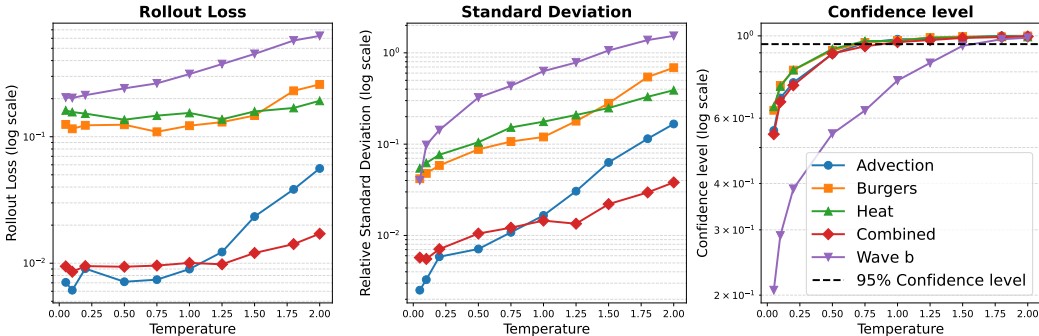

Figure 13: **Uncertainty quantification** with `Zebra`. The main parameter of this study is the temperature (x-axis). We then look from left to right at (1) The rollout loss, i.e. the relative $L^2$ loss between the predictions and the ground truth; (2) The relative standard deviation to quantify the spread around the mean; (3) The confidence level, that measures the frequency of observations that lie within the empirical confidence interval.

same dynamics as in the observations. Our objective is to assess three main aspects: 1) Are the generated trajectories faithful to the context example, i.e., do they follow the same dynamics as those observed in the context ? 2) How diverse are the generated trajectories—are they significantly different from each other? 3) What type of initial conditions does Zebra generate?

**Metrics** To quantify the first aspect, we propose a straightforward methodology. We generate ground truth trajectories using the physical solver that was originally employed to create the dataset, starting with the initial conditions produced by Zebra and using the ground truth parameters of the environment (that `Zebra` cannot access). We then compute the $L^2$ distance between the Zebra-generated trajectories and those generated by the physical solver. For the second aspect, we calculate the average distance between the Zebra-generated trajectories to measure diversity. These two metrics are presented in Table 9 for both the *Advection* and *Combined* Equations. Finally, as a qualitative analysis, we perform PCA on the initial conditions generated by Zebra, and we visualize the first two components in Figure 14 for the Combined Equation case.

**Sampling** We keep the default temperature ($\tau = 1.0$) to put the focus on diversity, and for each context trajectory, we generate 10 new trajectories in parallel.

**Results** According to Table 9, we can conclude that in this context, Zebra can faithfully generate new initial conditions and initial trajectories that respect the same physics as described in the context example. This means that our model has learned the statistical properties that relate the initial conditions with the later timestamps. The high average $L^2$ between generated samples indicate that the generated samples are diverse. We can visually verify this property by looking at fig. 14, noting that the generated samples cover well the distribution of the real samples.

Table 9: **Fidelity and diversity** - The $L^2$ is a proxy score for measuring the fidelity to the dynamics in the context. The average $L^2$ between samples quantifies the distance between each generation.

| Model | $L^2$ | Average $L^2$ between samples |
|---|---|---|
| Advection | 0.0185 | 1.57 |
| Combined Equation | 0.0136 | 1.59 |

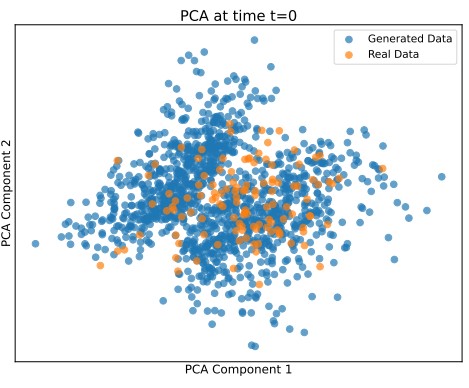 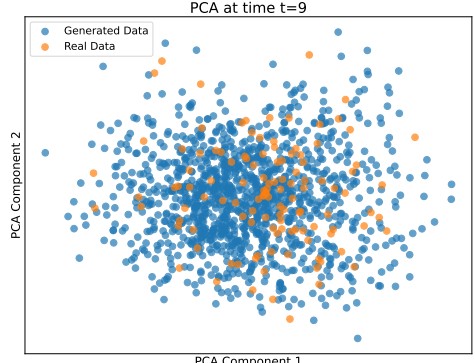

(a) Analysis of the distribution of the generated initial conditions (t=0).

(b) Analysis of the distribution of the generated trajectories (t=9).

Figure 14: **Qualitative analysis**. We generate new initial conditions and obtain rollout trajectories with Zebra on new test environments. We then perform a PCA in the physical space to project on a low-dimensional space, at two given timestamps to check whether the distributions match.

### D.3 DATASET SCALING ANALYSIS

We investigate how the zero-shot error on the test set evolves as we vary the size of the training dataset. To this end, we train the auto-regressive transformer on datasets containing 10, 100, 1000, and 12,000 trajectories and evaluate Zebra's generations on the test set, starting with two frames as inputs. The training time is proportional to the dataset size: for example, the number of training steps for 1,000 trajectories is 10 times the number of steps for 100 trajectories. The results are presented in Figure 15.

First, we observe that Zebra requires a substantial amount of data to generalize effectively to different parameter values, even within the training distribution. This aligns with findings in the literature that transformers, especially auto-regressive transformers, excel at scaling —performing well on very large datasets and for larger model architectures. However, for smaller datasets, this approach may not be the most efficient. We believe that Zebra's potential resides when applied to vast amounts of data, making it an ideal candidate for scenarios involving large-scale training.

Second, for the *Combined equation*, we notice that performance plateaus between 100 and 1,000 trajectories. This may be due to insufficient training or a lack of diverse examples, as the *Combined equation* is more challenging compared to the *Advection equation*, whose performance follows a more log-linear trend. This suggests that additional data or targeted training strategies might be needed to achieve better generalization for more complex equations.

### D.4 INFERENCE TIME COMPARISON

Table 10 compares the inference time for one-shot adaptation across different methods when predicting a single trajectory given a context trajectory and an initial condition. For Zebra, the inference process, which includes encoding, auto-regressive prediction, and decoding, is much faster in 1D and slightly faster in 2D. For Zebra, the bottleneck at inference is the autoregressive generation of tokens, which speed is about 128 tokens per second on a V100 for 2D and an RTX for 1D. The decoding is fast and can be done in parallel for the trajectory in one forward pass. In contrast,

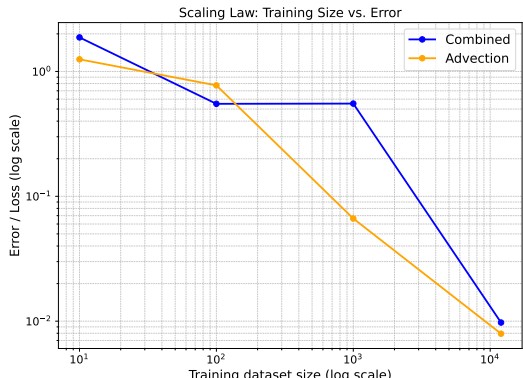

Figure 15: **Dataset scaling analysis**. Zero-shot error on the test set vs. the training dataset size.

for CODA and CAPE, the majority of the inference time is spent on adaptation and gradient-based steps. Here the times were reported with 100 gradient steps, note that we used 250 for the rest of the experiments. We believe Zebra's inference time could be further optimized by (1) improving the optimization code and leveraging specialized hardware such as H100 (for flash attention) and LPUs (which show significant speed-ups agains GPUs), and (2) increasing the number of tokens sampled per step (as in e.g. next-scale prediction Tian et al. (2024)).

Table 10: **Inference times for one-shot adaptation**. Average time in seconds to predict a single trajectory given a context trajectory and an initial condition. Times include adaptation and forecast for CODA and CAPE, while it includes encoding, auto-regressive prediction and decoding for Zebra.

|       | *Advection* | *Vorticity 2D* |
|-------|-------------|----------------|
| CAPE  | 18s         | 23s            |
| CODA  | 31s         | 28s            |
| Zebra | **3s**      | **21s**        |

## D.5 Influence of the codebook size

The codebook size $K$ is a crucial hyperparameter. It directly affects the quality of the reconstructions, since a larger codebook can improve the reconstructions quality. However, it also impacts the dynamics modeling stage: the smaller the codebook, the easier it is for the transformer to learn the statistical correlations between similar trajectories. To have a sense of this trade-off, we report the relative reconstruction errors and the one-shot prediction errors in Table 11. The reconstruction error decreases when the codebook size increases. However, the one-shot prediction error decreases from 32 to 64 codes but then gradually increases from 64 to 512. We can see that it follows a U-curve in Figure 16. This phenomenon was observed in a different context in Cole et al. (2024).

Table 11: **Influence of the codebook size**. Reconstruction error and one-shot prediction error on *Burgers* for different codebook sizes. Errors in relative L2.

| Codebook Size | Reconstruction Loss | One-shot Prediction |
|---------------|---------------------|---------------------|
| 32            | 0.0087              | 0.116               |
| 64            | 0.0043              | 0.097               |
| 128           | 0.0024              | 0.124               |
| 256           | 0.0019              | 0.163               |
| 512           | 0.0015              | 1.093               |

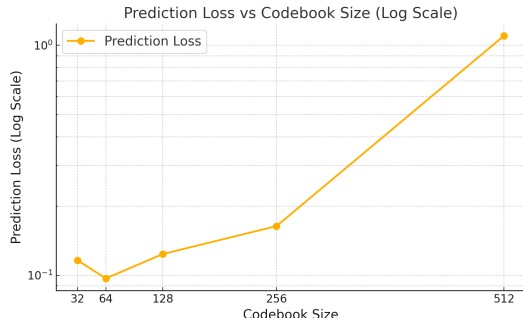

Figure 16: **One-shot accuracy vs codebook size**. One-shot prediction error on the test set for various codebook sizes. Error in relative L2.

## D.6 RECONSTRUCTION ERRORS

We report the accuracy of the reconstructions from our decoder in Table 12. Here, no dynamics is involved, we simply evaluate the quality of the encoding and of the decoding. On 1D and 2D datasets, the decoding errors are respectively of 0.1 % and 1% on the test set.

Table 12: **Reconstruction errors**. Test relative L2 loss between reconstructions from Zebra's VQ-VAE and the ground truths.

|  | *Advection* | *Heat* | *Burgers* | *Wave b* | *Combined* | *Wave 2D* | *Vorticity 2D* |
|---|---|---|---|---|---|---|---|
| VQVAE of `Zebra` | 0.0003 | 0.0019 | 0.0016 | 0.0011 | 0.0022 | 0.010 | 0.017 |

# E QUALITATIVE RESULTS

We provide visualizations of the trajectories generated with `Zebra` under different settings in the following figures:

- **Zero-shot prediction**: Figure 18, Figure 22, Figure 27, Figure 30, Figure 33, Figure 45, Figure 51.

- **One-shot prediction**: Figure 17, Figure 21, Figure 25, Figure 29, Figure 32, Figure 42, Figure 48.

- **Five-shot prediction**: Figure 19, Figure 23, Figure 26, Figure 34.

- **Uncertainty quantification**: Figure 20, Figure 24, Figure 28, Figure 31, Figure 28.

E.1   ADVECTION

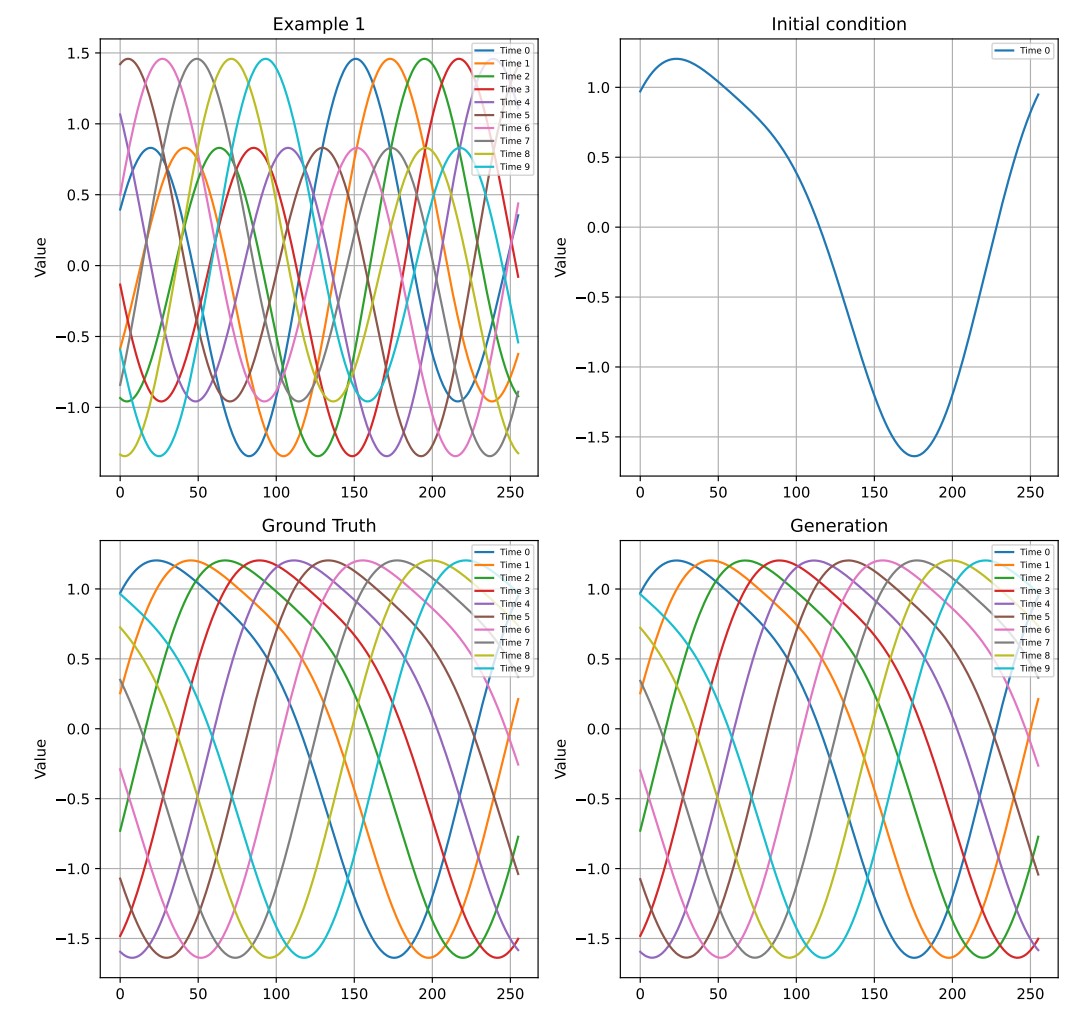

Figure 17: **One-shot** adaptation on Advection

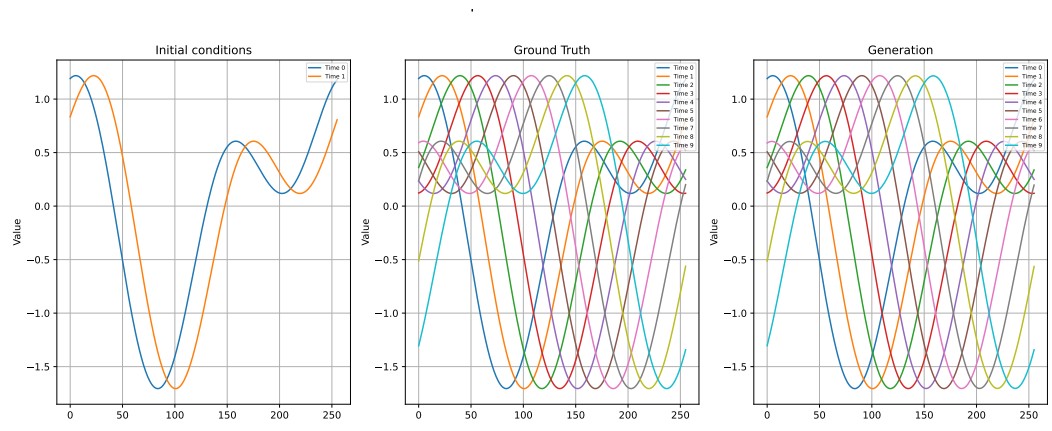

Figure 18: **Zero-shot** prediction on Advection

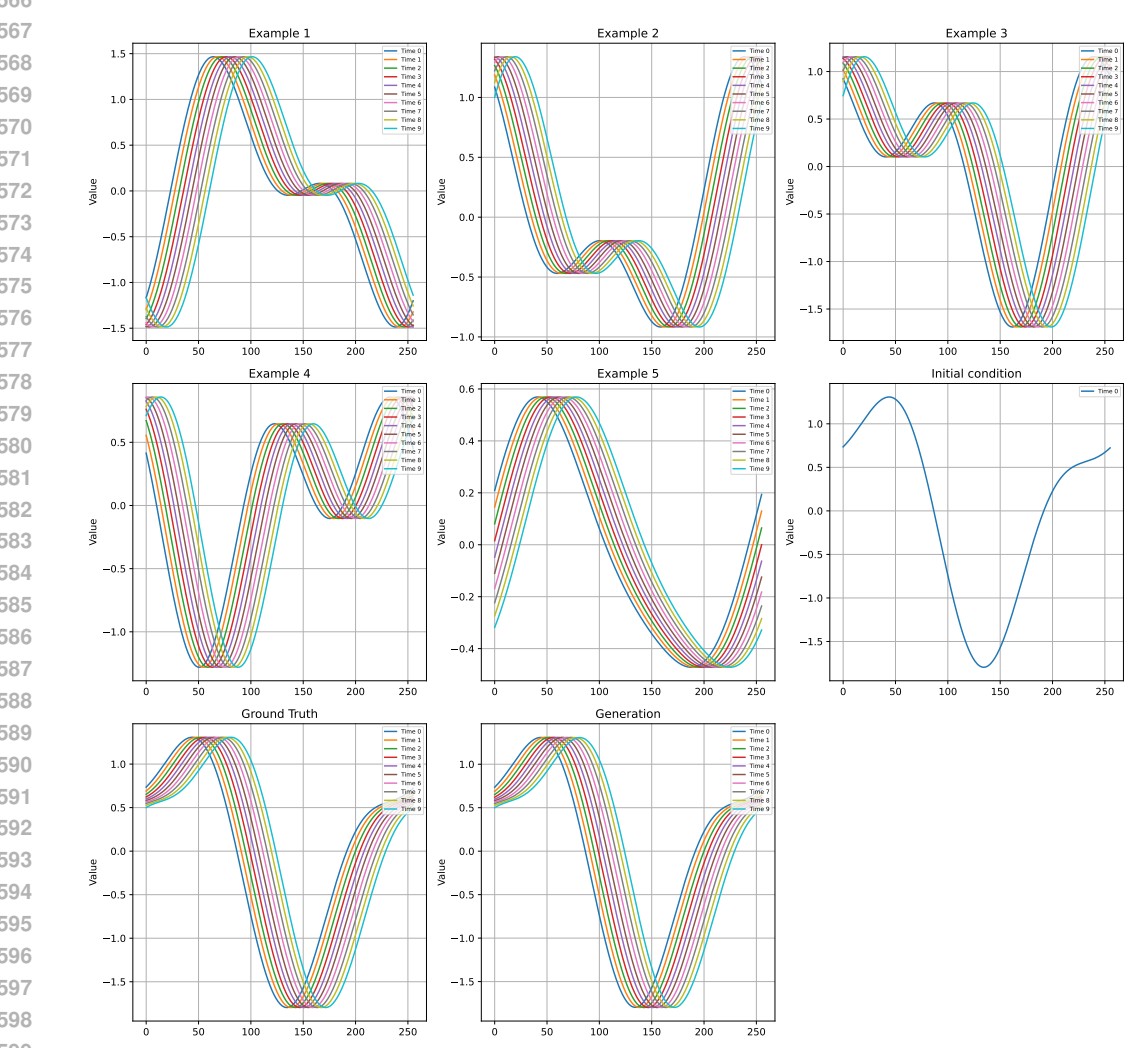

Figure 19: **Five-shot** adaptation on Advection

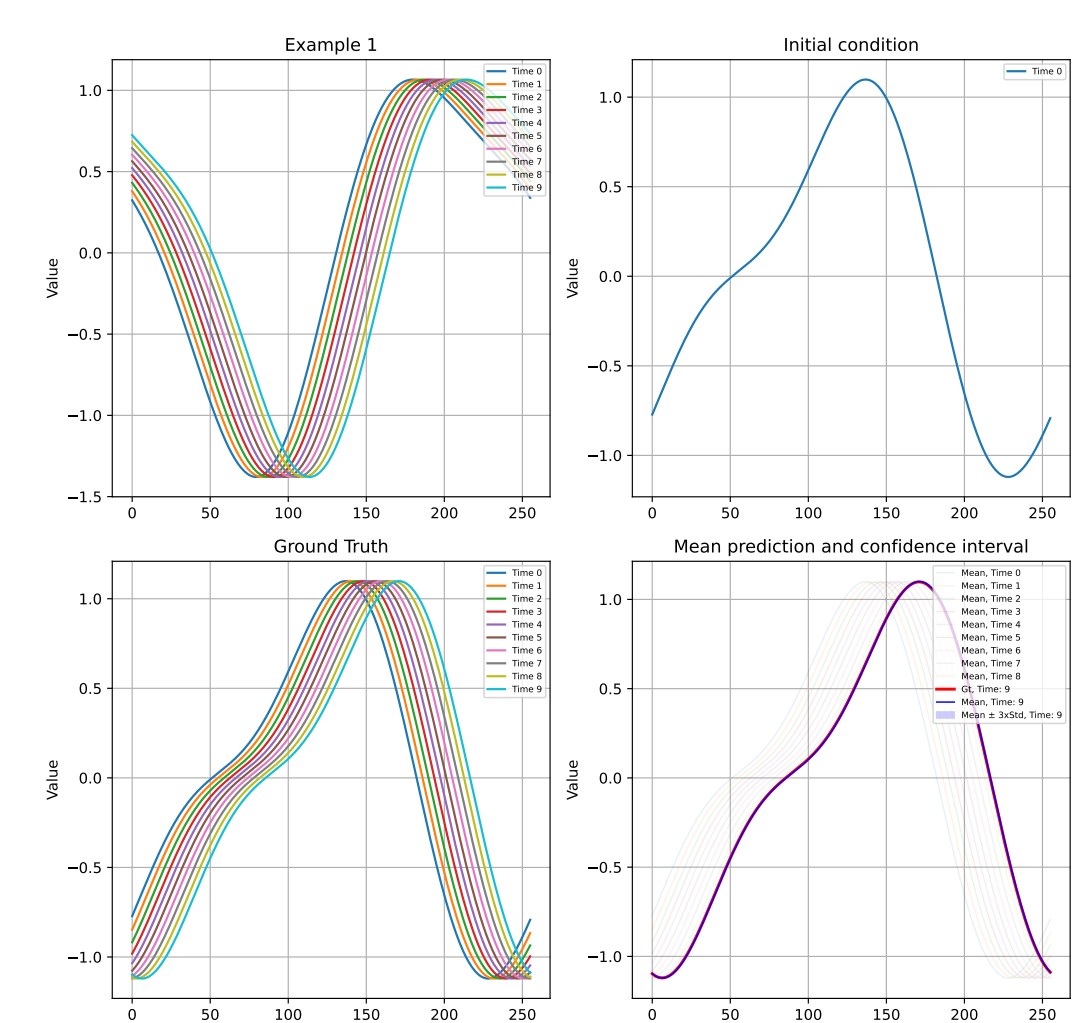

Figure 20: **Uncertainty quantification** on Advection

### E.2 BURGERS

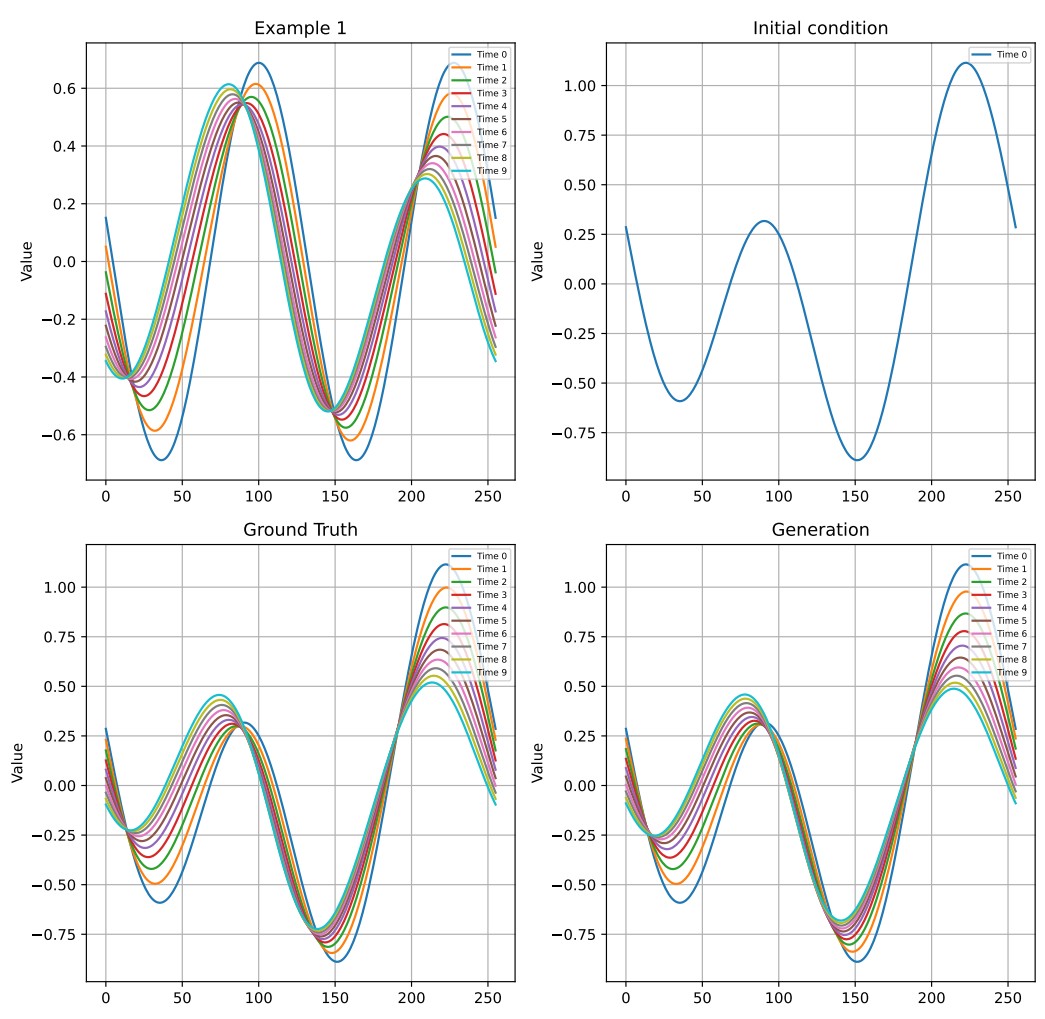

Figure 21: **One-shot** adaptation on Burgers

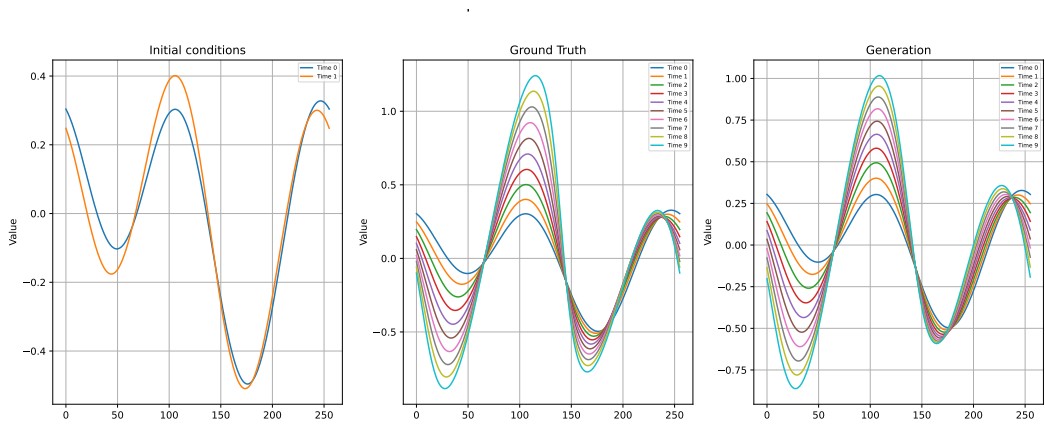

Figure 22: **Zero-shot** prediction on Burgers

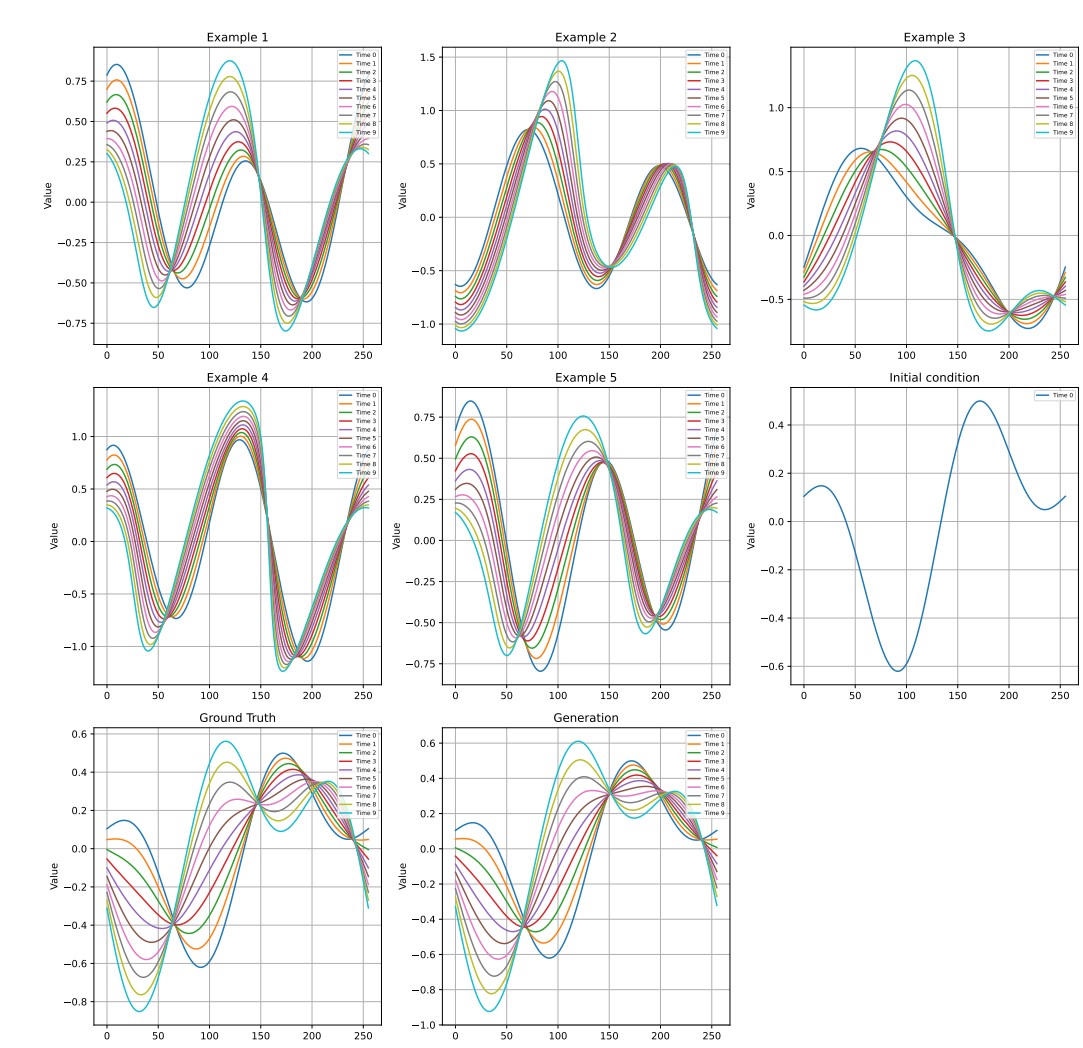

Figure 23: **Five-shot** adaptation on burgers

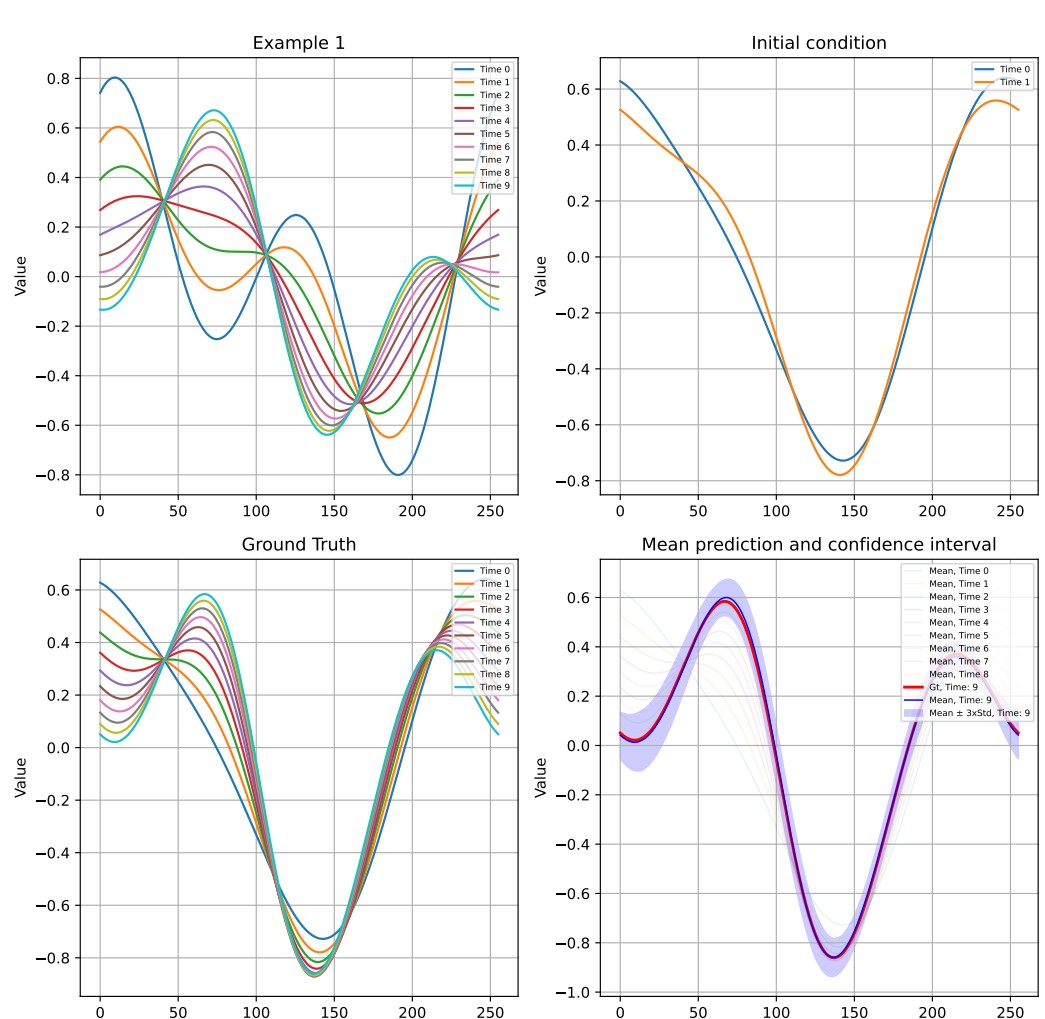

Figure 24: **Uncertainty quantification** on Burgers

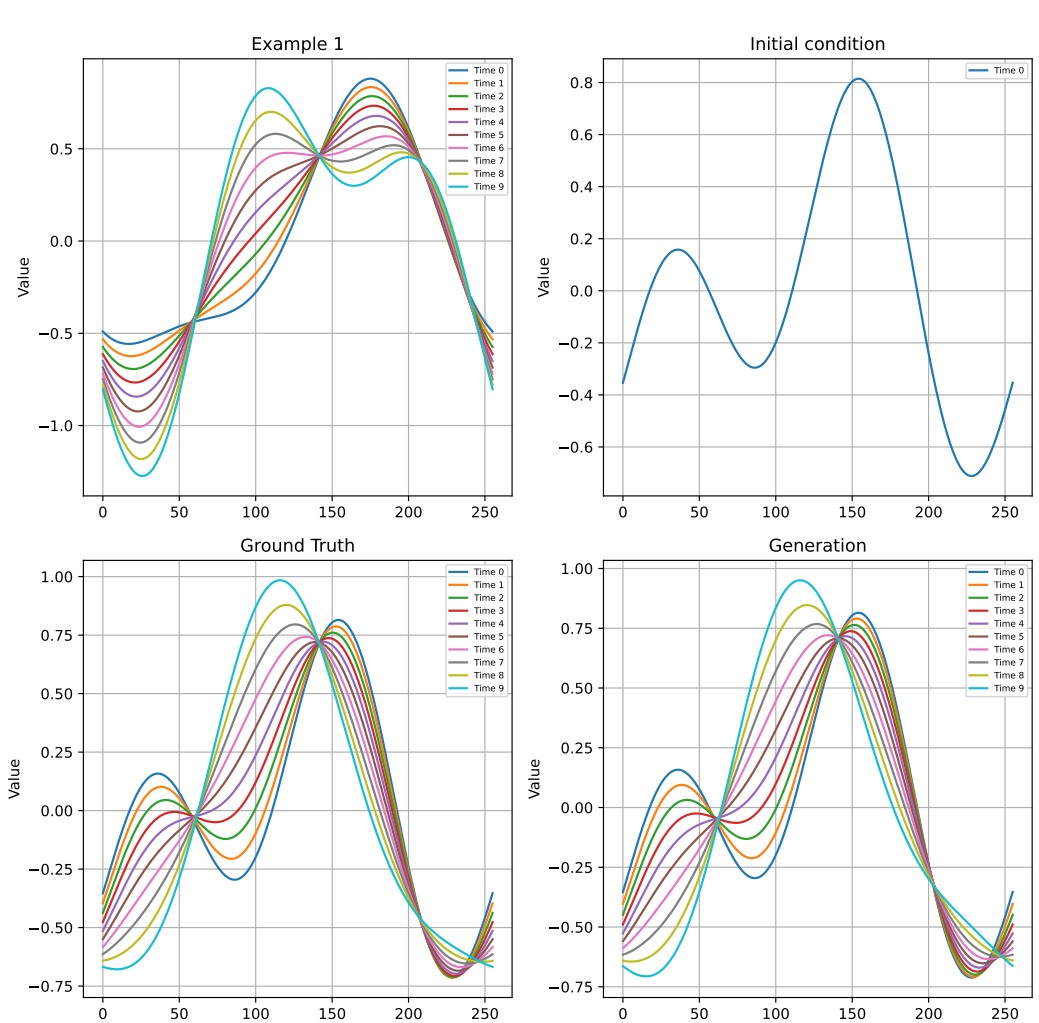

Figure 25: **One-shot** adaptation on Heat

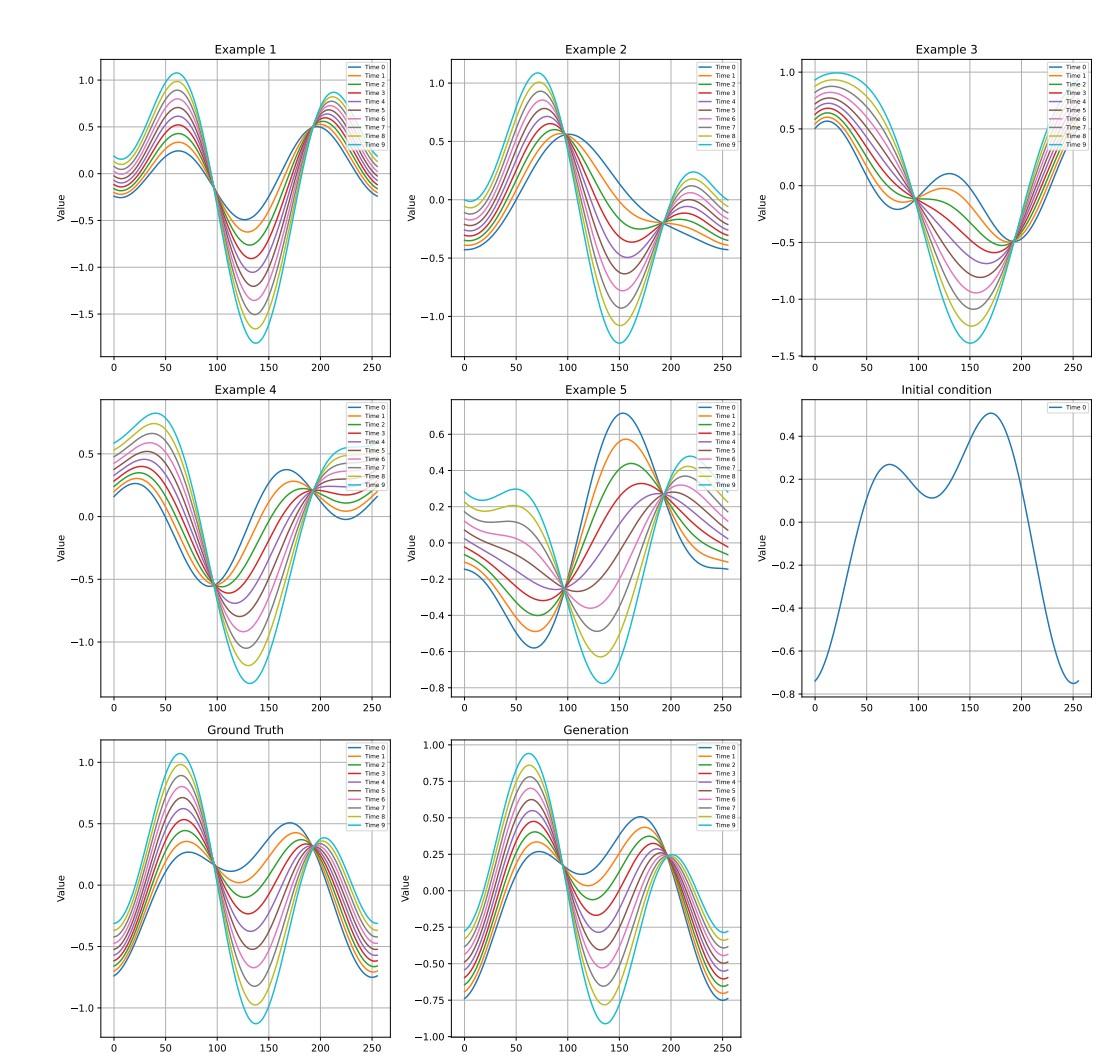

Figure 26: **Five-shot** prediction on Heat

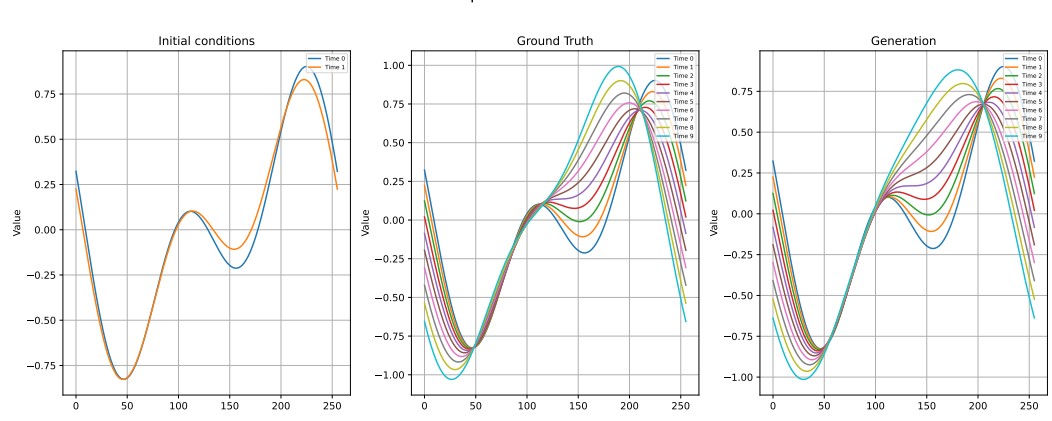

Figure 27: **Zero-shot** adaptation on Heat

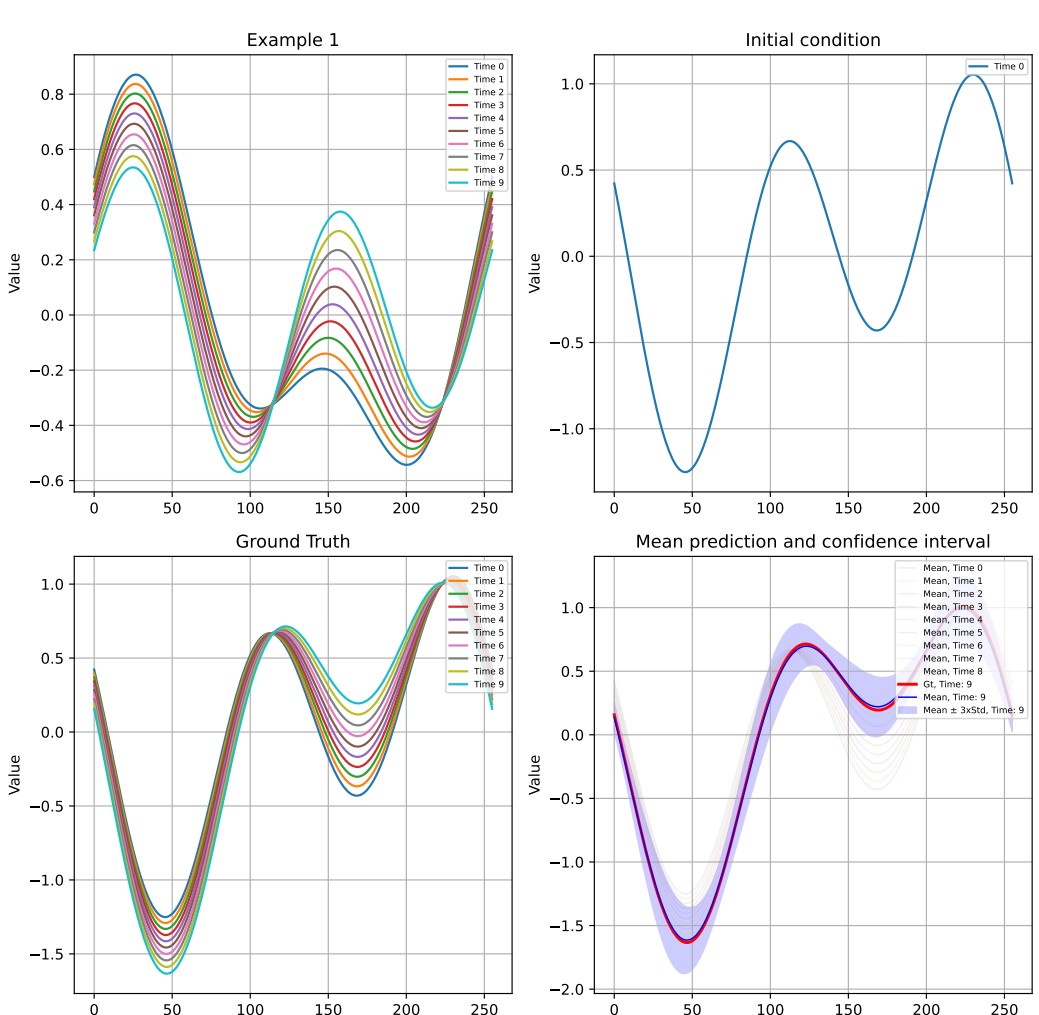

Figure 28: **Uncertainty quantification** on Heat

## E.3   HEAT

## E.4   WAVE BOUNDARY

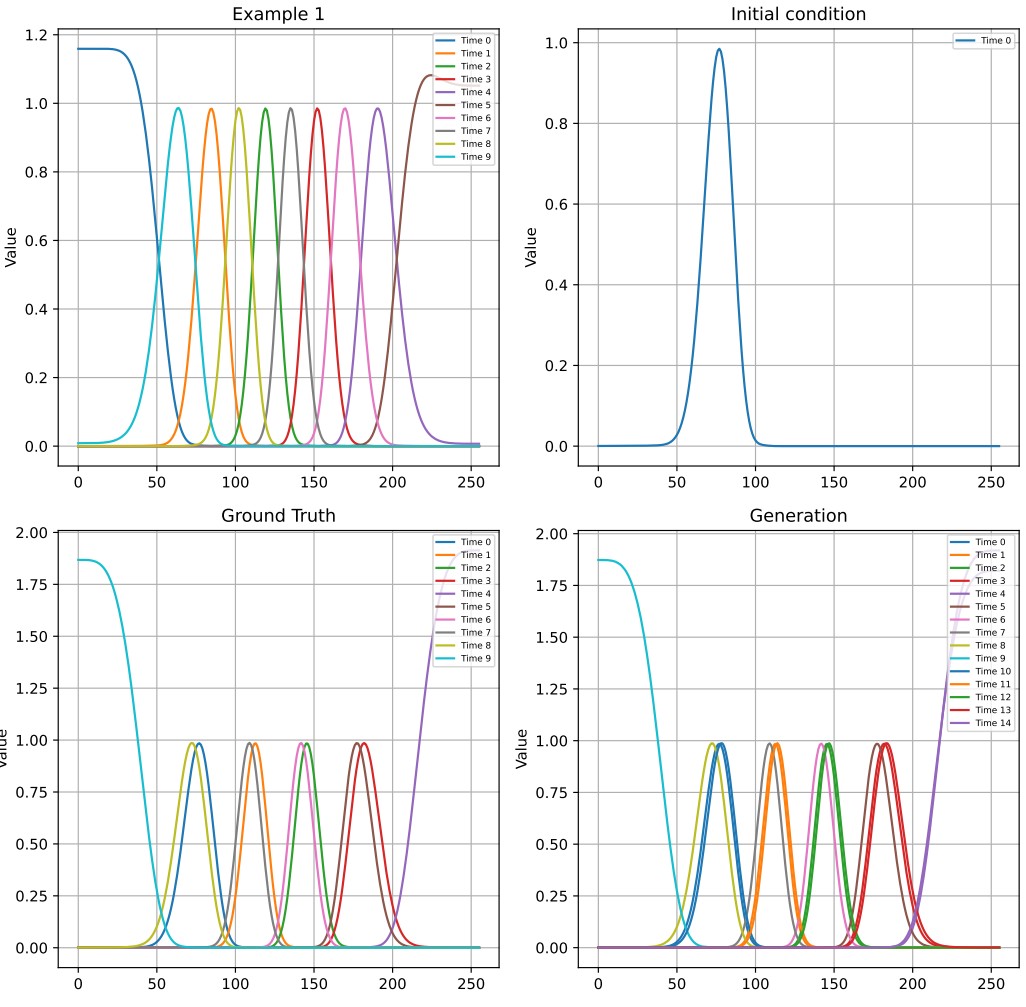

Figure 29: **One-shot** adaptation on Wave b

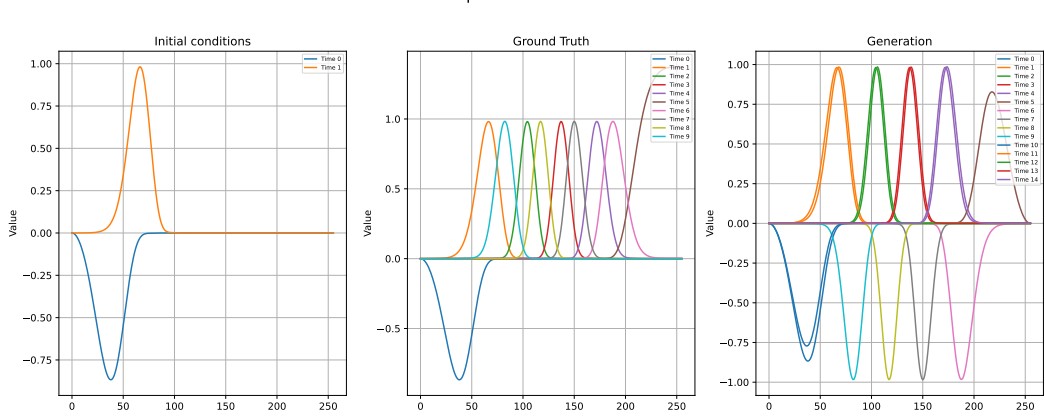

Figure 30: **Zero-shot** prediction on Wave b

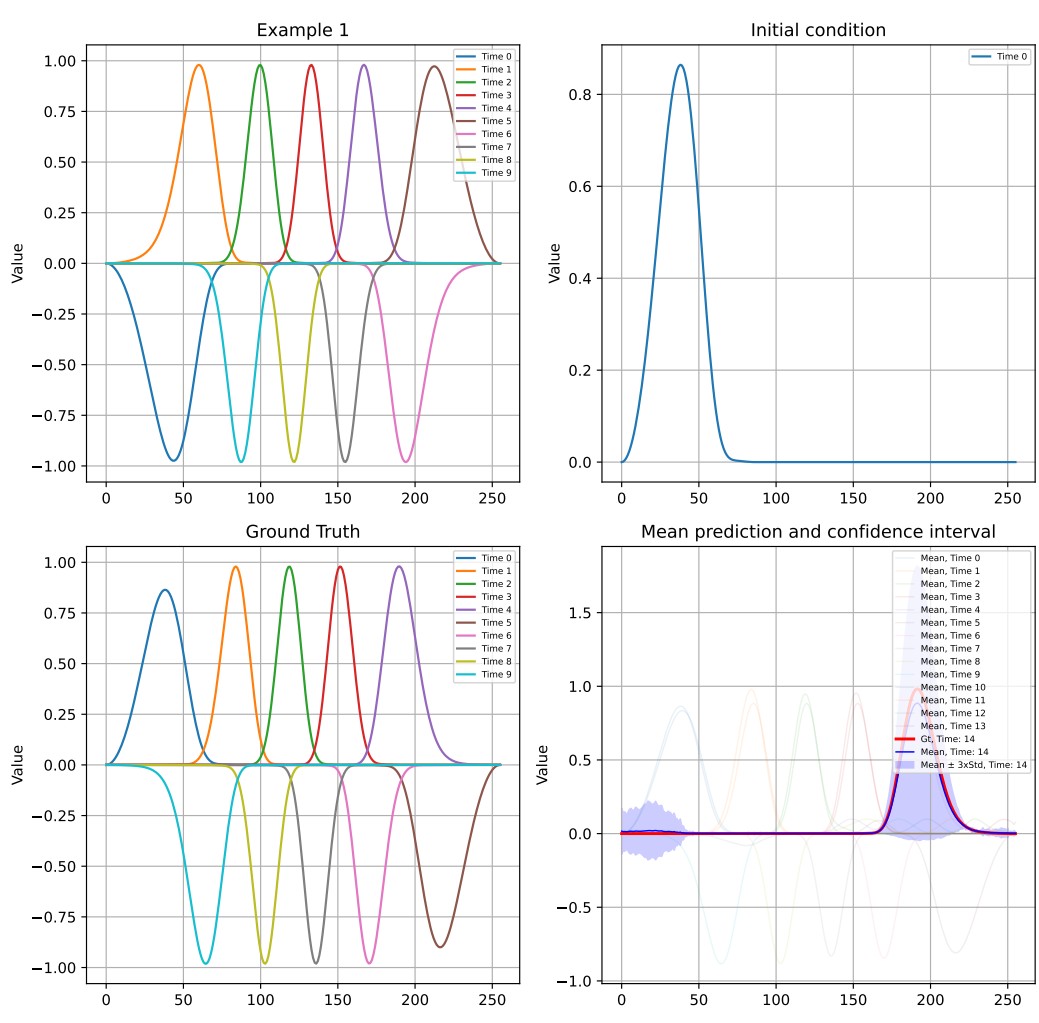

Figure 31: **Uncertainty quantification** on Wave b

## E.5 COMBINED EQUATION

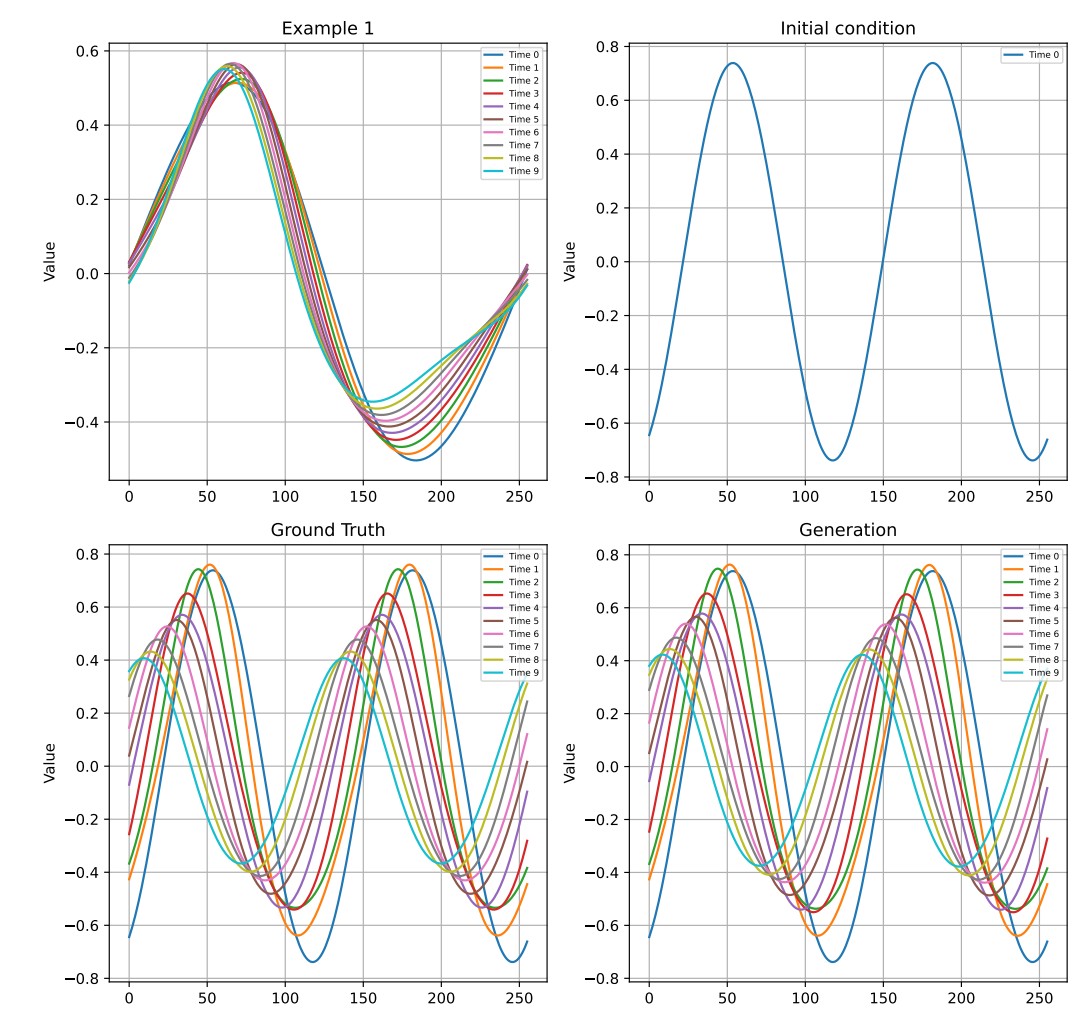

Figure 32: **One-shot** adaptation on Combined

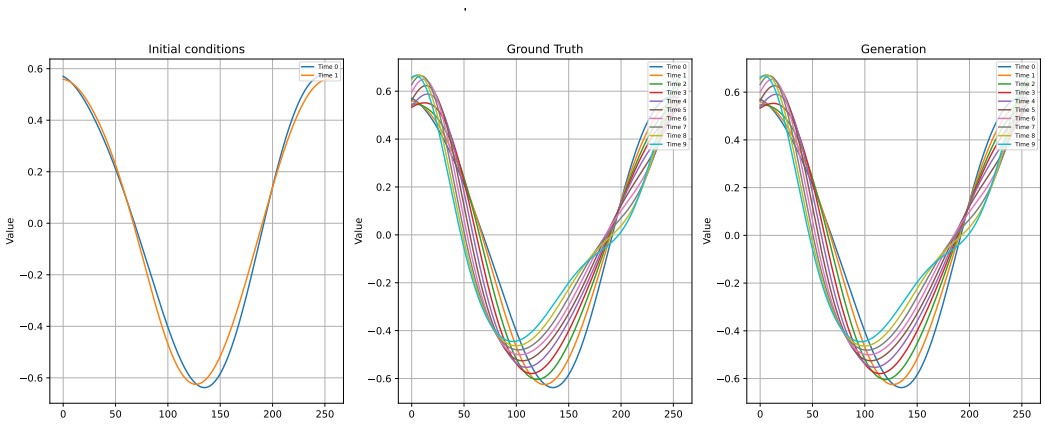

Figure 33: **Zero-shot** prediction on Combined

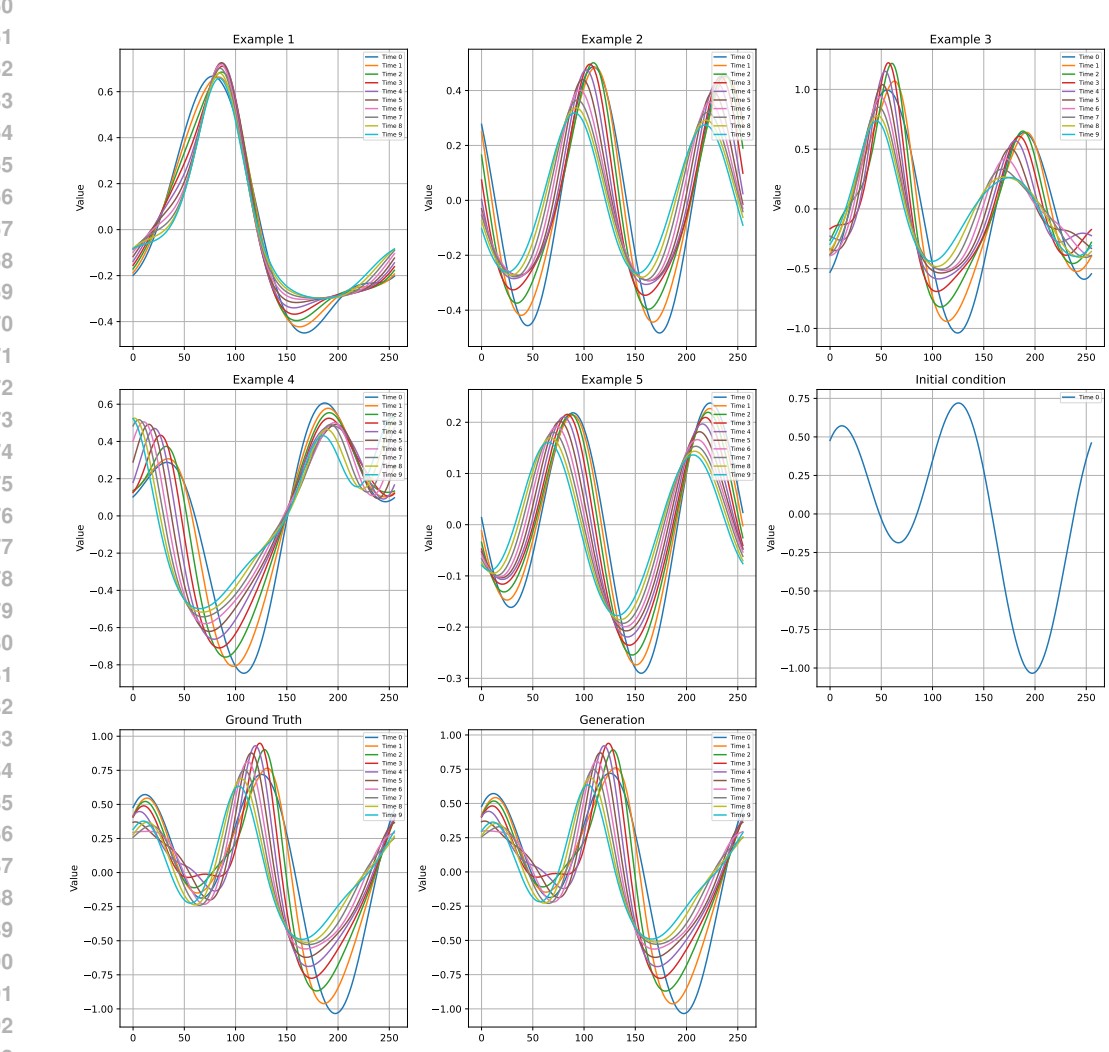

Figure 34: **Five-shot** adaptation on Combined

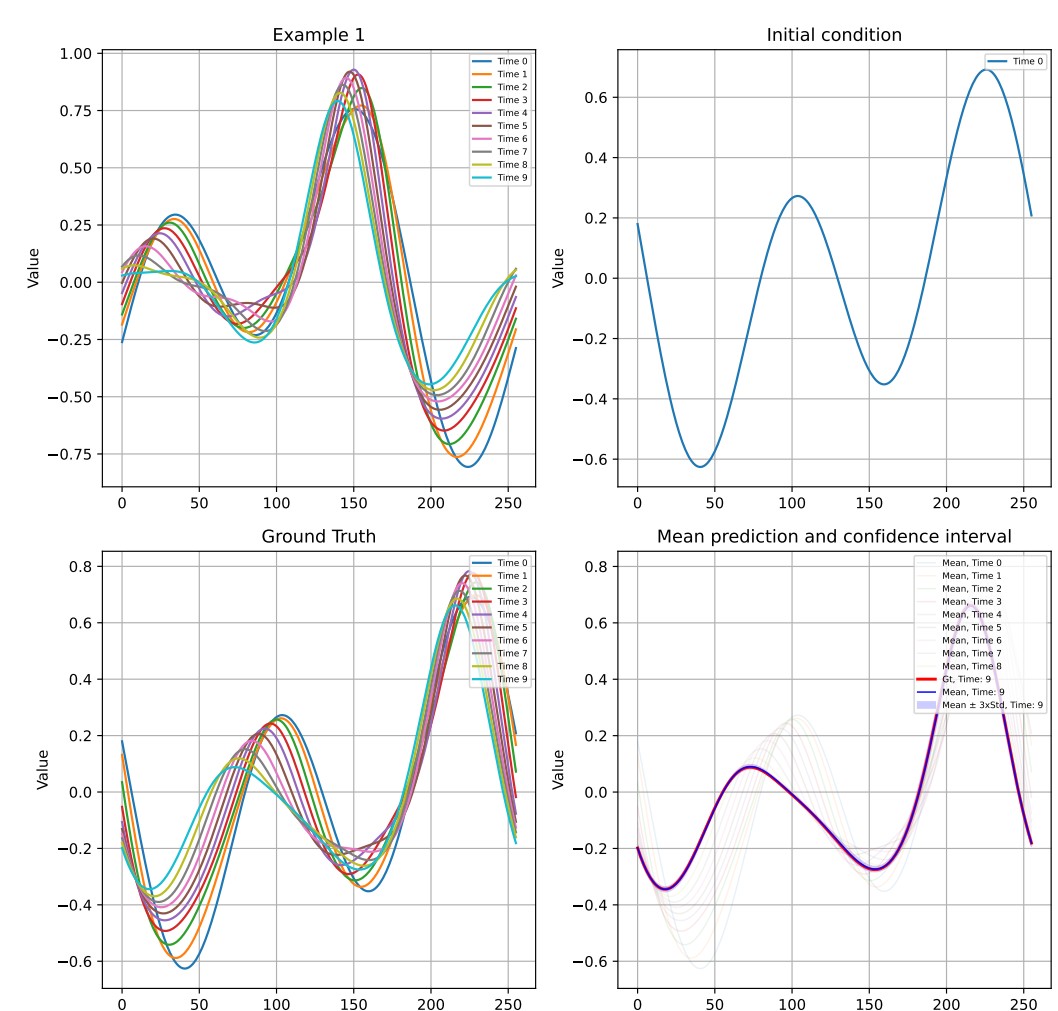

Figure 35: **Uncertainty quantification** on Combined equation

## E.6 VORTICITY

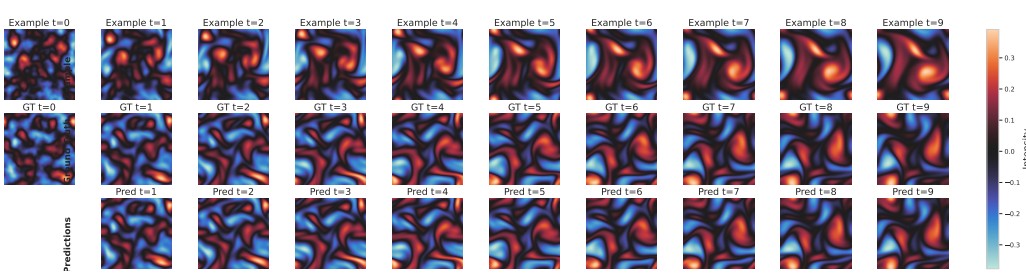

Figure 36: **One-shot** adaptation on Vorticity. Example 1.

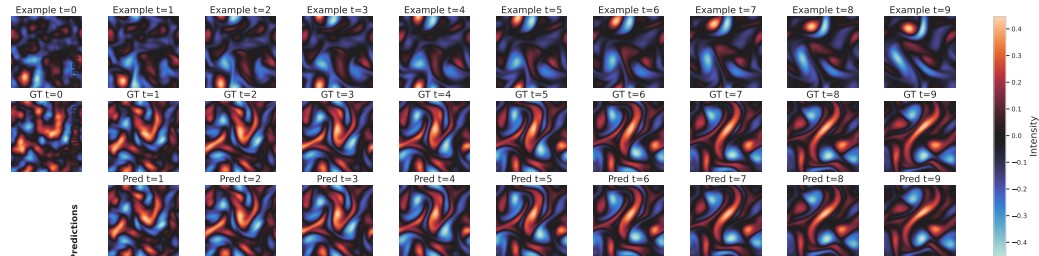

Figure 37: **One-shot** adaptation on Vorticity. Example 2.

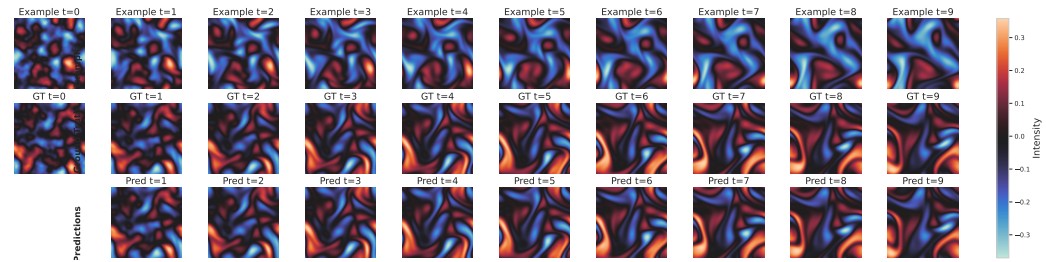

Figure 38: **One-shot** adaptation on Vorticity. Example 3.

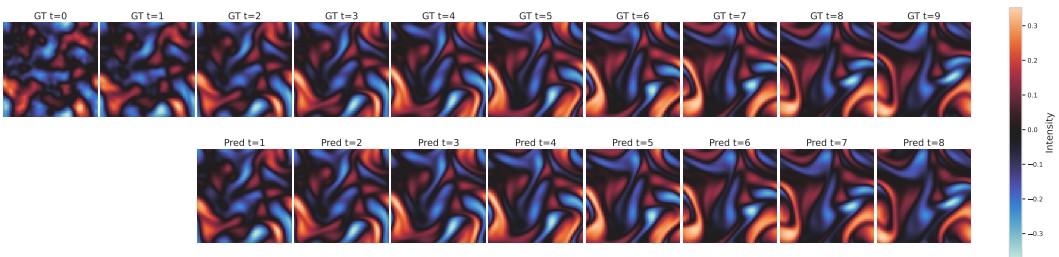

Figure 39: **Zero-shot** prediction on Vorticity. Example 1.

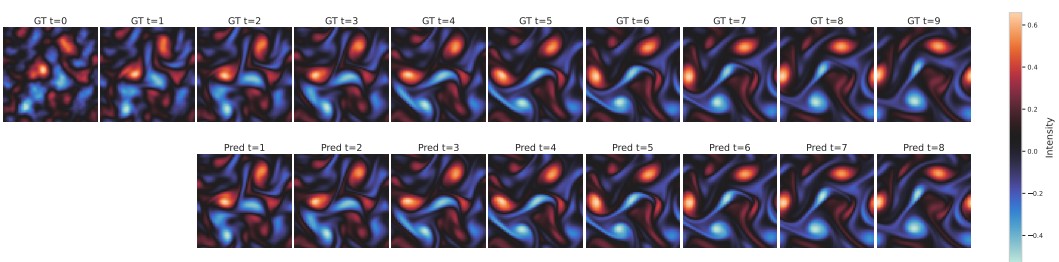

Figure 40: **Zero-shot** prediction on Vorticity. Example 2.

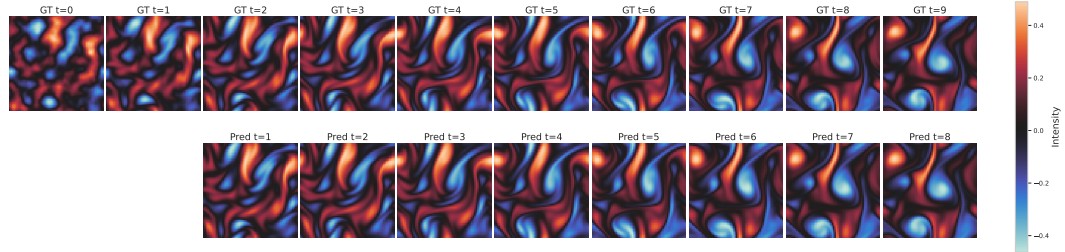

Figure 41: **Zero-shot** prediction on Vorticity. Example 3.

### E.6.1    OUT-OF-DISTRIBUTION

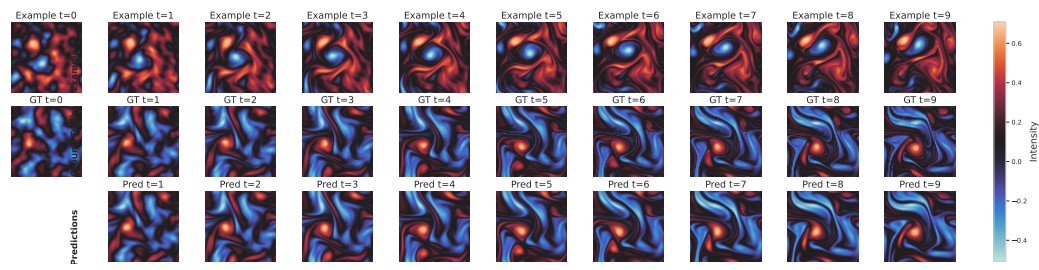

Figure 42: **One-shot OoD** adaptation on Vorticity. Example 1.

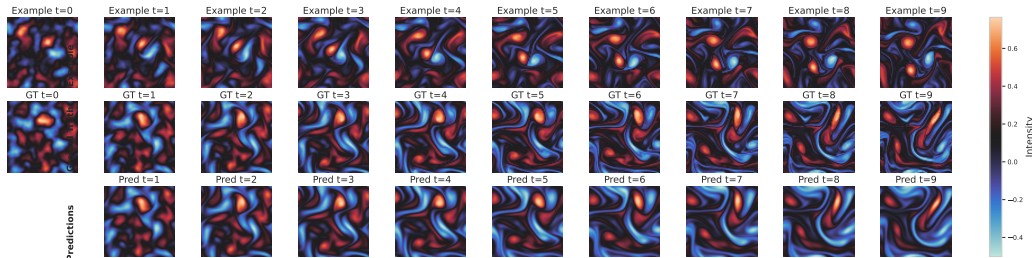

Figure 43: **One-shot OoD** adaptation on Vorticity. Example 2.

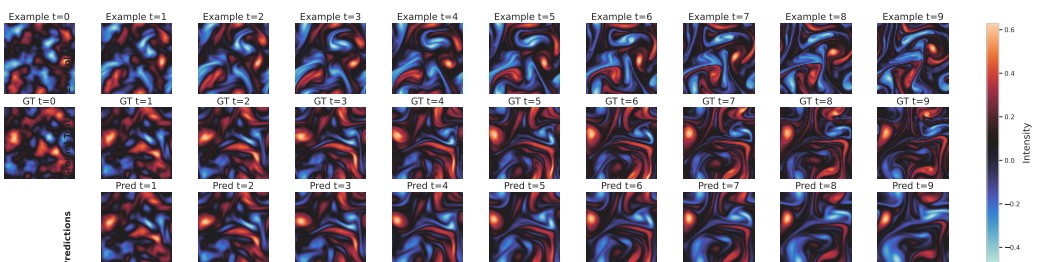

Figure 44: **One-shot OoD** adaptation on Vorticity. Example 3.

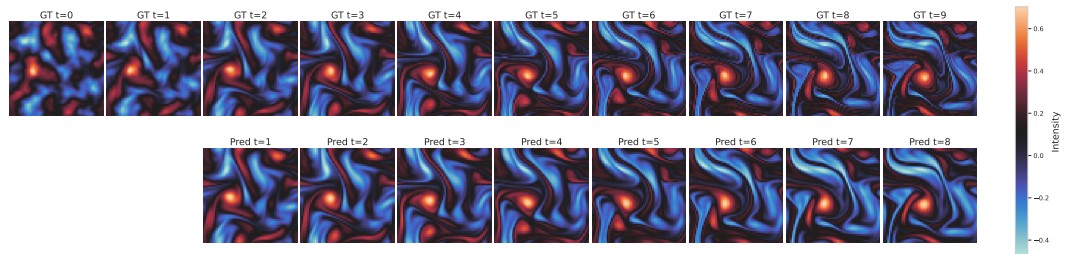

Figure 45: **Zero-shot OoD** prediction on Vorticity. Example 1.

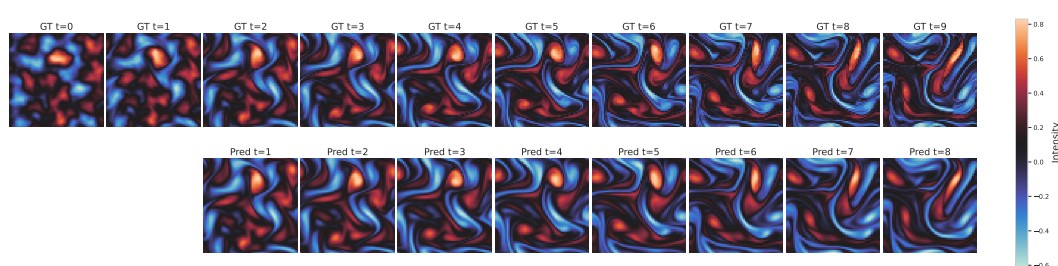

Figure 46: **Zero-shot OoD** prediction on Vorticity. Example 2.

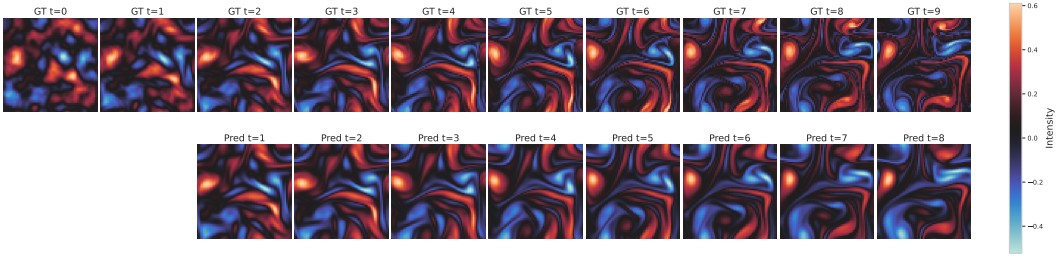

Figure 47: **Zero-shot OoD** prediction on Vorticity. Example 3.

### E.7 WAVE 2D

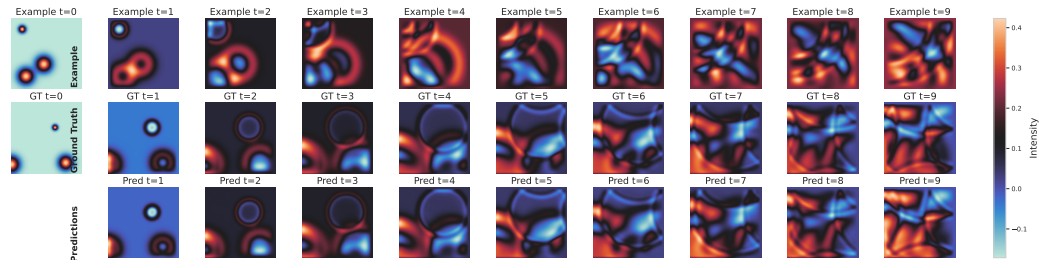

Figure 48: **One-shot** adaptation on Vorticity. Example 1.

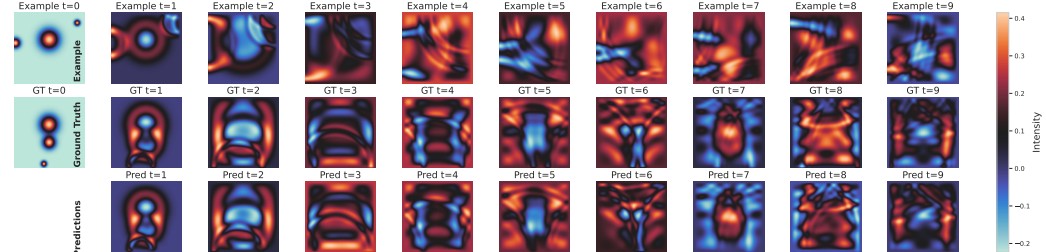

Figure 49: **One-shot** adaptation on Wave2d. Example 2.

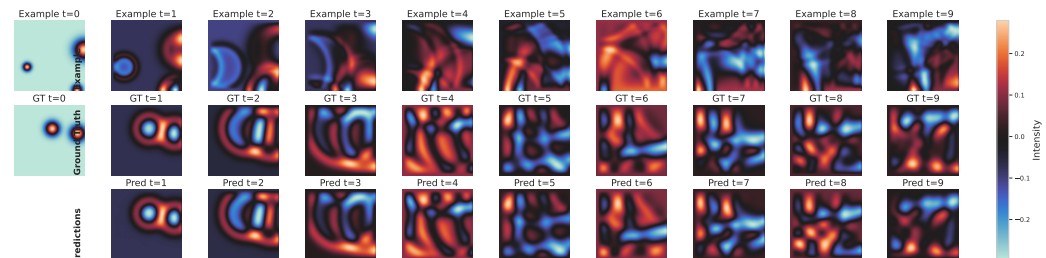

Figure 50: **One-shot** adaptation on Wave2d. Example 3.

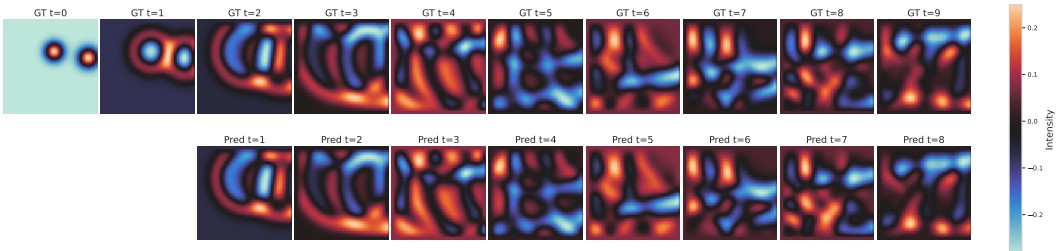

Figure 51: **Zero-shot** prediction on Wave2d. Example 1.

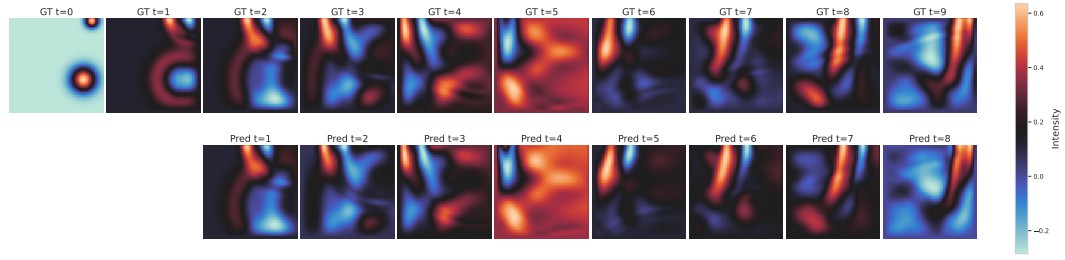

Figure 52: **Zero-shot** prediction on Wave2d. Example 2.

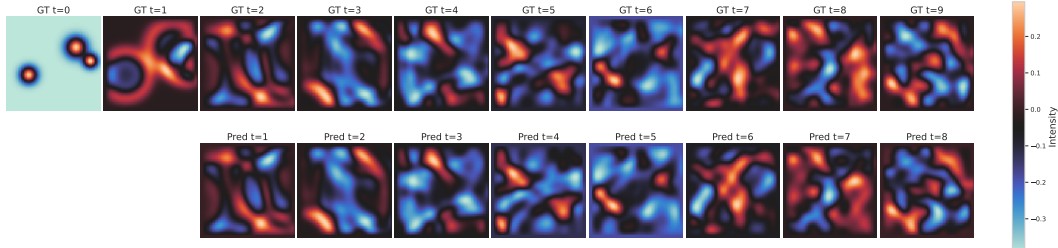

Figure 53: **Zero-shot** prediction on Wave2d. Example 3.

