# OpenReview forum: "Zebra: In-Context and Generative Pretraining for Solving Parametric PDEs"
_ICLR.cc/2025/Conference — Submitted to ICLR 2025_

### Official Review · Reviewer_UXkn · 2024-10-23

**Soundness:** 2
**Presentation:** 3
**Contribution:** 2
**Rating:** 5
**Confidence:** 3

**Summary:**

This paper addresses a critical challenge in solving parametric PDEs, where no prior knowledge of the underlying equations—such as PDE coefficients, boundary conditions, or forcing terms—can be utilized. To tackle this, the work makes two key contributions: 1) an auto-regressive framework based on VQ-VAE, and 2) an in-context learning paradigm designed to encode the implicit information of the underlying PDEs. The proposed methods are evaluated in two distinct settings: 1) zero-shot evaluation with temporal conditioning, where only the first few frames of data are used for inference, and 2) one-shot evaluation with context, where a new initial condition and a trajectory from the same environment are provided to guide inference.

**Strengths:**

* The paper tackles an interesting and important problem.

* This paper provides a well-organized presentation that lays out the shortcomings of existing approaches while clearly articulating the rationale behind the use of in-context learning. The figures effectively illustrate the framework of the method, enhancing clarity and aiding comprehension.

* Through extensive validation across a wide spectrum of PDEs, the method demonstrates remarkable generalization performance.

**Weaknesses:**

While the paper addresses an interesting problem, there are two primary concerns: 1) the proposed method lacks novelty compared to MPP [McCabe et al., 2023], which employs an autoregressive Transformer-based model and tackles a similar environment-unaware scenario with zero-shot evaluation via temporal conditioning, and 2) the empirical evaluation appears limited.

1. The main distinction between the proposed method and MPP lies in the use of the VQ-VAE framework and in-context learning. However, the rationale behind employing VQ-VAE is not clearly articulated. Since MPP’s architecture can also be trained in an autoregressive manner, what is the unique advantage of your proposed architecture for this task?

2. Although MPP focuses on zero-shot evaluation with temporal conditioning, extending it to the in-context learning setting is straightforward—by incorporating context examples into the input sequence. This reduces the novelty of your proposed approach. I suggest a direct comparison between Zebra and an MPP model trained with in-context learning, evaluating both context adaptation and zero-shot settings to highlight any distinct advantages.

3. While the in-context learning paradigm offers a method for addressing environment-unaware parametric PDEs, its effectiveness in encoding the underlying PDE information appears limited. The proposed Zebra shows only marginal improvement with additional context examples. For instance, Zebra’s performance in Vorticity 2D with a single context example (0.119) is worse than in the zero-shot scenario (0.0874). Moreover, as depicted in Figure 2, increasing the number of context examples does not consistently yield better results, with only modest gains observed (e.g., 1.14e-1 vs. 1.02e-1 in the Burger’s equation). Based on these findings, the introduction of in-context learning seems less impactful for solving parametric PDEs than it is for tasks in large language models (LLMs), offering no significant new insights for this problem domain. Could you elaborate on the possible reasons for the limited effectiveness of in-context learning in this domain, and do you have any suggestions for enhancing its impact?

4. For context adaptation settings, it would be beneficial to experiment with a larger number of context examples for both Zebra and the baselines.

5. In the zero-shot setting, consider using more frames during inference for both Zebra and the baselines.

6. Lastly, I recommend testing on more complex datasets, such as the one used in MPP, which involves a collection of four 2D PDEs (including incompressible and compressible Navier-Stokes equations, the shallow-water equation, and 2D diffusion-reaction equations) for pretraining.

7. The concept of in-context learning has been previously explored in the context of solving PDEs. Could you provide a specific comparison between Zebra and the approach in [Yang et al., 2023], highlighting key differences in methodology and applicability to different types of PDEs?

**Reference**

[McCabe et al., 2023] "Multiple physics pretraining for physical surrogate models." arXiv preprint arXiv:2310.02994 (2023).

[Yang et al., 2023] "In-context operator learning with data prompts for differential equation problems." Proceedings of the National Academy of Sciences 120.39 (2023): e2310142120.

**Questions:**

See weaknesses

---

> ### Author Response · Authors · 2024-11-22
> **Response**
>
> ## W0: Novelty
> We are sorry if there is some confusion here, but MPP and Zebra significantly differ both in their objective and in their modeling framework.
> As for the objective, MPP adopts the conventional pretraining (on a set of PDEs) and fine-tuning paradigm. The latter requires in particular a significant number of examples.
> On our side, we consider solving parametric PDEs with few-shot adaptation. Said otherwise the model is trained to adapt with only a few samples through the mechanism of in-context learning. Additionally, we focus on a scarce data regime for the adaptation.
> As for the models, MPP relies on a ViT architecture (Dosivitskiy 2020), while Zebra leverages a generative causal architecture inspired from Llama family models.
>
> ## W1: Distinction with MPP
> We believe there might be a misunderstanding: The novelty of our work lies in the in-context learning framework rather than the architecture itself. Furthermore, our model is fundamentally different from MPP. MPP employs a ViT (as noted in Appendix B of the MPP paper), whereas we use a causal architecture in a generative transformer, similar to a language model. Our model operates on discrete tokens, which is why we use a VQVAE. This enables Zebra to generate trajectory distributions—a capability not possible with MPP—and facilitates the use of in-context learning in a manner akin to language models.
>
> ## W2 : MPP-in-context
> Extending MPP for in-context learning is not straightforward. Video transformers have demonstrated limited empirical evidence of in-context learning capabilities compared to language transformers as Zebra. Following your suggestion, we however compared Zebra with MPP trained using in-context examples (referred to as MPP-in-context in the text) in both zero-shot (Table 4,  see also below) and one-shot (Table 3, see also below) settings, as detailed in Sections 4.2 and 4.3 of the revised manuscript. Zebra consistently outperforms MPP-in-context in one-shot adaptation and surpasses it on 5 out of 7 datasets in the zero-shot setting using only two frames. Notably, training MPP directly with in-context examples systematically degrades its zero-shot performance compared to vanilla MPP. This suggests that generative pretraining better models dynamics under varying context sizes, aligning with findings in the literature [1].
>
> ### W2.1 One-shot Adaptation
>
> Conditioning from a similar trajectory.
> Test results in relative L2 loss on the trajectory.
>
> | Method               | **Advection** | **Heat** | **Burgers** | **Wave b** | **Combined** | **Wave 2D** | **Vorticity 2D** |
> |----------------------|---------------|----------|-------------|------------|--------------|--------------|------------------|
> | **MPP-in-context**   | 0.0902        | 0.472    | 0.582       | 0.472      | 0.0885       | 0.390        | 0.173            |
> | **Zebra**            | _0.00794_     | _0.154_  | **0.115**   | **0.245**  | _0.00965_    | **0.207**    | **0.119**        |
>
> ### W2.2. Zero-shot Prediction from 2 Frames
>
> Conditioning from a trajectory history with 2 frames as input.
> Test results in relative L2 loss on the trajectory.
>
> | Method               | **Advection** | **Heat** | **Burgers** | **Wave b** | **Combined** | **Wave 2D** | **Vorticity 2D** |
> |----------------------|---------------|----------|-------------|------------|--------------|--------------|------------------|
> | **MPP[2]**          | 0.0075        | **0.0814** | **0.100**   | 1.0393     | 0.0250       | _0.285_      | _0.101_          |
> | **MPP-in-context**   | 0.197         | _0.204_   | _0.176_     | 1.13       | 0.0985       | 0.363        | 0.1393           |
> | **Zebra**            | _0.00631_     | 0.227     | 0.221       | _0.992_    | **0.0084**   | **0.201**    | **0.0874**       |
>
>
> ## W3 : In-context learning efficiency
> First, we emphasize that the zero-shot setting with two frames and the one-shot setting with one trajectory are not directly comparable. Performance varies according to the dataset: Zebra is better in the one-shot setting for Heat, Burgers and Wave boundary than in the zero-shot setting and it is the other way around for Advection, Combined, Wave 2D, and Vorticity 2D.
> The only comparable settings are presented in Figure 3 and Table 8 in Appendix D.1, where we see that including additional frames and examples consistently reduces error compared to pure zero-shot scenarios.
>
> Regarding the potential of in-context learning for solving PDEs, this is one of the very first works exploring this direction. We agree that demonstrating its full efficiency will likely require additional efforts from the community, as was the case with language models, for example. Specifically, this will necessitate the development and use of appropriate large-scale datasets, which are currently being created by this community [2][3].

---

> ### Author Response · Authors · 2024-11-22
> **Response 2**
>
> ## W4: Number of examples
> We experimented with up to 5 context trajectories for Zebra since our focus is specifically on few-shot learning. As for the baselines, gradient based adaptation methods, directly comparable to ZEBRA are known to saturate fast with respect to the number of context examples, see [4] for example (Appendix D.2, Table 5): test errors quickly plateau as the number of context trajectories increases.
>
>
> ## W5 : Number of frames
> Our primary goal is to evaluate the generalization of neural solvers given minimal observations since our focus is on scarce data settings. We chose a constrained regimes: 2 timestamps for zero-shot and 1 trajectory + 1 condition for one-shot settings. These configurations form the core focus of our paper.
>
>
> ## W6 : Datasets
> The PDEBench datasets used in MPP exhibit limited parameter variation (around 10), making them unsuitable for our setting. In contrast, our datasets are specifically designed to represent a rich variety of phenomena. For one equation, we considered up to 1200 different parameter values when generating our datasets, compared to the O(10) variations in PDEBench.
>
> ## W7 : ICON
> We tried, but were unable to compare experimentally our method to ICON because it could not handle the spatio-temporal forecasting tasks addressed in our paper. While there are some similarities, the two models target different applications and are built on distinct principles. ICON is an encoder-decoder model relying on cross-attention conditioned on context embeddings, generating outputs pointwise without explicitly modeling process dynamics. In contrast, ZEBRA is a generative model (similar to GPT) employing an encode-process-decode framework. It models dynamics in latent space through its process module. A more precise claim would be that ZEBRA is the first in-context generative model designed for PDEs.
>
>
> [1] Radford et al, 2018. Improving Language Understanding by Generative Pre-Training.
>
> [2] Ohana et al, Neurips 2024. The Well: a Large-Scale Collection of Diverse Physics Simulations for Machine Learning.
>
> [3] Bonnet et al., Neurips 2022. AirfRANS: High Fidelity Computational Fluid Dynamics Dataset for Approximating Reynolds-Averaged Navier-Stokes Solutions
>
> [4] Kassaï Koupaï et al., Neurips 2024. GEPS: Boosting Generalization in Parametric PDE Neural Solvers through Adaptive Conditioning.

---

> > ### Comment · Reviewer_UXkn · 2024-11-25
> >
> > I appreciate the authors' efforts in addressing the feedback during the rebuttal process. While some of my concerns have been resolved, I regret that I am unable to raise my score, as the current manuscript still does not meet the acceptance criteria in my opinion.
> >
> > ---
> >
> > ### 1. **Main Concern: Insufficient Demonstration of In-Context Learning Potential**
> >    - Given the manuscript’s lack of significant innovation, I would expect it to offer substantial practical contributions to the field. However, the current work falls short of fully demonstrating the potential of in-context learning for solving PDEs. For instance, as shown in Figure 2, increasing the number of context examples does not consistently lead to better results, with only modest improvements observed (e.g., 1.14e-1 vs. 1.02e-1 for Burger’s equation).
> >
> > ---
> >
> > ### 2. **Second Concern: Simplicity of the Benchmark**
> >    - One possible explanation for the limited improvement lies in the simplicity of the evaluation benchmark. In the zero-shot setting, the underlying PDE coefficients can be inferred from as few as two data frames, suggesting that minimal context information is sufficient to solve the problem.
> >    - While I recognize that the dataset generation aimed to increase parameter variation, the parameters remain within a regime of relatively simple patterns. For example, the viscosity for the 2D Navier-Stokes (VORTICITY) equation ranges from 1e-3 to 1e-2, which lies outside the turbulent regime.
> >
> > ---
> >
> > ### Recommendations
> > To address these limitations, I recommend evaluating the model in more complex settings where richer context information is essential for inferring intricate flow patterns. Examples include the incompressible Navier-Stokes equation with viscosities ranging from 1e-5 to 1e-4 or the compressible Navier-Stokes equation as explored in MPP. Such scenarios would provide a more rigorous assessment of the model’s capabilities. Additionally, tackling more challenging PDE benchmarks may necessitate innovative or tailored approaches, which could further enhance the novelty and impact of the work.
> >
> > ---
> >
> > ### Questions Regarding the Rebuttal
> > 1. You mentioned in **W3: In-context learning efficiency** that the zero-shot setting with two frames and the one-shot setting with one trajectory are not directly comparable. Could you clarify why this is the case? Are the training and testing datasets the same for these two settings?
> > 2. While I understand that an empirical comparison with ICON is infeasible, it would be helpful to include a discussion in the related works section about the differences between Zebra and ICON. This would provide readers with a clearer understanding of their unique characteristics and contributions.

---

> > > ### Author Response · Authors · 2024-11-27
> > > **Response**
> > >
> > > Thank you for your detailed feedback and the time taken for answering. We however respectfully disagree with some of your comments, and we hope the following clarifications address your concerns effectively.
> > >
> > >
> > > ### 1. Main Concern: Innovation
> > >
> > > **Innovation:**
> > > We believe the originality of our work lies in leveraging in-context examples to eliminate the need for gradient-based updates, as employed in SOTA adaptation frameworks like CoDA and CAPE. This pretraining strategy is novel in the PDE community, representing a significant contribution with the potential for a large impact if scaled with adequate computational resources and datasets.
> > >
> > > **Assessment:**
> > > In-context learning is conventionally evaluated in zero-shot, one-shot, and few-shot settings, as exemplified in the GPT-3 paper [1]. Our work adheres to this framework, focusing on zero-shot and one-shot evaluation. Notably, previous studies (e.g., [2]) have observed that adding more context examples can sometimes degrade performance due to attention saturation in longer sequences. Therefore, it is not surprising that increasing the number of context examples does not consistently yield better results in our case too. This phenomenon aligns with observations in the GPT-3 paper [1], where one-shot performance surpasses few-shot for smaller models (less than 1 billion parameters), while few-shot excels for larger models (Figure 3.12, page 25 of [1]).
> > >
> > > In our experiments, the optimal number of context examples depends on the dataset, and in practice, adding up to three examples often improves performance, as shown in the table below (with 2 frames in inputs). We believe these results clearly demonstrate the advantages of in-context learning.
> > >
> > >
> > > | Dataset    | 0-shot  | 1-shot  | 2-shot  | 3-shot  |
> > > |------------|---------|---------|---------|---------|
> > > | Advection  | 0.0063  | 0.0060  | 0.0059  | **0.0041** |
> > > | Heat       | 0.2276  | 0.0470  | 0.0464  | **0.0447** |
> > > | Burgers    | 0.2206  | 0.0538  | 0.0501  | **0.0478** |
> > > | Wave-b     | 0.9925  | 0.1912  | **0.1558** | 0.1726 |
> > > | Combined   | 0.0084  | 0.0054  | **0.0050** | 0.0062 |
> > >
> > > ---

---

> > > > ### Author Response · Authors · 2024-11-27
> > > > **Response 2**
> > > >
> > > > ### 2. Simplicity of the benchmark
> > > >
> > > > **Zero-shot setting:**
> > > > To clarify, the zero-shot setting involves observing a new trajectory with a fixed number of snapshots but without context examples. While we evaluated Zebra using a history of two frames, this choice is not restrictive. It does not imply that the entire dynamics is captured with only two frames, as suggested. The datasets used in our experiments represent complex dynamics, encompassing seven families of PDEs with varying coefficients, forcing terms, initial/boundary conditions. For a single equation, this translates to approximately 1,000 unique instances.
> > > >
> > > > Our setup aligns with other works in the literature. For instance, neural weather forecasters use two snapshots for prediction ([3,4]). Similarly, [5] proposed a foundation model pretrained using only one initial snapshot, which is even more limited than our setting but still applicable to complex datasets.
> > > >
> > > > Following your suggestion, we conducted a zero-shot evaluation by providing more input frames. To ensure consistency in the settings, we use the field at \( T=10 \) as the ground truth and perform a one-step prediction while varying the context history. Providing additional frames improves performance, as shown in the table below:
> > > >
> > > > | Dataset   | Model            | 2 frames | 5 frames | 9 frames |
> > > > |-----------|------------------|----------|----------|----------|
> > > > | Wave2d    | MPP-in-context   | 0.78     | 0.23     | 0.14     |
> > > > | Wave2d    | Zebra            | 0.27     | 0.09     | 0.08     |
> > > > | Vorticity | MPP-in-context   | 0.047    | 0.042    | 0.041    |
> > > > | Vorticity | Zebra            | 0.047    | 0.024    | 0.022    |
> > > >
> > > > ---
> > > >
> > > > Increasing the number of frames improves the performance for all the models. These results demonstrate the complexity of the datasets and the model's robustness when provided with additional temporal context. However our main evaluation framework leveraging one initial state as for classical PDE solving or 2 past frames allows a fair comparison of the models. Besides there are several simulation problems that rely on one initial state only.
> > > >
> > > > **Out-of-distribution results:**
> > > > Following your suggestion, we extended our evaluation to include novel trajectories with viscosity in the range of \( [10^{-5}, 10^{-4}] \). We have also added a visualization in the main section of the revised paper (Figure 4) and additional plots in Section E. The results below demonstrate Zebra's robustness to changes in viscosity:
> > > >
> > > > | Model            | Zero-shot (2 frames) | One-shot (1 frame) |
> > > > |-------------------|----------------------|--------------------|
> > > > | MPP-in-context    | 0.355               | 0.368              |
> > > > | Zebra             | **0.308**           | **0.317**          |
> > > >
> > > > ---
> > > >
> > > > ### Questions
> > > >
> > > > **Q1. Fixed examples vs. fixed frames:**
> > > > To ensure meaningful comparisons, we adopt two evaluation protocols: (1) fixing the number of in-context examples while varying the number of frames, or (2) fixing the number of frames while varying the number of context examples. This ensures consistency in the amount of information provided to the model. Combining these variations without control would lead to ambiguous conclusions.
> > > >
> > > > **Q2. ICON-related section:**
> > > > We will add a dedicated section on ICON in the final version.
> > > >
> > > > ---
> > > >
> > > > ### Additional Section on ICON
> > > >
> > > > ICON is an encoder-decoder model that leverages cross-attention conditioned on context embeddings to generate pointwise outputs without explicitly modeling underlying process dynamics, focusing primarily on operator learning. In contrast, Zebra adopts a generative approach inspired by GPT, employing an encode-process-decode framework. Zebra's process module captures dynamics in a latent space, enabling comprehensive modeling of trajectory evolution, which sets it apart from ICON.
> > > >
> > > > ---
> > > >
> > > > [1] Brown et al., 2020. Language Models are Few-Shot Learners.
> > > > [2] Jiuhai Chen et al., 2023. How Many Demonstrations Do You Need for In-Context Learning?
> > > > [3] Lam et al., 2023. GraphCast: Learning Skillful Medium-Range Global Weather Forecasting.
> > > > [4] Bodnar et al., 2024. A Foundation Model for the Earth System.
> > > > [5] Herde et al., 2024. POSEIDON: Efficient Foundation Models for PDEs.

---

> > > > > ### Comment · Reviewer_UXkn · 2024-12-02
> > > > >
> > > > > I appreciate the detailed feedback and the additional results provided. However, my concerns remain unresolved for the following reasons:
> > > > >
> > > > > 1. **Limited Impact of Increased Context:**
> > > > >    As shown in Fig. 2 (1-shot to 5-shot with 1 frame) and the newly reported results (1-shot to 3-shot with 2 frames), increasing the number of context examples results in only marginal improvements. While I acknowledge the authors’ explanation that this could be attributed to the limited scale of the current model, **this limitation diminishes the overall significance of Zebra in its current form**.
> > > > >
> > > > > 2. **Concerns About Benchmark Suitability for In-Context Learning:**
> > > > >    I apologize for previously using the term “simplicity of the benchmark,” which may have caused confusion. I fully recognize the complexity added by the diversity of PDE parameters in the benchmarks, including coefficients, forcing terms, and initial/boundary conditions. However, as demonstrated in the zero-shot setting, the parameters of a new trajectory can often be inferred from as few as two subsequent frames. This is evident in Table 4, where MMP[2] achieves less than 10% error on 5 out of 7 benchmarks. **These results indicate that inferring PDE parameters requires minimal context information, which raises concerns about the benchmarks' suitability for evaluating in-context learning**. To clarify, I am not questioning the zero-shot experimental setup or requesting results with more temporal frames. Rather, I am highlighting that the current benchmarks may not fully reflect the potential of in-context learning.
> > > > >
> > > > > 3. **Insufficient OOD Evaluation:**
> > > > >    While I appreciate the additional out-of-distribution (OOD) evaluation results with viscosity values ranging from 1e-5 to 1e-4, Zebra’s performance, with around 30% error, remains unsatisfactory in my opinion. Additionally, **a more appropriate evaluation of Zebra’s OOD performance would involve using diverse dataset including different equations for pretraining and testing its generalization on distinct equations**, rather than training separate models for each equation. For instance, MMP utilizes diverse pretraining data, including incompressible Navier-Stokes equations, the shallow-water equation, and 2D diffusion-reaction equations, and evaluates its OOD performance on compressible Navier-Stokes equations with distinct flow patterns.
> > > > >
> > > > > 4. **Zebra's ability to infer PDE parameters from context examples appears limited:**
> > > > >    The performance of Zebra in the zero-shot setting with two frames and the one-shot setting with one frame seems comparable to some extent. Starting with the first frame of a new trajectory, Zebra can incorporate information about PDE parameters in two ways: (1) by adding a subsequent frame (zero-shot setting with two frames) or (2) by adding another trajectory with the same PDE parameters (one-shot setting with one frame). Comparing these two settings provides insight into Zebra's ability to infer PDE parameters using these different approaches. However, in some cases, Zebra performs worse in the one-shot setting with one frame, which seems counterintuitive since the entire trajectory in the one-shot setting should offer more information for inferring PDE parameters than two subsequent frames. **This result suggests that Zebra has limitations in leveraging context examples for inferring PDE parameters compared to using two subsequent frames. Furthermore, the results do not clearly demonstrate the advantages of in-context learning over simply using temporal conditioning.**
> > > > >
> > > > > Overall, since in-context learning has been extensively explored in natural language processing, directly applying it to PDE problems lacks sufficient innovation. Therefore, I believe **it is essential to conduct experiments using larger-scale models and pretraining datasets with greater complexity and diversity to fully demonstrate its potential for solving PDE problems.**

---

> > > > > > ### Author Response · Authors · 2024-12-04
> > > > > > **Final response 1/2**
> > > > > >
> > > > > > Thank you for your feedback. We are sorry if our previous responses did not fully address your concerns. As the discussion phase is nearing its conclusion, we would like to take this opportunity to provide one final set of results to reinforce the validity of our approach and clarify some key points.
> > > > > >
> > > > > > ---
> > > > > >
> > > > > > ### W1: Comparison Between One-Shot and Few-Shot Settings
> > > > > >
> > > > > > We are unsure why you insist on comparing one-shot and few-shot settings, as we believe the more relevant comparison is between the **best few-shot performance** and the **zero-shot setting**. Below is a summary of the relative improvements in few-shot settings compared to 0-shot and 1-shot (with 2 frames as inputs):
> > > > > >
> > > > > > | Model          | Improvement vs. 1-shot | **Improvement vs. 0-shot** |
> > > > > > |---------------------|----------------------------|-----------------------------|
> > > > > > | Advection           | 32%                       | 35%                         |
> > > > > > | Heat                | 5%                        | 80%                         |
> > > > > > | Burgers             | 12%                       | 78%                         |
> > > > > > | Wave-b              | 20%                       | 84%                         |
> > > > > > | Combined            | 8%                        | 40%                         |
> > > > > > | **Average**         | **15%**                   | **63%**                     |
> > > > > >
> > > > > > As shown above, the few-shot setting reduces error by a factor greater than 2 compared to zero-shot. Even using only two or three examples yields a **63% relative improvement**, which is significant given that no fine-tuning is involved. This demonstrates that just a few similar trajectories can significantly enhance the accuracy of a neural solver that already uses a temporal history (here 2 frames).
> > > > > >
> > > > > >
> > > > > > ### W2: Benchmark suitability
> > > > > >
> > > > > > We respectfully disagree with your comment. We do not assume that two frames are sufficient to solve the task; this is simply the evaluation setup we adopted. While performance improves with more frames, two frames yielded reasonable results. It is worth noting that a recent benchmark [1] introducing complex datasets used 4 snapshots as inputs, which is not significantly different from our setup.
> > > > > >
> > > > > >
> > > > > > ### W3: OoD Performance
> > > > > >
> > > > > > It is unsurprising that OoD scores are worse in regimes like \(10^{-5}\)–\(10^{-4}\) compared to \(10^{-3}\)–\(10^{-2}\). However, Zebra demonstrates greater robustness compared to MPP-in-context and outperforms CAPE and CODA in these OoD scenarios. We believe this robustness is a significant contribution.
> > > > > >
> > > > > > We also want to reiterate that evaluating models on entirely different equations is outside the scope of our work. Our focus is on robustness within the parameter space, which aligns with the objectives of our study.

---

> > > > > > > ### Author Response · Authors · 2024-12-04
> > > > > > > **Response 2/2**
> > > > > > >
> > > > > > > ### W4: Understanding In-Context Learning in One-Shot Settings
> > > > > > >
> > > > > > > We believe that comparing the one-shot setting with one-frame and the zero-shot with two frames is not ideal, because in the one-shot scenario, the model first has to predict the second frame and then the rest of the sequence. This can give an edge to the zero-shot setting when the system is markovian. To demonstrate this, we conducted an additional experiment where the autoregressive transformer generates the second frame using the first frame and the context example. Afterward, we discarded the context example and used only the first two frames to generate the remainder of the sequence. We refer to this configuration as **Zebra one-shot truncated**.
> > > > > > >
> > > > > > > Below is the comparison of **Zebra one-shot truncated** with the standard zero-shot settings:
> > > > > > >
> > > > > > > | **Method**                  | **Advection** | **Heat** | **Burgers** | **Wave-b** | **Combined** | **Wave 2D** | **Vorticity 2D** |
> > > > > > > |-----------------------------|---------------|----------|-------------|------------|--------------|--------------|------------------|
> > > > > > > | Zebra zero-shot 2 frames    | 0.00631       | 0.227    | 0.221       | 0.992      | 0.0084       | 0.201        | 0.0874           |
> > > > > > > | **Zebra one-shot truncated**    | 0.0100        | 0.385    | 0.183       | 0.962      | 0.0118       | 0.234        | 0.1211           |
> > > > > > >
> > > > > > > From the results, we can see that **Zero-shot with 2 frames** outperforms Zebra one-shot truncated, as the former uses two ground-truth frames, while the latter uses one ground-truth frame and one prediction. This can explain why in the one-shot setting it is possible to obtain better results in the zero-shot setting with 2 frames on markovian systems (advection, combined, vorticity, wave).
> > > > > > >
> > > > > > > Then to understand, if in-context learning is used along the generation of the trajectory. We can compare it with the classical one-shot:
> > > > > > >
> > > > > > > | **Method**                  | **Advection** | **Heat** | **Burgers** | **Wave-b** | **Combined** | **Wave 2D** | **Vorticity 2D** |
> > > > > > > |-----------------------------|---------------|----------|-------------|------------|--------------|--------------|------------------|
> > > > > > > | Zebra one-shot 1 frame      | 0.00794       | 0.154    | 0.115       | 0.245      | 0.00965      | 0.207        | 0.119            |
> > > > > > > | **Zebra one-shot truncated**    | 0.0100        | 0.385    | 0.183       | 0.962      | 0.0118       | 0.234        | 0.1211           |
> > > > > > >
> > > > > > >
> > > > > > > Clearly, **Zebra one-shot** consistently outperforms Zebra one-shot truncated, demonstrating that the information in the context example is used throughout all the generation.  This shows that in-context learning allows the model to effectively leverage information from examples during the trajectory generation, resulting in better accuracy compared to truncating the context after generating the second frame and relying on temporal conditioning.
> > > > > > >
> > > > > > > ### Overall Comment
> > > > > > >
> > > > > > > To summarize the discussion, we would like to emphasize that:
> > > > > > >
> > > > > > > 1. Our work is distinct from foundation models. As explained to Reviewer 6wsG, our framework is not intended to propose a foundation model but instead focuses on in-context learning (ICL), which we consider a distinct and valuable research direction.
> > > > > > > 2. ICL is not tied to model size or dataset diversity. While larger models and diverse datasets can enhance performance, ICL can be effective even in smaller, targeted settings.
> > > > > > > 3. Our approach outperforms prior adaptation methods (e.g., CAPE, CODA), which also do not rely on large datasets, and performs on par with architectures like MPP designed for temporal conditioning.
> > > > > > > 4. Experimental results clearly demonstrate the effectiveness of ICL in improving neural solver accuracy and robustness, without relying on gradient-based adaptation.
> > > > > > >
> > > > > > > We hope this final round of evidence and clarifications sufficiently addresses your concerns and highlights the strengths and contributions of our work.
> > > > > > >
> > > > > > > Thank you for your time and consideration.
> > > > > > >
> > > > > > > Best regards.
> > > > > > >
> > > > > > > [1] Ohana et al. The Well: a Large-Scale Collection of Diverse Physics Simulations for Machine Learning

---

### Official Review · Reviewer_4Hd3 · 2024-10-28

**Soundness:** 3
**Presentation:** 3
**Contribution:** 2
**Rating:** 6
**Confidence:** 4

**Summary:**

This paper addresses the problem of generalization of purely data-driven neural solvers to new PDE parameter values, without the need for gradient descent at adaptation time. Building on its successes in the language and vision communities, it proposed to use in-context pretraining (ICP) with a generative autoregressive transformer model to quickly adapt to new situations based on flexible contexts that can either be extra trajectories or historical states from the same underlying dynamics.

**Strengths:**

1) One of this paper's strongest point is its creative combinations of existing ideas. It successfully learns dynamics in latent space and illustrates a transversal application of vision and language techniques to physical systems.
2) Although the related work section lacks in certain areas, the paper is overall well-presented, with extensive experimentation across a wide range of PDEs.
3) For Markovian systems such as the ones considered, zero-shot learning with temporal conditioning is just a special case of few-shots learning. This works picks up on that and suitably adapts CoDA [4] for zero shot learning. This is a great contribution, just as the adaption of the CLS vision transformers for parametric PDEs is.
4) The work has remarkable analysis of the uncertainty quantification capabilities built-in the Zebra framework. Specifically, the consideration of 3 different metrics helps understand how one can adjust the temperature $\tau$ to best suit our needs.

[4] Generalizing to New Physical Systems via Context-Informed Dynamics Model, ICML 2022

**Weaknesses:**

The paper has clear strenghts in its novel appreach, but lacts in other areas like related work and analysis. The main weaknesses I find are:
1) I fail to see how the third bullet point counts as a contribution (cf. line 90). Evaluation of one's proposed framework, however extensive, is to be expected.
2) Although it is addressed elsewhere in the paper, Table 1 needs proper referencing of the methods Zebra is compared to. Also, it should be made clear that the method mentioned is the vanilla CoDA since you've shown it can easily be adapted for temporal conditionining.
3) By "temporal" conditioning, it is clear that you mean that you are providing a history of previous states to the model. But you need better clarification of what "adaptive" conditioning means:
	- Wouldn't fine-tuning a subset of parameters count as adaptive conditioning ? These two appear to be contrasted in the section 2.2.
	- In appendix A.1, adaptive conditioning seems to be equated to meta-learning and multi-environment learning. The later two are quite different scenarios [3].
4) I remark limited analysis of the results in sections 4.2 and 4.3. Notably, in line 402, the authors hypothesize that leveraging more than a single sample per trajectory would ensure a more robust estimation. What would these samples be, if not the trajectories that are already provided ?
5) The main text and appendix literature on parametric PDE learning is very limited, especially concerning adaptive conditioning and uncertainty quantification. In recent years, several work have attempted to bring UQ into this field, e.g. CoNDP [1], NCF [2], etc.

[1] Stochastic Neural Simulator for Generalizing Dynamical Systems across Environments, IJCAI 2024.

[2] Neural Context Flows for Meta-Learning of Dynamical Systems, ICLR 2024 Workshop on AI4Differential Equations In Science.

[3] Bridging Multi-Task Learning and Meta-Learning: Towards Efficient Training and Effective Adaptation, ICML 2021.


### Other minor issues
- Throughout the paper and especially in section 3.1, the authors mention the "need to convert physical observations into discrete representations". They also point out that their model is trained on numerical simulation data, which is already a form of discretization. This causes confusion and needs better clarity.
- At line 217, and extra "$\times$" is present.
- The stopgradient operator "sg" would benefit from some explanation in the main text.

**Questions:**

1) Concerning the experimentation, why do you find it necessary to "drastically increase the different number of environments compared to previous studies", e.g. by roughly 2 orders of magnitude more than in CoDA ? Also, given that so few environments are used for testing (120 or 12) compared to training, it is not surprising that all methods would perform well in-distribution. It is crucial, however, to know how well Zebra and the baselines perform when the testing environments are taken out-of-distribution (whenever possible based on the varying parameter).

2) Related to Q1, given the need for a high-number of environments, on the Wave-b problem with varying boundary conditions, why weren't *periodic* boundary conditions considered to boost this environment count ? (This is all the more perplexing since the Wave-b dataset seems to be where the model struggles most - cf. Figure 3).

3) How quick is Zebra given that it tokenizes and de-tokenizes states during autoregressive inference (cf. Figure 6) ? I don't believe this walltime metric should be ignored, especially if it led to slower adaptation than say CoDA's 250 gradient steps followed by fast numerical integration.

4) From Table 4, why is Zebra so performant on **2D** zero-shot prediction tasks, and less so on 1D ? Is this somehow related to the structure of the codebook terms leaned in the VQ-VAE, where according to the equation in line 220, each spatial code corresponds to an entry, i.e. the codebook (and by proxy the model parameter count) grows exponentially with the dimensionality of the problem ?
5) Why does the CAPE baseline diverge so often during evaluations ? Is there something that can be done to avoid that ?

---

> ### Author Response · Authors · 2024-11-22
> **Response 1/2**
>
> ## W1: Contribution
> We agree, but this is just to clearly indicate the validation setting of the paper.
>
> ## W2: Referencing
> Thank you. We have added the references in Table 1.
>
> ## W3: Adaptive conditioning
> There might be better wordings. By "adaptive conditioning" we want to highlight that the model is trained to adapt to new examples. In Zebra, the model is trained using in-context examples, in CODA and related gradient based adaptation method, the model is trained to adapt through gradient on a new example at inference step.
>
> Fine tuning usually is used for pre-trained models in order to boost performance on a new task. In this setting the model is not trained to adapt.
>
> You are correct, we will correct the sentence tying multi-environments and meta-learning.
>
> ## W4: Analysis
>
> Yes, this has not been addressed yet due to the page limit. We will revise this section to emphasize the strengths and limitations of each method in both settings.
>
> Thank you, we have corrected the wording.
>
> ## W5: Uncertainty quantification
>
> We wished to focus here on generalization and in-context conditioning. However, as Zebra is a generative model, we also wanted to illustrate the possibility of quantifying uncertainty of the model's predictions. As this is not the main topic of the paper we have placed it in the appendix section. Addressing the UQ topic would require a dedicated paper in itself.
>
> We thank the reviewer for these references, we have added them in the related work section.
>
> Regarding the discretization from vector quantization, we will explicitly add that we refer to the discretization of values and not in space or time.
>
> ## Q1 : Training size and OoD
> We have conducted an analysis on advection and combined equation to understand the influence of the dataset training size, and show that generalizing even within the training range of parameters is challenging when the number of trajectories is reduced. As you suggested, we performed additional experiments in order to quantify this aspect. Please see Figure 14 in section D.3 for more detailed comments.
>
> We have also performed experiments in an OoD context in order to test the capabilities of the models to extrapolate outside of the training parameters range. The results are now in the new Section 4.4. Overall, in this OoD setting, compared to the baselines, Zebra performs best in the one-shot and zero-shot settings. The gradient-based methods fail to generalize even in the one-shot scenario, for which they were designed. MPP on the other hand performs well in the zero-shot setting. Overall, the OoD setting is challenging for all the methods. They generalize to some extent when the distribution is close to the training one, but severe shifts would require that imply a change in the dynamics cannot be handled.
>
> ## Q2: Periodic boundary
> This is indeed a relevant remark. We initially thought that all models would be able to learn the differences between the Neumann and Dirichlet boundary conditions, however it appears that only the in-context methods are capable of doing so in our setting (MPP-in-context to some extent). We had no time during this period to perform the experiments, because this implies retraining a large number of models but we agree that this should induce better generalization.

---

> > ### Author Response · Authors · 2024-11-22
> > **Response 2/2**
> >
> > ## Q3: Inference times
> > Thanks for the remark. We have added a table with the inference times (Table 10 in Appendix D.4, see table below). In its current implementation, Zebra compares favorably to the gradient-based adaptation methods such as CAPE and CODA. In the one-shot adaptation setting, it takes 3s to generate a trajectory on 1D data (vs around 20s for CAPE and CODA) and 21s on 2D data (vs 23s for CODA and CAPE). The two methods have very different bottlenecks. For CAPE and CODA, most of the inference time is allocated to the gradient-based adaptation (here 100 steps), while for Zebra the bottleneck is the autoregressive generation of tokens instead of frames. We have measured a speed of 128 tokens/s. We envisage two strategies to boost the inference speed: (1) use better hardware and optimized code (e.g. flash attention + H100), (2) generate more than one token at once as in [1].
> >
> > Average time in seconds to predict a single trajectory given a context trajectory and an initial condition.
> > - Times include adaptation and forecast for **CODA** and **CAPE**.
> > - Times include encoding, auto-regressive prediction, and decoding for **Zebra**.
> >
> > | Method  | **Advection** | **Vorticity 2D** |
> > |---------|---------------|------------------|
> > | CAPE    | 18s           | 23s             |
> > | CODA    | 31s           | 28s             |
> > | Zebra   | **3s**        | **21s**         |
> >
> > ## Q4 : Results on 2D
> > We believe this is more related to the transformer architecture - which appears to be more powerful than the FNO and UNet employed in CAPE and CODA - and coupled with a simpler optimization scheme. The evidence is that Zebra is not the only method to perform well on the 2D tasks, MPP also achieves good results, and the [CLS]ViT is able to generalize on the wave equation.
> >
> > ## Q5 : CAPE failure modes
> > We did our best to adapt CAPE to our setting and datasets, but we observed that the training was particularly challenging and the adaptation unstable. We also faced difficulties with CODA's adaptation but found a way to prevent divergence by tuning the adaptation learning rate for each dataset.
> >
> >
> > [1] Tian et al. Neurips 2024. Visual Autoregressive Modeling: Scalable Image Generation via Next-Scale Prediction

---

> ### Comment · Reviewer_4Hd3 · 2024-11-23
>
> I thank the authors for such a great effort in addressing my concerns. I understand the difficulties in rerunning some of the experiments I demanded (eg. Q2), but I hope they can be done for the final version of the paper (please do the same for Q5 as well).
>
> The explanations on superior 2D results are much appreciated, although I am not fully convinced. Thank you for other metrics as well, the quicker inference times boat well for the method's adoption by the ICLR community.
>
> Concerning W3, I would argue that fine-tuning on a new task is a form of adaptation, even thought the model wasn't trained with that in mind. On the plus side, I think in-context learning offers a much broader form of adaptation. It is important that our field clearly separates these paradigms with precise definitions at some point.
>
> Concerning W5, thank you for the added results. However, it is quite a shame that uncertainty quantification is now less emphasized. I really liked the parts about __UQ__, and I personally found this to be, more than any other contribution, __the most useful for the community__ -- especially since it builds on techniques from LLM prone to hallucinations. I agree with Reviewer k4yp that $\pm 3\sigma*$ is quite arbitrary and that high CLs should not be anyone's objective, but I don't find the aleatoric/epistemic distinction to be particularly helpful, at least not to the point of removing that section entirely. Regardless, I hope a future work will fully investigate this topic.
>
> All this taken into account, I've updated my various scores, particularly the overall score from 5 to 6.

---

> > ### Author Response · Authors · 2024-11-27
> > **Response**
> >
> > We sincerely thank the reviewer for their thoughtful response and for increasing their score.
> >
> > We apologize for placing the entire section on uncertainty quantification (UQ) in the appendix. As several reviewers highlighted the importance of out-of-distribution (OoD) results, we decided to prioritize those in the main text by replacing the UQ section. However, we recognize that a different structure in the experiments section could better emphasize the strengths and novelty of our method. For the camera-ready version, we suggest organizing the experiment section as follows: (1) Setting details ; (2) One-shot adaptation; (3) OoD; (4) UQ. This way, the most important findings are displayed in the main text.
> >
> > Would this revised structure better highlight the key contributions of our work?

---

> > > ### Comment · Reviewer_4Hd3 · 2024-11-30
> > >
> > > Yes, I believe it would.
> > >
> > > To be precise, if your change means suppressing (or significantly reducing) the current section 4.3 from the main text, then yes, I fully agree. Indeed, with Markovian systems, temporal conditioning doesn't bring that much compared to section 4.2  - Context Adaptation from Similar Trajectories, like I pointed in my original review.
> > >
> > > Thank you,

---

### Official Review · Reviewer_6wsG · 2024-10-31

**Soundness:** 2
**Presentation:** 3
**Contribution:** 1
**Rating:** 3
**Confidence:** 4

**Summary:**

This paper employs in-context learning to address a wide range of PDEs with varying equation parameters. The proposed method demonstrates competitive performance compared to baselines on the evaluated synthetic datasets.

**Strengths:**

1. This paper targets a important problem: generalization within a wide scope of PDEs.
2. This paper is clearly written and esay to understand.

**Weaknesses:**

1. **In-Context Learning:** The novelty of employing in-context learning appears limited. It seems that in-context learning is conceptually similar to “leveraging historical data to condition the neural network.” Tokenizing each frame individually could also enable handling arbitrary-sized context token inputs.
2. **Performance Concerns:** The performance results are not particularly compelling, and the dataset used appears relatively simple. Could you conduct additional experiments on some real-world datasets?
3. **Irregular data:** The paper focuses exclusively on grid-based data, which can be easily tokenized using VQ-VAE, as is common in latent diffusion models (LDM). What distinctions do you perceive between grid-based PDE data and images? Moreover, how would you extend your method to accommodate irregular data?
4. **Out-of-distribution testing:** The test environments are all within the distribution of the training environments, which is uncommon in real-world applications. Could you consider incorporating out-of-distribution test settings, particularly for extrapolation tasks? Since the test cases fall outside the training distribution, the zero-shot inference performance is suboptimal. How would you adapt the trained model to address these cases? Perhaps fine-tuning the VQ-VAE would be required initially, which may complicate adaptation compared to baseline models.

**Questions:**

See Weaknesses.

---

> ### Author Response · Authors · 2024-11-22
> **Response**
>
> ## W1: In-Context Learning
>
> Sorry if there is a misunderstanding. In-context learning and learning using historical data are two clearly distinct settings. In the former, the model learns from examples provided in the context to perform a similar task with a new initial condition. The context includes multiple trajectories, each corresponding to a different initial condition. In the latter, only a single trajectory is considered, and the task is simply to forecast future states based on the past history of this single trajectory.
>
> ## W2: Performance Concerns
>
> We respectfully disagree. The datasets cover a wide range of parameter values, capturing diverse dynamics for each equation. This includes variations in coefficients, initial and boundary conditions, and physical variable values. Consequently, the training task is complex, as reflected in the baseline performance. Regarding the performance, for one-shot adaptation (Table 3), Zebra ranks first in 4 out of 7 datasets and second in the remaining 3. The zero-shot results (Table 4) exhibit a similar pattern. Furthermore, the additional out-of-distribution (OoD) experiments provided in the rebuttal show a consistent trend.
>
> ## W3: Irregular data
>
> This is a very relevant issue. In this study, we focus on generalization with respect to the parameters of the PDE and then considered regular grids for simplicity and for a better control of the experiments and of their conclusions.
> Our model could be naturally extended in many different ways to handle irregular meshes or even point sets. A good candidate could be using the encoder and decoder from [1] or [2] for example.
>
> ## W4: OoD
>
> As you suggested, we have provided out-of-distribution results on Heat, Vorticity and Wave2D in the new section 4.4 in Table 5 (see also table below). We tested all the methods - including Zebra - directly without finetuning, on the one-shot and zero-shot setting. In the one-shot setting, Zebra performs the best followed by MPP and then CODA. This suggests stronger generalization properties of in-context learning vs. gradient-based adaptation. Similarly, in the zero-shot setting, Zebra performs best on 2D while MPP achieves the lowest error on Heat equation. Again, CODA suffers from poor generalization in this setting.
>
> -----
> Test results in relative L2 on the trajectory. `--` indicates inference has diverged.
> For each dataset:
> - **One-shot adaptation**: Conditioning from a similar trajectory.
> - **Zero-shot prediction**: Conditioning from a trajectory history with 2 frames as input.
>
> |            | Heat (One-shot) | Heat (Zero-shot) | Wave 2D (One-shot) | Wave 2D (Zero-shot) | Vorticity 2D (One-shot) | Vorticity 2D (Zero-shot) |
> |------------------|-----------------|------------------|--------------------|---------------------|-------------------------|--------------------------|
> | CAPE            | 0.47            | 0.33             | --                 | --                  | --                      | --                       |
> | CODA            | 1.03            | 0.66             | 1.51               | 1.32                | 1.71                    | 1.59                     |
> | MPP[2]          | --              | **0.19**         | --                 | 0.70                | --                      | _0.22_                   |
> | MPP-in-context  | 0.52            | _0.32_           | **0.68**           | _0.66_              | _0.30_                  | 0.28                     |
> | Zebra           | **0.15**        | 0.34             | **0.68**           | **0.55**            | **0.24**                | **0.21**                 |
>
> [1] Serrano et al. Neurips 2024. AROMA: Preserving Spatial Structure for Latent PDE Modeling with Local Neural Fields
>
> [2] Zhou et al. 2024. Text2PDE: Latent Diffusion Models for Accessible Physics Simulation

---

> > ### Comment · Reviewer_6wsG · 2024-11-25
> >
> > Thank you for your rebuttal. However, my concerns remain. I would like to further elaborate on my concerns to ensure clarity.
> >
> > 1. **The In-Context Learning (ICL) Paradigm:** This paradigm, as proposed in NLP, is fundamentally built upon the existence of a large pre-trained model trained on large-scale text data. However, in the context of PDE solving, the existence of such a foundational model remains an open problem. Without such a model, what is the inherent value of ICL? Can all PDEs be solved effectively by simply providing a few examples for the model? For instance, can a model trained on Navier-Stokes equations directly solve a Darcy equation through ICL?
> >
> > 2. **Implementation of ICL:** From an implementation perspective, ICL seems no different from solving problems by “utilizing history.” A complete history can simply be regarded as a specific case of multiple examples of the same equation. The paper does not propose any new techniques tailored to the PDE domain nor sufficiently justify why techniques developed in other domains do not require modifications for application in this field. This raises concerns about the technical novelty of the work.
> >
> > 3. **Experimental Scope:** The experiments conducted in the paper are limited to a family of equations controlled by a single coefficient, making the task relatively simple. I would like to see this method tested on more challenging tasks, such as mixed training on numerous distinct equations, followed by using ICL to solve new equations.
> >
> > In summary, I believe the full potential of ICL can only be realized with the development of a large foundational model for PDE solving. Merely proposing the use of ICL in this context is insufficient to demonstrate significant contributions.

---

> > > ### Author Response · Authors · 2024-11-27
> > > **Response**
> > >
> > > Thank you for taking the time to share further insights into your concerns. We appreciate the opportunity to clarify and elaborate on the contributions and significance of our work. While we value your perspective, we respectfully disagree with some of your arguments and conclusions.
> > >
> > > ### 1. Addressing the Problem of Parametric PDE
> > >
> > > Our work focuses on the challenging and largely open problem of learning surrogate models for solving parametric PDEs. This problem is distinct from the development of foundation models, as highlighted by recent advancements in the field. [1,2,3,5,6]
> > >
> > > - **ICL and Foundation Models Are Distinct Paradigms**
> > >   The distinction between in-context learning (ICL) and foundation models is well-established in the NLP literature. For instance, [4] demonstrates that ICL can be highly effective even in scenarios with limited data and smaller models. Regarding your question, *"Can a model trained on Navier-Stokes equations directly solve a Darcy equation through ICL?"*—our response is "probably no." Addressing cross-physics generalization falls under the foundation model paradigm, which is outside the scope of our paper. Instead, our work aligns with recent studies [4, 5] on in-context learning for PDEs, focusing on solving parametric equations within the same family.
> > >
> > > - **ICL and Training on History Are Fundamentally Different**
> > >   In our ICL framework, the context is composed of different instances of the same equation, representing various dynamics. This is distinct from training on temporal histories. Our approach is specifically tailored for the physics domain and the training of PDE surrogates. The experiments presented in the paper demonstrate that our model performs on par with or better than recent state-of-the-art adaptation frameworks.
> > >
> > > ### 2. Experimental Setup and Scope
> > >
> > > You mentioned, *"The experiments conducted in the paper are limited to a family of equations controlled by a single coefficient, making the task relatively simple."* We respectfully disagree, as this statement contradicts the experimental setup described in the paper. Our experiments cover seven families of equations with parameters that go beyond a single coefficient. These include:
> > >
> > > - PDE coefficients (up to 3 parameters)
> > > - Forcing terms (15 parameters)
> > > - Initial and boundary conditions (2 parameters)
> > >
> > > This results in approximately 1,200 unique instances for a single equation, encompassing a diverse range of parameter values. For further details, please refer to Table 2 in Section 3 of the paper.
> > >
> > > ### 3. Robustness and Additional Evaluations
> > >
> > > As noted in our earlier response, we have extended our evaluations to include out-of-distribution (OoD) scenarios, which highlight the robustness of our model. In addition, we conducted experiments on turbulent flows with viscosity in the range of \( [10^{-5}, 10^{-4}] \). The results, presented below, underscore Zebra's ability to handle changes in viscosity for vorticity tasks.  We also added a visualization in the main section of the paper (Figure 4) and additional plots in Section E.
> > >
> > > | Model            | Zero-shot (with 2 frames) | One-shot (with 1 frame) |
> > > |-------------------|----------------------|--------------------|
> > > | MPP-in-context    | 0.355               | 0.368              |
> > > | Zebra             | **0.308**           | **0.317**          |
> > >
> > > These evaluations reinforce the versatility and reliability of our approach.
> > >
> > >
> > > [1]  Desmond Nzoyem et al., 2024. Neural Context Flows for Meta-Learning of Dynamical Systems.
> > >
> > > [2] Liu et al., 2024. Stochastic Neural Simulator for Generalizing Dynamical Systems across Environments.
> > >
> > > [3] Musekamp et al, 2024. Active Learning for neural pde solvers.
> > >
> > > [4] Eldan and Li, 2023. TinyStories: How Small Can Language Models Be and Still Speak Coherent English?
> > >
> > > [5] Yang et al., 2023. In-Context Operator Learning with Data Prompts for Differential Equation Problems.
> > >
> > > [6] Chen et al., NeurIPS 2024. Data-Efficient Operator Learning via Unsupervised Pretraining and In-Context Learning.

---

### Official Review · Reviewer_k4yp · 2024-10-31

**Soundness:** 2
**Presentation:** 3
**Contribution:** 3
**Rating:** 6
**Confidence:** 3

**Summary:**

The manuscript proposes a data-driven method for solving systems of PDEs given an initial condition and/or trajectories of previous states as context, leveraging in-context learning. The proposed generative model is capable of zero-shot and one-shot learning, generalizing across different parameterizations of a given system of PDEs. To achieve this, the approach combines a VQ-VAE to represent physical states (given by images, i.e., regular meshes) and a transformer architecture trained for next-state prediction.

**Strengths:**

The paper is well written and quite accessible. To the best of my knowledge, the approach is novel, albeit being just a combination of a VQ-VAE and a transformer model. The quality of the research seems to be on a high level, as the approach appears valid and the experiments appear well designed and executed. In the experiments, the proposed approach performs excellently in terms of the relative L2 error (but some experimental details are not clear).

**Weaknesses:**

I have a few concerns that prevent me from assigning a higher score, and I would appreciate hearing the authors' thoughts on them:
- I am missing some references from the literature that may be relevant in this context. For example, how does Zebra compare to DeepONets (which are mesh free) and to MeshGraphNets [1] (which can use irregular meshes)? I assume that DeepONets and MeshGraphNets can be used to solve similar problems, even if the underlying approach is different. I would thus appreciate a numerical comparison, or at least a paragraph why such a comparison is not reasonable. This is important to put Zebra in context with the existing literature.
- In Section 3.3, it is mentioned that the transformer has a causal structure. This is not clear a priori, as the attention mechanism can in general be acausal (if I understood correctly). I would appreciate a few clarifying statements here. How was this causal structure enforced?
- In the experiments, some aspects are not fully clear.
  - For example, in Sec. 4.2, does "the same underlying dynamics as the target" refer to the same parameter set $\mu$, or to something different? In the same section, does "Predictions are generated using a single sample" refer to the fact that, the future trajectory that has to be predicted has length 1? Or does it mean something else (and how would then multiple samples be used in the generative model)? In the same section, the results from Fig. 2 are not very strong as they do not exhibit a clear trend, suggesting that these figures can be moved to the appendix.
  - In Section 4.3, Fig. 3 could benefit from including the same analysis using CAPE and CODA, thus enabling a more fair comparison between these methods.
  - In both Sections 4.2 and 4.3, it may be a good idea to display ground truth trajectories together with predictions from all approaches. I briefly saw that the predictions of Zebra are included in the appendix, but I think one or two exemplary results (with those from the competing approaches) could be moved to the main text.
  - Section 4.4 is not entirely clear. On the one hand, it is not clear whether the quantified uncertainty is epistemic (resulting from too little training data) or aleatoric (influenced by inherent noise). The fact that it relies on $\tau$ suggests at least an aleatoric component. From that perspective, the usefulness of this is not clear. It is little surprising that for a large temperature $\tau$ the output is more random, hence there is a larger probability that the true trajectory is drawn by chance. Given that $\sigma_*$ can be made arbitrarily large via $\tau$, it is not clear to me if a high confidence level can be a meaningful objective. I suggest to clarify this, or to remove this section entirely.


### Minor:
- Line 068: "can can"
- Line 217: There is a double $\times$ in the exponent.
- What is the meaning of $sg[Z_q^t]$ in line 221? The function $sg$ was never introduced.
- Line 495: "traedoff"
- Figure 5: The axis labels and plot titles are not clear. What is the "rollout loss"?

[1]: https://arxiv.org/abs/2010.03409

**Questions:**

- How does the proposed approach compare to DeepONets (which are mesh free) and to MeshGraphNets [1] (which can use irregular meshes)?
- How was the causal structure enforced for the transformer mentioned in Sec. 3.3?
- In Sec. 4.2, does "the same underlying dynamics as the target" refer to the same parameter set $\mu$?
- Does "Predictions are generated using a single sample" refer to the fact that, the future trajectory that has to be predicted has length 1?
- Is the uncertainty quantified in Sec. 4.4 epistemic, aleatoric, or both?

---

> ### Author Response · Authors · 2024-11-22
> **Response**
>
> ## W0: Novelty
> We would like to highlight that the originality of our framework does not lie in the proposed architecture, which leverages existing modules, although the combination is new for solving PDE problems. The originality of our framework is to propose in-context generative pretraining to allow adaptation without gradient steps for PDE solving.  This is why we position our framework versus existing adaptation frameworks (adaptive conditioning and temporal conditioning) and show that our in-context model (i) outperforms SOTA baselines in this context, (ii) does not require gradient-based fine tuning at inference as these baselines do.
>
> ## W1: Irregular meshes
> The problem we address is generalization. We have then focused on regular grids for simplicity. We agree that the model could be extended in many different ways to handle irregular meshes or even point sets. A good candidate could be using the encoder and decoder from [1] or [2] for example.
>
> ## W2 : Causal mask
> We use a transformer based on Llama's architecture, which uses by default a causal attention mask (as in GPT, Gemini, etc.) where the token $s_k$ can only attend previous tokens $s_{< k}$. Note that we employ a causal scheme as we pretrain the transformer to predict the next token, however it could be beneficial to explore other pretraining strategies, such as the one proposed in [3].
>
> ## W3: Misc.
> * Yes, "the same underlying dynamics as the target" means the context trajectory and target trajectory share the same parameters with different initial conditions. We will clarify this wording in the final version. "Predictions are generated using a single sample" means that we predict the trajectory by generating a single sequence of tokens.  We generate one trajectory per initial condition.
>
> * Trends on Figure 2. You are right but, Figure 2 illustrates that the context size plays a role in the model performance. This is not the case for example for gradient-based adaptation where the performance saturates fast [4]. We think that this is a distinctive feature that we would like to highlight.
>
> * Figure 3 with CAPE and CODA. We agree, we will try to finish the experiments for the final version.
>
> * We will provide more visualizations in the final version of the manuscript.
>
> * This section aims at a tentative explanation of the interpretability of the generative aspect of our model. To better understand the type of uncertainty of our model, we have conducted an analysis where we provide more or less information in the context at inference. Please see the additional Table 8 of Appendix D.1 in the revised version (see also table below). We have systematically observed a reduction of the standard deviation of the predictions when including more information, which experimentally suggests that some of the uncertainty is epistemic. Although there remains aleatoric uncertainty as our model is not perfect. The ability to predict trajectory distributions is a property of our generative model, but is shown here as an illustration. The topic would require a dedicated analysis which is beyond our scope. We have then moved this section in the appendix (Appendix D.1) with a mention in the main text.
> ----
> Conditioning from a trajectory example and 1 frame or 2 frames as initial conditions.
> Metrics include:
> - **Relative L2 loss** (average accuracy).
> - **Relative standard deviation** (average spread around the average prediction).
> The temperature is fixed at 0.1.
>
> | Metric          |  # Frames    | **Advection** | **Heat** | **Burgers** | **Wave b** | **Combined** |
> |------------------|-----------|---------------|----------|-------------|------------|--------------|
> | **Rel. L2 ** | 1 frame   | 0.006         | 0.156    | 0.115       | 0.154      | 0.008        |
> | **Rel. L2** | 2 frames  | 0.004         | 0.047    | 0.052       | 0.075      | 0.005        |
> | **Rel. Std.**     | 1 frame   | 0.003         | 0.062    | 0.048       | 0.074      | 0.005        |
> | **Rel. Std.**     | 2 frames  | 0.002         | 0.019    | 0.018       | 0.040      | 0.003        |
>
> ----
> ## W4 (minor)
> We thank the reviewer for spotting these errors, and have fixed them.
>
> [1] Serrano et al. Neurips 2024. AROMA: Preserving Spatial Structure for Latent PDE Modeling with Local Neural Fields
>
> [2] Zhou et al. 2024. Text2PDE: Latent Diffusion Models for Accessible Physics Simulation
>
> [3] Tian et al. Neurips 2024. Visual Autoregressive Modeling: Scalable Image Generation via Next-Scale Prediction
>
> [4] Kassaï Koupaï et al., Neurips 2024. GEPS: Boosting Generalization in Parametric PDE Neural Solvers through Adaptive Conditioning.

---

> > ### Comment · Reviewer_k4yp · 2024-11-23
> >
> > I thank the authors for the clarifications, they indeed show the paper in a new light. I appreciate the willingness to update the manuscript with me experiments with CAPE and CODA. Despite the response in W0 (and the response to another reviewer claiming that the main contribution lies in the generative nature of Zebra) I think that the main text should contain a a few words about why DeepONets and MeshGraphNets cannot be reasonably compared to. I think especially DeepONets are capable of learning operators and may also be able to generalize across different dynamics (but of course I can be wrong here).
> >
> > I slightly raised my score but lowered my confidence. I will reconsider my score in der reviewer discussion phase.

---

> > > ### Author Response · Authors · 2024-11-27
> > > **Response**
> > >
> > > Thank you for your insightful feedback. We agree with the reviewer that positioning our approach alongside methods such as DeepONet and MeshGraphNet would help clarify our contributions to the field. To address this, we will add a detailed discussion on operator learning in the related work section.
> > >
> > > Additionally, we recognize that operator learning architectures like DeepONet can be directly applied in scenarios where a trajectory history is available (e.g., using two frames to predict the next one). To this end, we have conducted additional experiments to evaluate DeepONet in our zero-shot settings. However, it is important to note that DeepONet cannot be applied to the adaptation scenario (one-shot setting) that we consider in this study, which further highlights the distinction between our approach and operator learning models. From these preliminary results, we can conclude that DeepONet can be effectively used with temporal conditioning for the zero-shot setting for 1D but struggles more to model 2D-dynamics. We will add these results in the final version.
> > >
> > >
> > > | Method               | **Advection** | **Heat** | **Burgers** | **Wave b** | **Combined** | **Wave 2D** | **Vorticity 2D** |
> > > |----------------------|---------------|----------|-------------|------------|--------------|--------------|------------------|
> > > | **DeepONet**   | 0.008        | 0.223    | 0.151       | --      | 0.040       | 1.15        | 0.875            |
> > >
> > >
> > > ## Additional paragraph on Operator Learning and Irregular grids
> > > - to be added in the related work section.
> > >
> > > **Operator learning** has emerged as a key paradigm for solving PDEs using neural networks. In this framework, the objective is to learn a mapping from an input function space to an output function space, independent of the spatial discretization. In other words, a good operator learning architecture should be capable of handling changes in the input discretization, demonstrating robustness (i.e., resolution invariance or minimal sensitivity to resolution changes), and enabling queries of the output function at arbitrary points within the domain. While this focus on grid invariance represents an important direction in the field, it differs fundamentally from the approach we tackle in this study.
> > > Among the most prominent examples of operator learning architectures are the DeepONet [1] and Fourier Neural Operator [2], which have seen widespread use across scientific fields. Additionally, graph-based architectures such as [3] have been proposed to address generalization challenges related to geometry and discretization. These operator-learning models can also be coupled with adaptation schemes, as demonstrated in frameworks such as CODA and CAPE, to further enhance performance and flexibility.
> > >
> > > [1] Lu et al 2019, DeepONet: Learning nonlinear operators for identifying differential equations based on the universal approximation theorem of operators
> > > [2] Li et al 2020, Fourier Neural Operator for Parametric Partial Differential Equations
> > > [3] Pfaff et al, 2020, Learning Mesh-Based Simulation with Graph Networks

---

### Official Review · Reviewer_oj6X · 2024-10-31

**Soundness:** 3
**Presentation:** 3
**Contribution:** 3
**Rating:** 8
**Confidence:** 3

**Summary:**

The paper proposes a foundation model to solve time-dependent parametric PDEs. Their architecture consists of a learned CNN encoder-decoder which makes the PDE trajectories into a latent sequence space, and an autoregressive transformer which learns the solution operator on the latent space. The transformer is trained using the cross-entropy loss on the latent space. The model is trained on several classes of parametric PDEs, such as heat, advection, and burgers equations, whose parameters are varied during training. The trained model is then able to perform inference for new equations without any parameter updates. They compare their results with baselines, considering one-shot adaptation, in-context learning, and temporal conditioning. They also explore the ability of the generative model to perform uncertainty quantification on various statistics of the PDE trajectories.

**Strengths:**

1. The authors take on an important challenge in scientific machine learning, developing reliable foundation models for PDEs. Their method has the additional advantage of not requiring any parameter updates at inference time. Their architecture appears novel and their results are competitive with the compared baselines.

2. Experiments are conducted on a fairly diverse class of PDEs exhibiting different physical phenomena and their results are achieved using a relatively small amount of training data compared to other scientific foundation models.

3. The paper is well-written and easy to follow considering its technical nature.

**Weaknesses:**

1.The main limitation I find in the paper is that they do not provide any experiments demonstrating the out-of-distribution generalizability of their ICL solver. While the authors vary the PDE parameters between training and inference, it seems that the distribution of the parameters remains the same. For example, for the heat equation, the coefficients of the forcing terms are generated from uniform distributions in a fixed range. It would be interesting to shift the distribution of these coefficients at test time and compare the performance of the pre-trained model. For a more extreme shift, one could use the pre-trained model to generate solutions corresponding to new PDEs that were not seen during pre-training using few-shot conditioning.

2. The decoder does not seem to reconstruct the trajectories very well. However, the authors address this in the main text and I see this as only a minor weakness.

I still think the paper should be accepted, because their architecture appears novel in the Sci-FM literature and it may inspire further developments.

**Questions:**

1. Have you compared your generative solver with the In-Context Operator Network (ICON) architecture introduced in [1,2]? It seems they are not directly comparable because with ICON, the goal is to generate the PDE solution given an initial condition, whereas your model also generates the initial condition. However, I think it is worth discussing the key technical differences between your model and ICON, and, if possible, outline how a direct experimental comparison might be set up. In addition, Table 1 seems to suggest that Zebra is the first model which attempts to solve PDEs in-context. Could you clarify if this claim of novelty is specifically about the combination of in-context learning and generative modeling in PDEs?

2. Could you explain your rationale for using the L^2 norm as an evaluation metric for the generated solutions? I would imagine, since you are predicting a PDE solution, that it might be natural to use the H^1 norm as a metric, or other metrics which are tailored to PDE data.

3. Do you have a sense of how your solution error depends on the parameters of your VQVAE, e.g., the number of codes or the number of parameters of the CNN encoder? You mention that in its current construction, the decoder is limited in its ability to accurately reconstruct the trajectories. I am curious because in [3], the authors identify that the test error follows a U-curve with respect to the spatial discretization size for in-context learning elliptic equations. While the setting of [3] is more mathematically idealized than yours, I wonder if your architecture might exhibit a similar phenomenon.

References:

[1] Yang, Liu, et al. "In-context operator learning with data prompts for differential equation problems." Proceedings of the National Academy of Sciences 120.39 (2023): e2310142120.

[2] Yang, Liu, and Stanley J. Osher. "Pde generalization of in-context operator networks: A study on 1d scalar nonlinear conservation laws." arXiv preprint arXiv:2401.07364 (2024)

[3] Cole, Frank, et al. "Provable In-Context Learning of Linear Systems and Linear Elliptic PDEs with Transformers." arXiv preprint arXiv:2409.12293 (2024).

---

> ### Author Response · Authors · 2024-11-22
> **Response**
>
> ## W1: OoD testing
>
> We thank the reviewer for their suggestion on evaluating the models also on out-of-distribution settings. To this end we have included additional results concerning zebra and the main baselines on different scenarios. We have conducted several out-of-distribution tests: (1) forcing terms for heat equation, (2) smaller viscosity for viscosity 2D (3) larger celerity and damping terms for wave 2D. Overall the performance of all the methods degrade in OoD settings compared to in-distribution evaluations, and they extrapolate to parametric regimes that are close to the training distribution. For one-shot evaluation ZEBRA performs best, and for zero-shot it is best on 2 out of 3 evaluations. Please see table below.
> Note that interpolating in a large range of parameters as we did in our experiments is already a challenging task, that few models are able to handle.
>
> ------------------------------
> Test results in relative L2 on the trajectory. `--` indicates inference has diverged.
> For each dataset:
> - One-shot adaptation: Conditioning from a similar trajectory.
> - Zero-shot prediction: Conditioning from a trajectory history with 2 frames as input.
>
> |            | Heat (One-shot) | Heat (Zero-shot) | Wave 2D (One-shot) | Wave 2D (Zero-shot) | Vorticity 2D (One-shot) | Vorticity 2D (Zero-shot) |
> |------------------|-----------------|------------------|--------------------|---------------------|-------------------------|--------------------------|
> | CAPE            | 0.47            | 0.33             | --                 | --                  | --                      | --                       |
> | CODA            | 1.03            | 0.66             | 1.51               | 1.32                | 1.71                    | 1.59                     |
> | MPP[2]          | --              | **0.19**         | --                 | 0.70                | --                      | _0.22_                   |
> | MPP-in-context  | 0.52            | _0.32_           | **0.68**           | _0.66_              | _0.30_                  | 0.28                     |
> | Zebra           | **0.15**        | 0.34             | **0.68**           | **0.55**            | **0.24**                | **0.21**                 |
> ------------------------------
>
> ## W2 : Decoder reconstruction
>
> In most cases, the decoder can reconstruct accurately the frames. The relative L2 reconstruction loss goes from 3e-4 on advection, to 1.7e-2 on vorticity (please see table below). We have provided an additional Table 12 in Appendix D.6, where we display the relative errors between the reconstruction and the ground truth for each dataset used in the paper.
>
>
> | Method                | Advection | Heat | Burgers | Wave b | Combined | Wave 2D | Vorticity 2D |
> |-----------------------|---------------|----------|-------------|------------|--------------|--------------|------------------|
> | VQVAE of Zebra      | 0.0003        | 0.0019   | 0.0016      | 0.0011     | 0.0022       | 0.010        | 0.017           |
>
>
> ## Q1 : ICON
> We were not able to compare our method to ICON, because we did not succeed to make it work on our examples. ICON cannot handle the spatio-temporal forecasting task dealt with in the paper.
> You are correct in pointing out similarities, however the two models target different applications and rely on different principles. ICON is an encoder-decoder model and not a generative mode. It generates outputs using cross-attention conditioned on the context embeddings. The output is generated pointwise.  There is no explicit modeling of process dynamics there. ZEBRA is a generative model (like GPT) that relies on an encode-process-decode framework. It learns to model the dynamics in a latent space through the process module. A more precise claim could be that ZEBRA is the first in-context generative model for PDEs.
>
> ## Q2 : Sobolev norm
> We thank the reviewer for this suggestion. Indeed, the sobolev norm is prominent in PDE analysis, and has been proposed for training neural operators recently [1]. However, here we follow the current practice in the literature of reporting the metrics with the L2 norm.
>
> ## Q3 : VQVAE influence
> We thank the reviewer for the suggested analysis. We have conducted an ablation to understand the influence of the codebook size (which plays the role of the number of basis used for projection) on the reconstruction and prediction loss. We also observe a U shaped curve as a function of the size of the codebook. Please find the results in Appendix D.5 of the revised version, in Table 11 and Figure 15.
>
> [1] Cho et al, 2024. Sobolev Training for Operator Learning

---

> > ### Comment · Reviewer_oj6X · 2024-11-22
> > **Response to authors**
> >
> > I thank the reviewers for thoroughly answering my questions, and for providing substantial additions to the paper during a short rebuttal period. I am impressed with the performance of the model under distribution shifts, and I think the U-curve with respect to the VQVAE parameters is interesting and hopefully insightful for other practitioners. Due to the significant improvements to what was already a fairly strong paper, I have raised my score.
> >
> > Regarding Q1, the explanation on the differences between Zebra and ICON makes sense and I think it is fair not to compare the two methods. If possible, I would still encourage the authors to highlight this difference in the related work section (or even in the appendix). I think that spelling this out clearly would highlight the novelty of the paper compared to previous works and make the work more accessible (e.g., to researchers from PDEs who have only seen in-context learning in the setting of ICON).

---

> > > ### Author Response · Authors · 2024-11-25
> > > **Response**
> > >
> > > Thank you for your valuable comments and suggestions, which we believe have significantly improved the quality of our manuscript.
> > >
> > > Regarding ICON, we will explicitly highlight the differences in the camera-ready version between the two approaches to better emphasize the novelty and distinct contributions of our work.

---

### Author Response · Authors · 2024-11-22
**General response**

We sincerely thank the reviewers for their thoughtful and constructive feedback. Based on your comments, we have revised our manuscript and included the following key updates:

- **Out-of-Distribution (OoD) Testing** (oj6X, 6wsG): Results for OoD evaluation on three datasets (Heat, Vorticity, Wave) are now included in Section 4.4, covering both zero-shot and one-shot scenarios.
- **MPP-in-Context**: (UXkn) In response to UXkn's suggestion, we trained MPP on all datasets using an in-context setup similar to ours. Results are presented in Sections 4.2, 4.3, and 4.4.
- **Scaling Analysis**: (4Hd3) Appendix D.3 features an analysis of how the size of the training dataset impacts test generalization error for new initial conditions and parameters within the distribution.
- **Codebook Size Influence**: (oj6X) An ablation study on the role of codebook size is included in Appendix D.5.
- **Inference Times**: (4Hd3) A comparison of inference times between gradient-based adaptation methods and Zebra is detailed in Appendix D.4.
- **Uncertainty Quantification**: (k4yp) The uncertainty analysis has been enriched and is presented in Appendix D.1, with additional results in Table 8 exploring the effect of varying context information.
- **Reconstruction Errors**: (oj6X) Appendix D.6 reports the reconstruction errors of the VQVAE across all datasets.

From these results, we draw the following insights:

- **Sensitivity to Distribution Shifts**: All methods are affected by distribution shifts, with gradient-based methods being the most sensitive. Zebra outperforms others in the one-shot setting and remains competitive in zero-shot. MPP shows robustness in the zero-shot setting.
- **In-Context Learning with MPP**: Adding context examples is possible with MPP, but the results show that (i) Zebra outperforms MPP in the one-shot setting, and (ii) MPP-in-context performs worse than MPP trained with two frames in the zero-shot setting.
This suggests that in-context learning as proposed here is not directly extendable to MPP and would require a more involved adaptation, while they appear effective for generative models like Zebra.

- **Generalization and Training Data Size**: Zebra generalizes well in-domain, but requires a large number of training trajectories.
- **Codebook Size**: The size of the codebook is critical to the framework. Larger codebooks improve signal reconstruction but make dynamics modeling more complex and hence results in a U-shaped performance curve.

- **Inference Efficiency**: Zebra is faster than CAPE and CODA for one-shot adaptation in 1D and slightly faster in 2D.
- **Input Information and Uncertainty**: Adding more context information consistently reduces the standard deviation in predictions.

We hope these new results address the reviewers’ concerns and demonstrate the strengths of our approach.

---

### Meta-Review · Area_Chair_CEcK · 2024-12-20

**Metareview:**

This paper presents a novel approach to solving parametric PDEs using in-context learning, a technique inspired by large language models, with the authors proposing a generative model, Zebra, which leverages a VQ-VAE framework and an autoregressive architecture to adapt to new environments with few-shot learning. While the reviewers have raised several concerns, including the novelty of the approach, the effectiveness of in-context learning, and the suitability of the benchmarks, the authors have addressed some of these concerns, highlighting Zebra's improved performance over some baseline models, its robustness in out-of-distribution scenarios, and its ability to leverage information from context examples throughout trajectory generation. However, despite these efforts, concerns remain regarding the limited innovation of the approach, the inconsistent demonstration of in-context learning advantages, and the potential unsuitability of the benchmarks for evaluating this technique, as the parameters of a new trajectory can often be inferred from few frames. Hence, after discussion with the authors and among themselves, the reviewers find the paper to still be very borderline, with three reviewers leaning towards acceptance and two towards rejection. We will therefore need to reject the paper in its current form. We would still like to encourage the authors to resubmit an improved version of the paper in the future.

**Additional Comments On Reviewer Discussion:**

see above

---

### Decision · Program_Chairs · 2025-01-22

Reject